# NATADIFF: ADVERSARIAL BOUNDARY GUIDANCE FOR NATURAL ADVERSARIAL DIFFUSION

**Max Collins**
School of Physics, Maths and Computing
The University of Western Australia
Perth, WA 6009
`max.collins@research.uwa.edu.au`

**Jordan Vice**
School of Physics, Maths and Computing
The University of Western Australia
Perth, WA 6009
`jordan.vice@uwa.edu.au`

**Tim French**
School of Physics, Maths and Computing
The University of Western Australia
Perth, WA 6009
`tim.french@uwa.edu.au`

**Ajmal Mian**
School of Physics, Maths and Computing
The University of Western Australia
Perth, WA 6009
`ajmal.mian@uwa.edu.au`

## ABSTRACT

Adversarial samples exploit irregularities in the manifold "learned" by deep learning models to cause misclassifications. The study of these adversarial samples provides insight into the features a model uses to classify inputs, which can be leveraged to improve robustness against future attacks. However, much of the existing literature focuses on constrained adversarial samples, which do not accurately reflect test-time errors encountered in real-world settings. To address this, we propose 'NatADiff', an adversarial sampling scheme that leverages denoising diffusion to generate natural adversarial samples. Our approach is based on the observation that natural adversarial samples frequently contain structural elements from the adversarial class. Deep learning models can exploit these structural elements to shortcut the classification process, rather than learning to genuinely distinguish between classes. To leverage this behavior, we guide the diffusion trajectory towards the intersection of the true and adversarial classes, combining time-travel sampling with augmented classifier guidance to enhance attack transferability while preserving image quality. Our method achieves comparable white-box attack success rates to current state-of-the-art techniques, while exhibiting significantly higher transferability across model architectures and improved alignment with natural test-time errors as measured by FID. These results demonstrate that NatADiff produces adversarial samples that not only transfer more effectively across models, but more faithfully resemble naturally occurring test-time errors when compared with other generative adversarial sampling schemes.

## 1 INTRODUCTION

Deep learning models can react unpredictably when there is domain difference between training and test data (Szegedy et al., 2014; Goodfellow et al., 2015). *Constrained* adversarial attacks exploit this vulnerability, adding visually imperceptible pixel-level perturbations to deliberately *fool* models into misclassification (Szegedy et al., 2014; Goodfellow et al., 2015; Madry et al., 2018; Croce & Hein, 2020). More recently, *unconstrained* adversarial attacks have been proposed which allow for unrestricted perturbation magnitudes, provided the resulting adversarial image lies sufficiently close to the natural image manifold (Song et al., 2018; Chen et al., 2023a;b).

Defences to these attacks have been proposed (Szegedy et al., 2014; Madry et al., 2018; Gu & Rigazio, 2015; Xu et al., 2018; Samangouei et al., 2018; Nie et al., 2022); however, they largely target attacks formed by adding perturbations to natural images–overlooking the existence of *natural* adversarial

samples. Natural adversarial samples are more commonly known as test-time errors, and they represent the strongest class of unconstrained adversarial attack, as they are valid (perturbation-free and naturally occurring) model inputs that are erroneously classified (Hendrycks et al., 2021). The absence of an adversarial perturbation renders many defensive measures ineffective (Agarwal et al., 2022). Furthermore, natural adversarial samples have been widely studied in the literature, and they have been found to exhibit high transferability–where multiple classifiers incorrectly classify the same sample (Hendrycks et al., 2021). It is hypothesized that this is caused by classifiers independently learning to rely on the same erroneous contextual cues to shortcut classification and reduce training losses without generalising to the underlying task (Hendrycks et al., 2021; Geirhos et al., 2020; Arjovsky et al., 2020).

Generating natural adversarial samples offers an opportunity to better understand the mechanisms underpinning test-time errors. Prior work has sought to achieve this by using classifier gradients to perturb the sampling process of generative adversarial networks (GANs) and denoising diffusion models (Song et al., 2018; Chen et al., 2023b; Dai et al., 2024; Chen et al., 2025). However, GAN-based approaches lack theoretical justification for perturbing the sample path, and doing so often degrades image quality (Karras et al., 2019; Abdal et al., 2019). Alternatively, directly injecting classifier gradients into the diffusion sampling trajectory can result in generating constrained adversarial samples (Vaeth et al., 2024; Shen et al., 2024) (see Figure 1 (c)). Moreover, existing methods do not account for the link between learned erroneous contextual cues and test-time errors.

We propose *NatADiff*, a highly transferable, diffusion-based (Ho et al., 2020; Song et al., 2021a) adversarial sample generation method. NatADiff leverages the link between contextual cues and test-time errors by guiding the diffusion sampling trajectory towards the intersection of the adversarial and true classes, a technique we define as "adversarial boundary guidance". Additionally, we incorporate classifier augmentations to reduce the strength of the constrained adversarial perturbation and to further guide the sampling trajectory towards regions of the image manifold that incorporate features from the adversarial class. We find that NatADiff-generated samples achieve comparable white-box (same target and victim classifier) attack success rates to current state-of-the-art adversarial attacks, while exhibiting significantly higher transferability (different target and victim classifier) across models. Furthermore, samples generated using NatADiff align more closely with known test-time errors (with respect to their Fréchet inception distance (FID) (Fréchet, 1957)) than those generated through adversarial classifier guidance alone (Dai et al., 2024). These results demonstrate that NatADiff produces adversarial samples that not only transfer more effectively across models, but more faithfully resemble naturally occurring test-time errors. To summarize our contributions: (i) We propose NatADiff, incorporating classifier transformations, gradient normalization, and time-travel sampling (Shen et al., 2024; Lugmayr et al., 2022; Yu et al., 2023) to improve adversarial classifier guidance and image quality; (ii) We design an adversarial boundary guidance algorithm to reliably navigate the complex, learned manifold, allowing us to generate natural adversarial samples with significantly higher transferability than existing approaches. (iii) We explore how convolution and transformer based classifiers perceive natural adversarial samples, exposing interesting properties of the feature representations learned by deep learning models.

## 2 DEFINITIONS AND PRELIMINARIES

**Constrained, unconstrained, and natural adversarial samples.** Broadly speaking there are three categories of adversarial sample: unconstrained, constrained, and natural. Let $f : \mathcal{I}_{\mathcal{U}} \to \mathcal{Y}$ be a trained image classifier, $\mathcal{I}_{\mathcal{U}}$ be the set of allowable image inputs, $\mathcal{I}_{\mathcal{N}} \subseteq \mathcal{I}_{\mathcal{U}}$ be the set of natural images, $\mathcal{Y}$ be the set of image classification labels, and $\mathcal{O} : \mathcal{I}_{\mathcal{U}} \to \mathcal{Y}$ be an oracle ("perfect" human) classifier. Unconstrained adversarial samples require only that the image is misclassified: $\mathcal{A}_U \triangleq \{x \in \mathcal{I}_{\mathcal{U}} : f(x) \neq \mathcal{O}(x)\}$ (Song et al., 2018). Constrained adversarial samples are restricted to an $\epsilon$-neighbourhood about some natural image: $\mathcal{A}_C \triangleq \{x + \delta \in \mathcal{I}_{\mathcal{U}} : x \in \mathcal{I}_{\mathcal{N}}, \|\delta\|_p \leq \epsilon, f(x + \delta) \neq \mathcal{O}(x + \delta)\}$ (Szegedy et al., 2014). Natural adversarial samples are natural images that are misclassified: $\mathcal{A}_N \triangleq \{x \in \mathcal{I}_{\mathcal{N}} : f(x) \neq \mathcal{O}(x)\}$ (Hendrycks et al., 2021). Finally, it follows from the above definitions that $\mathcal{A}_{\mathcal{N}} \subseteq \mathcal{A}_{\mathcal{C}} \subseteq \mathcal{A}_{\mathcal{U}}$.

Natural adversarial samples are a well-documented phenomenon in deep learning (Hendrycks et al., 2021). Literature suggests that they typically occur when deep learning models learn to rely on erroneous contextual cues to shortcut classification, as opposed to truly learning to distinguish

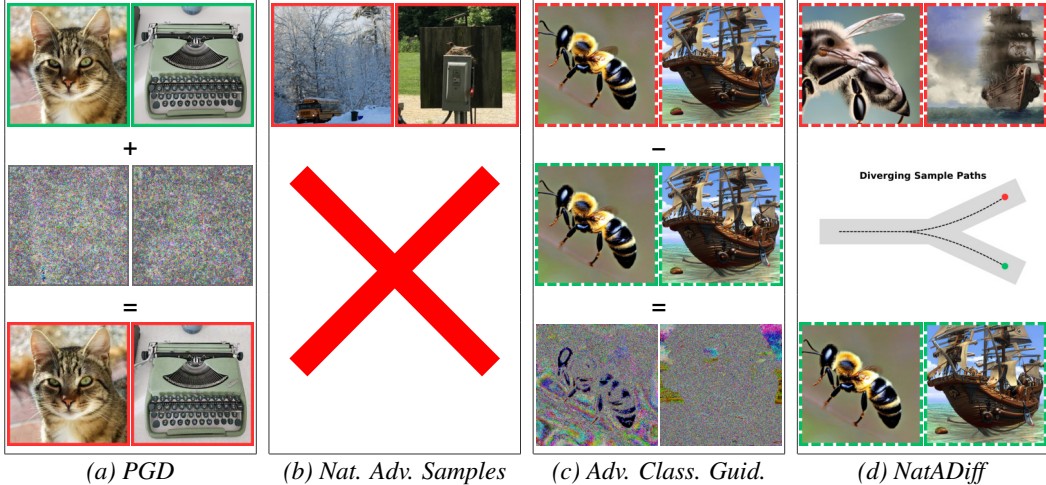

*(a) PGD*  *(b) Nat. Adv. Samples*  *(c) Adv. Class. Guid.*  *(d) NatADiff*

Figure 1: A comparison of different types of adversarial samples. Green and red borders indicate non-adversarial and adversarial samples, respectively. A dotted border denotes artificially generated images, while a solid border indicates real-world photographs. (a) Constrained adversarial attacks (PGD (Madry et al., 2018) used here) add perturbations to clean images. (b) Natural adversarial samples are test-time errors that do not contain perturbations. (c) Adversarial classifier guidance (Dai et al., 2024) produces constrained adversarial samples, as the difference between images generated with and without the guidance is minimal–their difference amounts to a constrained perturbation. (d) Adversarial samples generated with NatADiff diverge from those generated without NatADiff.

between classes (Hendrycks et al., 2021; Geirhos et al., 2020; Arjovsky et al., 2020) (see Appendix E for examples). These cues are typically features within an image that are highly correlated with a target class but not indicative of the class. For instance, a model that uses oceanic environments as a cue for predicting "shark" may misclassify an image of a shark lying on sand. By exploiting these easy-to-learn cues, models can reduce training loss without correctly learning to generalize to the underlying classification task. Additionally, it has been observed that natural adversarial samples are able to bypass common adversarial defences (Agarwal et al., 2022), and they exhibit high transferability i.e., the same image is misclassified by multiple classifiers (Hendrycks et al., 2021). The significant transferability of natural adversarial samples can be attributed to classifiers independently learning to rely on the same contextual cues, which is likely a consequence of shared correlations between cues and class labels across independent datasets (Hendrycks et al., 2021).

**Denoising diffusion generative models** (Ho et al., 2020) leverage a stochastic differential equation (SDE) to "learn" the space of natural images, allowing for the generation of natural, within-distribution images. The SDE is characterized by the forward process:

$$d\boldsymbol{x}_t = f(t)\boldsymbol{x}_t dt + g(t) \cdot d\boldsymbol{B}_t \quad \forall\, t \in [0, T]. \tag{1}$$

where $\boldsymbol{x}_t \in \mathbb{R}^m$, $f(t) : \mathbb{R} \to \mathbb{R}$ and $g(t) : \mathbb{R} \to \mathbb{R}$ are continuous functions of $t$, and $\cdot d\boldsymbol{B}_t$ denotes an Itô integral with respect to the standard multi-dimensional Brownian motion process $\boldsymbol{B}_t \in \mathbb{R}^m$ (Pavliotis, 2014b). Functions $f$ and $g$ are chosen such that the forward process progressively "destroys" structure in the image, $\boldsymbol{x}_0$, adding noise until it is approximately marginally Gaussian at termination time, T, i.e., $p(\boldsymbol{x}_T) \approx \mathcal{N}(0, \sigma_T^2 I)$. To generate a natural image, the forward process can be reversed, and structure recovered from Gaussian noise using either Anderson's reverse-time diffusion (Anderson, 1982) or the flow ODE (Song et al., 2021b) derived from (1). Both formulations require an estimate of the score function, $\nabla_{\boldsymbol{x}_t} \log(p(\boldsymbol{x}_t))$, which is approximated by a neural network, $\boldsymbol{\epsilon}_\theta(\boldsymbol{x}_t, t)$. This network is trained to predict the original image from a noisy version using the objective:

$$\min_\theta \mathbb{E}_{\boldsymbol{x}_0, \boldsymbol{x}_t, t \sim p(\boldsymbol{x}_0, \boldsymbol{x}_t, t)} \left[ \|\boldsymbol{x}_0 - (\boldsymbol{x}_t - \beta(t)\boldsymbol{\epsilon}_\theta(\boldsymbol{x}_t, t))/\alpha(t)\|_2^2 \right], \tag{2}$$

where $\alpha(t) = e^{\int_0^t f(u)du}$ and $\beta(t)^2 = \alpha(t)^2 \int_0^t \frac{g(u)^2}{\alpha(u)^2} du$. Given an optimal solution, $\boldsymbol{\epsilon}_{\theta^\star}(\boldsymbol{x}_t, t)$, to the above, the score function is given by (see Theorem P.3 in Appendix P):

$$\nabla_{\boldsymbol{x}_t} \log(p(\boldsymbol{x}_t)) = -\boldsymbol{\epsilon}_{\theta^\star}(\boldsymbol{x}_t, t)/\beta(t) \quad \forall\, t \in (0, T],\ \boldsymbol{x}_t \in \mathbb{R}^m. \tag{3}$$

Additionally, while $\boldsymbol{\epsilon}_{\theta^\star}(\boldsymbol{x}_t, t)$ can be used to directly estimate $\boldsymbol{x}_0$, it is typically of a lower quality than samples generated iteratively from the reverse-time diffusion or flow ODE (Ho et al., 2020; Song et al., 2021b; Nichol & Dhariwal, 2021).

**Denoising diffusion class guidance** provides finer control over the diffusion process by sampling from $\boldsymbol{x}_0 \sim p(\boldsymbol{x}_0|y)$ instead of $\boldsymbol{x}_0 \sim p(\boldsymbol{x}_0)$, where $y$ represents some conditioning information. To control the strength of class-guided diffusion, the marginal distribution is treated as $\bar{p}(\boldsymbol{x}_t|y) \propto p(y|\boldsymbol{x}_t)^\omega p(\boldsymbol{x}_t)/p(y)$, where $\omega \in \mathbb{R}_{>0}$ governs how strictly the diffusion adheres to the class constraint. For $\omega > 1$, the probability mass of $p(y|\boldsymbol{x}_t)^\omega$ is "tightened" around the regions of most probable $y$, while for $\omega < 1$, the probability mass is more diffuse. This results in stronger and weaker class adherence, respectively.

Class conditioning is incorporated directly into the diffusion score function in (3) by replacing $p(\boldsymbol{x}_t|y)$ with $\bar{p}(\boldsymbol{x}_t|y)$ as follows:

$$\nabla_{\boldsymbol{x}_t} \log(\bar{p}(\boldsymbol{x}_t|y)) = \omega \nabla_{\boldsymbol{x}_t} \log(p(y|\boldsymbol{x}_t)) - \frac{1}{\beta(t)} \boldsymbol{\epsilon}_{\theta^\star}(\boldsymbol{x}_t, t) \tag{4}$$

$$= -\frac{1}{\beta(t)} \big[ \omega \boldsymbol{\epsilon}_{\theta^\star}(\boldsymbol{x}_t, t, y) + (1 - \omega) \boldsymbol{\epsilon}_{\theta^\star}(\boldsymbol{x}_t, t) \big], \tag{5}$$

where $\boldsymbol{\epsilon}_{\theta^\star}(\boldsymbol{x}_t, t)$ and $\boldsymbol{\epsilon}_{\theta^\star}(\boldsymbol{x}_t, t, y)$ are neural networks trained to estimate the noise added in the forward diffusion process when $\boldsymbol{x}_0 \sim p(\boldsymbol{x}_0)$ and $\boldsymbol{x}_0 \sim p(\boldsymbol{x}_0|y)$, respectively. The distinction between (4) and (5) illustrates the difference between the two forms of class-guided diffusion: *classifier* and *classifier-free guidance*. In classifier guidance, a separate model, $p_\theta(y|\boldsymbol{x}_t)$, is trained to predict the probability of $y$ given $\boldsymbol{x}_t$ (Dhariwal & Nichol, 2021). In contrast, classifier-free guidance requires training a diffusion model to directly estimate $\nabla_{\boldsymbol{x}_t} \log(p(\boldsymbol{x}_t|y))$ (Ho & Salimans, 2022).

It is important to note that $\bar{p}(\boldsymbol{x}_t|y)$ does not represent the marginal distribution that arises from applying the diffusion in (1) to $\boldsymbol{x}_0 \sim p(\boldsymbol{x}_0|y)$ (Karras et al., 2024). Instead, it is a mechanism that forces the sampling trajectory of $\boldsymbol{x}_t$ into regions with a higher probability of $p(y|\boldsymbol{x}_t)$, and in doing so, deviates from reverse-time diffusion and flow ODE dynamics. However, despite foregoing theoretical guarantees of sampling convergence, class-guided diffusion often exhibits superior sampling quality (Dhariwal & Nichol, 2021; Ho & Salimans, 2022).

## 3 RELATED WORK

**Generating unconstrained adversarial samples.** Previous work has shown that modern generative models are capable of creating artificial unconstrained adversarial samples (Song et al., 2018; Zhao et al., 2018; Chen et al., 2023b; Dai et al., 2024). Initial approaches used GANs as the generative backbone for these attack; however, GANs are sensitive to perturbations to their sampling path, and they lack theoretical justification for such perturbations (Karras et al., 2019; Abdal et al., 2019). Recent methods have leveraged denoising diffusion models (Ho et al., 2020). Diffusion models possess superior generation quality to GANs, and provide theoretical justification for perturbing the sampling path (Dhariwal & Nichol, 2021). Dai et al. (2024) leveraged these properties to develop *AdvDiff*, which treats the true image class, $y$, and adversarial target, $\tilde{y}$, as random variables. The joint distribution can be decomposed as $p(\boldsymbol{x}_t, y, \tilde{y}) = p(y|\boldsymbol{x}_t)p(\tilde{y}|\boldsymbol{x}_t)p(\boldsymbol{x}_t)$, where it is assumed that $y$ and $\tilde{y}$ are conditionally independent given the noisy image, $\boldsymbol{x}_t$. Thus, given the forward diffusion in (1), the corresponding diffusion score function (see (4) and (5)) becomes

$$\nabla_{\boldsymbol{x}_t} \log(\bar{p}(\boldsymbol{x}_t|y, \tilde{y})) = -\frac{1}{\beta(t)} \big[ \omega \boldsymbol{\epsilon}_{\theta^\star}(\boldsymbol{x}_t, t, y) + (1 - \omega) \boldsymbol{\epsilon}_{\theta^\star}(\boldsymbol{x}_t, t) \big] + s \nabla_{\boldsymbol{x}_t} \log(p(\tilde{y}|\boldsymbol{x}_t)), \tag{6}$$

where $\omega$ and $s$ control the strength of the guidance, $\boldsymbol{\epsilon}_{\theta^\star}$ is a network trained to remove noise from $\boldsymbol{x}_t$, and the adversarial gradient, $\nabla_{\boldsymbol{x}_t} \log(p(\tilde{y}|\boldsymbol{x}_t))$, is derived from a victim classifier that provides class probabilities. Since AdvDiff directly uses the victim classifier gradient, it can be considered a form of classifier guidance, and is therefore susceptible to the same issues as classifier-guided diffusion.

Classifier-guided diffusion requires training a model, $p_\theta(y|\boldsymbol{x}_t)$, to predict the class of an image that has been corrupted with Gaussian noise (Dhariwal & Nichol, 2021), which is a form of adversarial training (He et al., 2019; Li et al., 2019). When a non-adversarially robust classifier is used instead, the diffusion process typically generates visually coherent samples that do not adhere to the desired class conditioning, but are erroneously classified as the desired class (Vaeth et al., 2024; Shen et al., 2024). We hypothesize that this phenomenon arises due to constrained adversarial samples frequently lying within an $\epsilon$-neighborhood of natural samples (Goodfellow et al., 2015; Madry et al., 2018). Under this hypothesis, the diffusion model acts as a constraint that pushes the sample towards the natural image manifold, while the non-adversarially robust classifier introduces a perturbation that directs the sample towards the nearest region containing samples of the desired class. The resulting trade-off between the diffusion model and classifier guidance incentivizes the diffusion trajectory to converge towards the adversarial regions that frequently lie imperceptibly close to the natural image manifold–that is, pockets of constrained adversarial samples (Shen et al., 2024).

## 4 METHODOLOGY

Natural adversarial samples frequently occur when classifiers over-rely on contextual cues to shortcut classification (Hendrycks et al., 2021). We incorporate this key observation into our proposed, diffusion-based natural adversarial sampling scheme–NatADiff (see Algorithm 1). NatADiff leverages adversarial boundary guidance to incorporate features from the adversarial class. In addition, we use augmented classifier guidance and time-travel sampling to enhance attack transferability while preserving image quality.

**Accounting for sample noise.** Classifier-guided diffusion specifically trains a classifier to predict the class label of a noisy sample $\boldsymbol{x}_t$. However, in *adversarial* diffusion guidance, the victim model is typically an "off-the-shelf" classifier that was never trained on noisy samples. Directly passing $\boldsymbol{x}_t$ to this classifier will likely degrade classification accuracy, leading to inferior diffusion guidance. To address this, we take the same approach as (Yu et al., 2023; Bansal et al., 2024; Shen et al., 2024) and use Tweedie's formula (Efron, 2011) to pass the classifier the current estimate of $\boldsymbol{x}_0$ at time $t$:

$$\hat{\boldsymbol{x}}_0(\boldsymbol{x}_t) = (\boldsymbol{x}_t - \beta(t)\boldsymbol{\epsilon}_{\theta^*}(\boldsymbol{x}_t, t, y))/\alpha(t). \tag{7}$$

**Reducing adversarial gradient.** Constrained adversarial attacks are sensitive to image transformations, with rotations, crops, and translations reducing the success rates of common attack algorithms (Guo et al., 2018). We leverage this by applying differentiable image transforms to reduce the effect of the adversarial gradient that points in the direction of constrained adversarial perturbations. We find that this increases the prevalence of visible adversarial features (see Appendix G.1 for ablation study). These transformations are similar to the ones used by Shen et al. (2024) to perform training-free classifier-guided diffusion. The local adversarial signal is "averaged out", reducing the likelihood of generating constrained adversarial samples, and forcing the manifestation of features from the–in our case–adversarial class conditioning (Shen et al., 2024).

Given a collection of differentiable image transforms: $\boldsymbol{\mathcal{T}} = \{\mathcal{T}_1, \mathcal{T}_2, \dots\}$, we compute the adversarial classifier gradient as

$$\nabla_{\boldsymbol{x}_t} \log(p(\tilde{y}|\boldsymbol{x}_t)) = \boldsymbol{g}(\boldsymbol{x}_t)/\|\boldsymbol{g}(\boldsymbol{x}_t)\|_2, \tag{8}$$

where $\boldsymbol{g}(\boldsymbol{x}_t) = \nabla_{\boldsymbol{x}_t} \log \left(\sigma_{\tilde{y}} \left(\frac{1}{|\boldsymbol{\mathcal{T}}|} \sum_{i=1}^{|\boldsymbol{\mathcal{T}}|} h(\mathcal{T}_i(\hat{\boldsymbol{x}}_0(\boldsymbol{x}_t)))\right)\right)$, $h : \mathbb{R}^m \to \mathbb{R}^{|\mathcal{Y}|}$ is a function that returns the victim classifier's logit predictions, and $\sigma_{\tilde{y}} : \mathbb{R}^{|\mathcal{Y}|} \to \mathbb{R}$ is a sigmoid function that returns the probability of the target adversarial class.

**Adversarial boundary guidance.** Initial experiments showed that substituting the improved adversarial gradient from (8) into (6) did not steer the diffusion trajectory towards natural adversarial samples (see Appendix G.2 for ablation study). This may occur because classifier augmentations eliminate many of the constrained adversarial samples that lie close to the image manifold, but not those further away. Consequently, if the initial sampling point of the diffusion trajectory is too distant from a region of natural adversarial samples, adversarial guidance will push the sample off the image manifold.

To address this, we leverage the connection between natural adversarial samples and the use of contextual cues as a classification shortcut (Hendrycks et al., 2021; Geirhos et al., 2020; Arjovsky et al.,

2020). We propose adversarial boundary guidance as a method of directing the diffusion trajectory towards samples that incorporate erroneous contextual cues, i.e., features from the adversarial class. We define adversarial boundary guidance as

$$\nabla_{\boldsymbol{x}_t} \log(\bar{p}(\boldsymbol{x}_t|y, \tilde{y})) = -\frac{1}{\beta(t)} \left[ \boldsymbol{\epsilon}_{\theta^\star}(\boldsymbol{x}_t, t) + (\omega - \mu\omega)\boldsymbol{v}_y + \mu\rho\boldsymbol{v}_{y \cap \tilde{y}} \right] + s\nabla_{\boldsymbol{x}_t} \log(p(\tilde{y}|\boldsymbol{x}_t)), \quad (9)$$

where $\omega, \rho, s \in \mathbb{R}_{\geq 0}$, $\mu \in [0, 1]$, $\boldsymbol{v}_y = \boldsymbol{\epsilon}_{\theta^\star}(\boldsymbol{x}_t, t, y) - \boldsymbol{\epsilon}_{\theta^\star}(\boldsymbol{x}_t, t)$, and $\boldsymbol{v}_{y \cap \tilde{y}} = \boldsymbol{\epsilon}_{\theta^\star}(\boldsymbol{x}_t, t, y \cap \tilde{y}) - \boldsymbol{\epsilon}_{\theta^\star}(\boldsymbol{x}_t, t)$. $\omega$ and $\rho$ govern the strength of classifier-free guidance, $s$ controls adversarial classifier guidance strength, and $\mu$ regulates how strongly the sample tends towards the intersection of the true and adversarial classes. For sufficiently large $\mu$, the sampling trajectory should approach the class intersection, incorporating enough elements from the adversarial class to cause a misclassification, while remaining within the bounds of the true class from a human's perspective. Note when $\mu = 0$, adversarial boundary guidance is equivalent to adversarial classifier guidance (Dai et al., 2024).

To justify (9), we note that the classifier-free score function can be rewritten as $\nabla_{\boldsymbol{x}_t} \log(\bar{p}(\boldsymbol{x}_t|y)) = -\frac{1}{\beta(t)} \left[ \boldsymbol{\epsilon}_{\theta^\star}(\boldsymbol{x}_t, t) + \omega\boldsymbol{v}_y \right]$ where $\boldsymbol{v}_y = \boldsymbol{\epsilon}_{\theta^\star}(\boldsymbol{x}_t, t, y) - \boldsymbol{\epsilon}_{\theta^\star}(\boldsymbol{x}_t, t)$ is a vector that points towards regions of the manifold containing images of class $y$. Additionally, recall that $\bar{p}(\boldsymbol{x}_t|y)$ is not the marginal density arising from a valid diffusion, rather it is a magnification of the guidance provided by a network that has "learned" the image manifold. To further exploit the information contained in this network, we introduce $\boldsymbol{v}_{y \cap \tilde{y}}$, which directs the sampling trajectory towards the class intersection.

**Time-travel sampling.** Significant disruption to the diffusion sampling path risks degradation in sample quality, or falling off the image manifold (Lugmayr et al., 2022; Yu et al., 2023). To mitigate these issues, we incorporate time-travel sampling into our diffusion scheme, which has been shown to increase image quality in cases where standard diffusion sampling would otherwise fail (Lugmayr et al., 2022; Yu et al., 2023; Shen et al., 2024).

By injecting additional sampling steps, time-travel sampling allows the diffusion model to explore a wider region of the sample space and recover from suboptimal trajectories. This helps maintain sample quality and prevents the generation process from diverging away from the image manifold. More concretely, given a sequence of sampling times $\{t_i\}_{i=1}^N$ with $t_{i+1} > t_i$ for all $i$, time-travel sampling resets the diffusion state at time $t_i$ by running the forward process, $\boldsymbol{x}_{t_{i+k}} \sim p(x_{t_{i+k}}|x_{t_i})$, and then resampling $\boldsymbol{x}_{t_i}$ using the reverse process (Anderson, 1982; Song et al., 2021b). This procedure is repeated $R$ times before $\boldsymbol{x}_{t_i}$ is accepted, after which sampling proceeds to $\boldsymbol{x}_{t_{i-1}}$. To improve efficiency, time-travel sampling can be applied to a subset of diffusion steps (Yu et al., 2023).

**Similarity targeting.** Many popular adversarial attacks operate in an untargeted setting (Szegedy et al., 2014; Goodfellow et al., 2015; Madry et al., 2018; Croce & Hein, 2020), where the only requirement is that the predicted class differs from the true class, i.e., $\tilde{y} \neq y$. These attacks often update the adversarial target dynamically during optimization, selecting the most probable incorrect class at each step, and they frequently outperform targeted variants (Croce et al., 2020). To extend NatADiff to untargeted settings, we propose *similarity targeting* (see Algorithm 2 in Appendix F).

Similarity targeting is based on the assumption that it is easier to incorporate adversarial features from classes that are semantically similar to the true class. To heuristically measure this similarity, we leverage the CLIP (Radford et al., 2021) text encoder, which maps class labels into a shared image-text embedding space. We then select the adversarial target as the class most similar to the true class in this embedding space, as measured by cosine similarity. Concretely, given the CLIP text encoder $C_{\text{enc}} : \mathcal{Y} \to \mathbb{R}^m$, the true class label, $y_i$, and the set of candidate adversarial labels $\mathcal{Y}_{\text{cand}} = \{y_1, \ldots, y_n\} \setminus y_i$, we define the adversarial target as

$$\tilde{y} = \arg\max_{y \in \mathcal{Y}_{\text{cand}}} \frac{C_{\text{enc}}(y_i) \cdot C_{\text{enc}}(y)}{\|C_{\text{enc}}(y_i)\|_2 \|C_{\text{enc}}(y)\|_2}. \quad (10)$$

## 5 EXPERIMENTS

### 5.1 EXPERIMENT DETAILS

We evaluate the effectiveness of NatADiff on the ImageNet (Deng et al., 2009) classification task, which requires a model to classify an image into one of 1,000 distinct object categories. We target a range of off-the-shelf ImageNet classifiers and assess the attack success rates and visual quality

of the generated samples. All experiments are conducted on an NVIDIA RTX 4090 GPU, and each sample takes approximately 103 seconds to generate (see Appendix N for runtime comparisons).

**Surrogate and victim models.** NatADiff and other comparable attack methods require access to classifier gradients when generating adversarial samples. The model whose gradients are used in this way is referred to as the *surrogate model*, and we test ResNet-50 (RN-50) (He et al., 2016), Inception-v3 (Inc-v3) (Szegedy et al., 2016), and Vision Transformer (ViT-H) (Dosovitskiy et al., 2021) surrogates. We examine the performance of these adversarial samples across RN-50, Inc-v3, ViT-H, adversarially trained ResNet (AdvRes) and Inception (AdvInc) (Kurakin et al., 2018), ResNet-152 (RN-152) (He et al., 2016), Max-ViT (Tu et al., 2022), Swin-B (Liu et al., 2021), and DeIT (Touvron et al., 2021) *victim* models.

**Diffusion model.** We use Stable Diffusion 1.5 (Rombach et al., 2022) (SD1.5) as our base diffusion model for NatADiff and adversarial classifier guidance (Dai et al., 2024). SD1.5 is a pretrained latent text-to-image diffusion model. The diffusion process is performed in a latent space, and a variational autoencoder (VAE), (Kingma & Welling, 2014), $V_{\text{dec}}$, is used to decode latent samples into the image space. To facilitate the use of adversarial classifier guidance the VAE must be incorporated into the gradient calculation. Specifically, given a sample, $z_t$, from the latent diffusion process, we introduce the VAE, $V_{\text{dec}}$, into (7) as $\hat{x}_0(z_t) = V_{\text{dec}}((z_t - \beta(t)\epsilon_{\theta^*}(z_t, t, y))/\alpha(t))$, and take the gradient with respect to $z_t$ instead of $x_t$ in (6) and (8). Finally, we use 200 sampling steps under the DDIM (Song et al., 2021a) parameterization, which defines the drift and diffusion coefficients in (1) as $f(t) = \frac{1}{2}\frac{d}{dt}\log(\hat{\alpha}_t)$ and $g(t)^2 = -\frac{d}{dt}\log(\hat{\alpha}_t)$, respectively (Han, 2024).

---

**Algorithm 1** NatADiff

**Require:** adversarial guidance parameters: $\omega, \rho, \mu, s$; true and adversarial classes: $y, \tilde{y}$; victim classifier: $h$; forward diffusion functions: $\alpha(t), \beta(t)$; stable diffusion model: $\epsilon_{\theta^*}$; VAE decoder: $V_{\text{dec}}$; collection of differentiable image transforms: $\{\mathcal{T}_1, \mathcal{T}_2, \dots\}$; sequence of sampling steps with $t_1 = 0$, $t_N = T$, and $t_{i+1} > t_i$: $\{t_i\}_{i=1}^N$; time-travel parameters: $R, r_l, r_u$; adversarial classifier bounds: $c_l, c_u$; number sampling attempts: $S$; guidance scalers: $\delta_\mu, \delta_s$

$z_T \sim \mathcal{N}(0, I)$
**for** $s = 1, \dots, S$ **do**
    **for** $i = N, \dots, 1$ **do**
        **if** $r_l \le t_i \le r_u$ **then**
            $\tilde{R} = R$
        **else**
            $\tilde{R} = 1$
        **end if**
        **for** $r = \tilde{R}, \dots, 1$ **do**      ▷ Time-travel loop
            $v_y = \epsilon_{\theta^*}(z_{t_i}, t_i, y) - \epsilon_{\theta^*}(z_{t_i}, t_i)$
            $v_{y \cap \tilde{y}} = \epsilon_{\theta^*}(z_{t_i}, t_i, y \cap \tilde{y}) - \epsilon_{\theta^*}(z_{t_i}, t_i)$
            $\hat{\epsilon} = \epsilon_{\theta^*}(x_{t_i}, t_i) + (\omega - \mu\omega)v_y + \mu\rho v_{y \cap \tilde{y}}$
            **if** $c_l \le t \le c_u$ **then**
                $\hat{x}_0 = V_{\text{dec}}\left(\frac{z_{t_i} - \beta(t_i)\hat{\epsilon}}{\alpha(t_i)}\right)$
                $g = \nabla_{z_{t_i}} \log\left(\sigma_{\tilde{y}}\left(\frac{1}{|\mathcal{T}|}\sum_{j=1}^{|\mathcal{T}|} h(\mathcal{T}_j(\hat{x}_0))\right)\right)$
                $g = \frac{g}{\|g\|_2}$
                $\hat{\epsilon} = \hat{\epsilon} - s\beta(t)g$
            **end if**
            $z_{t_{i-1}} \leftarrow$ reverse diffusion step using $\hat{\epsilon}$
            **if** $r > 1$ **then**   ▷ Sampling $z_{t_i} \sim p(z_{t_i}|z_{t_{i-1}})$
                $a = \frac{\alpha(t_i)}{\alpha(t_{i-1})}$
                $b^2 = \beta(t_i)^2 - (a\beta(t_{i-1}))^2$
                $z_{t_i} \sim \mathcal{N}\left(az_{t_{i-1}}, b^2 \cdot I\right)$
            **end if**
        **end for**
    **end for**
    **if** $\arg\max(h(V_{\text{dec}}(z_0))) \ne \tilde{y}$ **then**
        $\mu = \mu + \delta_\mu$
        $s = s + \delta_s$
    **else**
        **break**     ▷ End the search early if sample is found
    **end if**
**end for**
**return** $V_{\text{dec}}(z_0)$

---

**NatADiff settings.** We run NatADiff under both *targeted* and *untargeted* attack settings. For targeted attacks, we assign a random adversarial target to each sample. For untargeted attacks, we use similarity targeting from Section 4. During adversarial boundary guidance we use the text prompt *"<class name of $y$>"* as the conditioning for the true class guidance, $y$. For intersection guidance, $y \cap \tilde{y}$, we use the prompt *"<class name of $\tilde{y}$> and <class name of $y$>"*. We delay adversarial classifier guidance until timestep $t \le 700$, i.e., we set $s = 0$ for all $t > 700$. Finally, we choose a conservative value of $\mu = 0.2$ for all experiments, and select $s$ based on the target classifier (see Appendix G.2 for ablation study and additional experiment details). We generate 2,000 adversarial samples in each experiment run.

**Comparison methods.** We compare NatADiff to state-of-the-art constrained and unconstrained adversarial attacks: PGD Madry et al. (2018), AutoAttack (AA) Croce & Hein (2020), NCF (Yuan et al., 2022), DiffAttack (Chen et al., 2025), ACA (Chen et al., 2023b), and adversarial classifier guidance (AdvClass) (Dai et al., 2024). All methods use their default parameter settings and for comparison methods that alter a pre-existing "clean" image, we use their suggested ImageNet-compatible dataset as our clean baseline (Kurakin et al., 2017). We apply AdvClass under both *targeted* and *untargeted* attack settings (using the same similarity targeting as NatADiff). Finally, we acknowledge that although NCF, ACA, and AdvClass are classified as unconstrained attacks, only

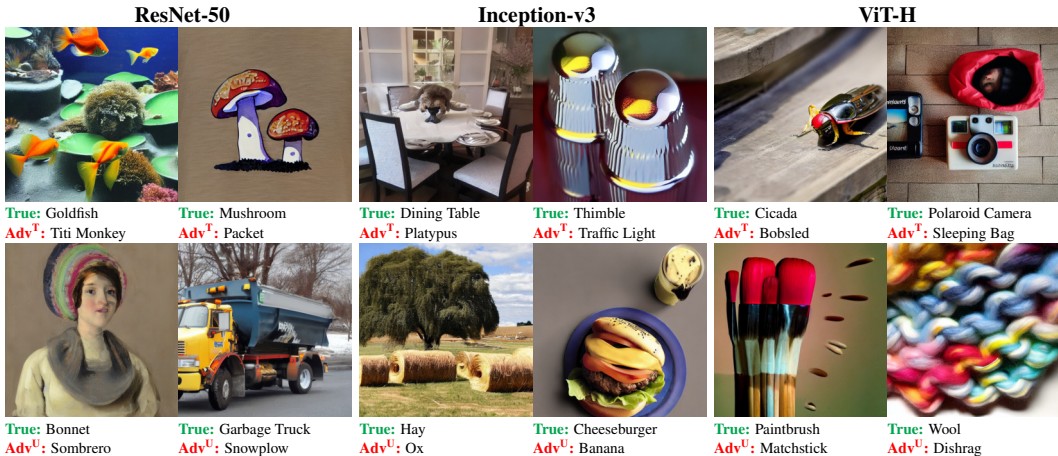

Figure 2: Adversarial samples generated by NatADiff with ResNet-50 (He et al., 2016), Inception-v3 (Szegedy et al., 2016), and ViT-H (Dosovitskiy et al., 2021) surrogate models (see column labels). We report the true class and adversarial target for each image. Superscripts T and U denote random and similarity targeted attacks, respectively.

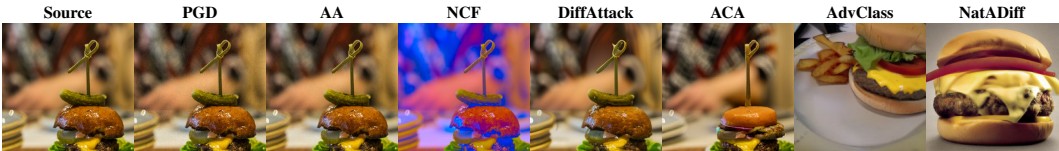

Figure 3: Source and adversarial samples generated by PGD Madry et al. (2018), AutoAttack (AA) Croce & Hein (2020), NCF (Yuan et al., 2022), DiffAttack (Chen et al., 2025), ACA (Chen et al., 2023b), AdvClass (Dai et al., 2024), and NatADiff with a ResNet-50 (He et al., 2016) surrogate model. Note true class is "burger" and adversarial target for AdvClass and NatADiff is "banana".

NatADiff and AdvClass have a fully unrestricted attack domain, as they are not bound to any initial clean image and are capable of synthesising entirely new images.

**Metrics.** To assess attack performance, we follow Chen et al. (2023b), and report attack success rate (ASR) as the percentage of misclassified samples. To evaluate image quality, we use the Inception Score (IS) (Salimans et al., 2016) and Fréchet Inception Distance (FID) (Fréchet, 1957). IS provides a direct measure of image quality, while FID estimates the similarity between the distributions of generated and real images. We compute FID with respect to both the ImageNet-Val (Deng et al., 2009) and ImageNet-A (Hendrycks et al., 2021) datasets to assess how closely NatADiff samples resemble natural images and known natural adversarial examples, respectively.

## 5.2 RESULTS

**Attack success.** NatADiff had comparable white-box ASR to current state-of-the-art attacks, but vastly superior transferability across all experiments (see Table 1). This suggests that NatADiff is able to more effectively leverage the mechanisms underpinning natural adversarial samples. Additionally, adversarial training did not offer any meaningful robustness to NatADiff, with both targeted and untargeted attacks transferring to adversarially trained ResNet and Inception models.

PGD (Madry et al., 2018) and AA (Croce & Hein, 2020) had the lowest transferability, likely because both are perturbation-based attacks, i.e., they rely on the small pockets of adversarial samples that neighbor natural images. These adversarial regions are not guaranteed to align across classifier architectures, especially architectures that are dissimilar, e.g., convolutional vs. transformer. Similarly, the lower transferability of NCF (Yuan et al., 2022) and DiffAttack (Chen et al., 2025) can be explained by their limited attack surface. NCF is restricted to attacking the color distribution of a "clean" source image (see Figure 3). In contrast, DiffAttack crafts adversarial perturbations for a

Table 1: **Attack success rate** (%) and **image quality** of adversarial samples. Superscripts T and U denote random and similarity targeted attacks, respectively. **Bold** and underlined values highlight the best and second best scores for each surrogate model. White-box ASR (same surrogate and victim model) is denoted with an $^*$. Note we do not report image quality for constrained perturbation-based attacks (these attacks make imperceptible image alterations).

| Surrogate Model | Attack | Victim Model ASR (%) | | | | | | | | | Average ASR | IS (↑) | FID-Val (↓) | FID-A (↓) |
| | | CNNs | | | | | Transformers | | | | | | | |
| | | RN-50 | Inc-v3 | RN-152 | AdvRes | AdvInc | ViT-H | Max-ViT | Swin-B | DeIT | | | | |
| | Clean | 5.3 | 7.6 | 2.9 | 3.0 | 5.8 | 10.9 | 3.8 | 4.5 | 7.4 | 5.7 | 55.0 | 58.0 | 94.7 |
| RN-50 | PGD | 99.4* | 11.8 | 5.2 | 4.9 | 8.1 | 10.5 | 4.4 | 5.5 | 8.2 | 17.6 | - | - | - |
| | AA | 100* | 13.3 | 10.0 | 3.9 | 8.8 | 10.5 | 5.4 | 5.6 | 8.0 | 18.4 | - | - | - |
| | NCF | 74.8* | 33.4 | 37.3 | 28.2 | 31.2 | 17.2 | 24.0 | 31.7 | 37.2 | 35.0 | 30.4 | 69.7 | 85.5 |
| | DiffAttack | 92.5* | 47.1 | 52.5 | 35.3 | 43.3 | 28.4 | 44.6 | 42.4 | 38.9 | 47.2 | 26.8 | 64.1 | 76.8 |
| | ACA | 78.8* | 53.3 | 52.7 | 49.8 | 53.1 | 41.8 | 46.4 | 49.3 | 50.6 | 52.9 | 23.9 | 65.0 | 77.9 |
| | AdvClass$^T$ | 99.6* | 35.0 | 32.1 | 31.4 | 33.5 | 25.8 | 30.0 | 30.8 | 32.8 | 39.0 | 38.3 | 48.9 | 92.4 |
| | AdvClass$^U$ | 99.9* | 42.5 | 44.3 | 38.7 | 41.1 | 29.7 | 37.6 | 38.4 | 39.1 | 45.7 | 38.5 | 50.2 | 92.7 |
| | NatADiff$^T$ | 96.9* | 60.1 | 56.5 | 55.3 | 58.9 | 36.8 | 45.3 | 49.0 | 52.3 | 56.8 | 26.0 | 66.5 | 77.3 |
| | NatADiff$^U$ | 99.3* | 68.3 | 72.1 | 65.3 | 66.8 | 45.3 | 64.1 | 65.2 | 67.0 | 68.2 | 43.2 | 51.4 | 95.9 |
| Inc-v3 | PGD | 6.0 | 99.7* | 4.0 | 5.1 | 10.4 | 10.2 | 4.1 | 5.6 | 7.4 | 16.9 | - | - | - |
| | AA | 7.3 | 100* | 4.9 | 4.8 | 12.8 | 10.6 | 5.7 | 6.1 | 8.0 | 17.8 | - | - | - |
| | NCF | 31.0 | 66.7* | 23.1 | 29.0 | 36.3 | 15.8 | 18.3 | 20.4 | 30.5 | 30.1 | 31.7 | 69.1 | 83.0 |
| | DiffAttack | 29.0 | 74.6* | 23.7 | 30.0 | 39.9 | 18.9 | 22.9 | 26.5 | 25.8 | 32.4 | 33.2 | 63.7 | 78.2 |
| | ACA | 50.9 | 67.8* | 48.2 | 54.2 | 60.1 | 43.6 | 45.1 | 48.8 | 51.3 | 52.2 | 23.1 | 68.0 | 78.8 |
| | AdvClass$^T$ | 35.1 | 99.6* | 34.5 | 35.6 | 39.5 | 28.8 | 32.4 | 34.0 | 35.7 | 41.7 | 33.7 | 51.0 | 89.2 |
| | AdvClass$^U$ | 38.0 | 99.9* | 38.7 | 40.4 | 44.2 | 30.0 | 36.0 | 36.6 | 38.9 | 44.8 | 39.7 | 49.4 | 93.3 |
| | NatADiff$^T$ | 53.4 | 97.9* | 49.4 | 57.3 | 62.6 | 35.4 | 44.4 | 45.1 | 50.8 | 55.2 | 27.7 | 66.6 | 78.2 |
| | NatADiff$^U$ | 67.4 | 99.4* | 65.7 | 70.1 | 75.7 | 44.4 | 60.3 | 60.2 | 63.1 | 67.4 | 47.0 | 50.5 | 98.9 |
| ViT-H | PGD | 5.8 | 11.0 | 3.6 | 4.0 | 7.8 | 96.2* | 4.5 | 5.4 | 9.2 | 16.4 | - | - | - |
| | AA | 6.5 | 9.8 | 3.9 | 4.3 | 8.6 | 100* | 4.5 | 5.9 | 9.9 | 17.0 | - | - | - |
| | NCF | 20.0 | 19.4 | 14.8 | 15.4 | 18.5 | 50.6* | 11.9 | 15.6 | 21.2 | 20.8 | 39.8 | 63.1 | 86.4 |
| | DiffAttack | 20.5 | 25.0 | 17.2 | 18.9 | 22.4 | 73.2* | 18.1 | 22.3 | 20.6 | 26.5 | 35.2 | 63.4 | 80.0 |
| | ACA | 50.5 | 54.5 | 48.1 | 49.1 | 52.8 | 75.8* | 47.5 | 49.7 | 50.5 | 53.2 | 25.5 | 64.2 | 80.9 |
| | AdvClass$^T$ | 33.9 | 35.9 | 33.4 | 34.4 | 34.4 | 92.6* | 31.9 | 33.4 | 36.0 | 40.7 | 38.9 | 48.5 | 95.2 |
| | AdvClass$^U$ | 35.2 | 37.5 | 35.8 | 35.2 | 36.0 | 98.7* | 33.9 | 34.9 | 37.7 | 42.8 | 39.2 | 48.5 | 98.8 |
| | NatADiff$^T$ | 70.7 | 73.5 | 68.4 | 71.3 | 72.1 | 98.5* | 65.7 | 66.9 | 71.7 | 73.2 | 15.3 | 88.0 | 93.5 |
| | NatADiff$^U$ | 66.8 | 67.0 | 65.3 | 64.9 | 65.8 | 99.6* | 63.9 | 65.4 | 68.6 | 69.7 | 31.9 | 53.9 | 96.2 |

source image by perturbing the diffusion latent space subject to the constraint that the reconstructed adversarial image must remain sufficiently close to the original (see Figure 3).

Adversarial classifier guidance (Dai et al., 2024) is outperformed by ACA (Chen et al., 2023b) and NatADiff in all experiments. This can be attributed to the limited guidance provided by injecting non-robust classifier gradients into the diffusion sampling trajectory (Shen et al., 2024). ACA is the most comparable to NatADiff performance-wise; however, ACA alters the semantic structure of a source image and is thus constrained by the semantics of the initial image. In contrast, NatADiff has a wider attack surface as it is free to generate any image that fools a surrogate classifier. Furthermore, NatADiff uses a diffusion model to incorporate adversarial features that are classifier-agnostic (as seen in Figure 2), and as such, is the only method that does not solely rely on the gradient of a surrogate classifier.

ViT-H (Dosovitskiy et al., 2021) is the current state-of-the-art in image classification and is the most resistant to transfer attacks. This is unsurprising, as it uses the modern transformer architecture and is the largest model examined. ViT-H learns a more robust feature representation than convolutional and smaller transformer models, which makes it less susceptible to both constrained and natural adversarial samples. However, despite the strengths of the ViT-H architecture, NatADiff is able to reliably generate samples that transfer to ViT-H—albeit at a lower ASR than equivalent attacks against all other models.

When comparing NatADiff's targeted attacks with their untargeted counterparts, we see that untargeted attacks outperform targeted attacks both in terms of victim classifier performance and transferability for all classifier *except* ViT-H. We believe that the discrepancy between targeted and untargeted ViT-H samples can be attributed to the strength of the decision boundary learnt by the ViT-H model. The diffusion model struggles to generate feasible *targeted* adversarial samples for ViT-H, leading to the introduction of image artifacts (as seen in Figure 18 from Appendix K.5, and supported by the IS and FID-Val scores in Table 1). These artifacts substantially degrade image quality and artificially inflate the attack success rate of the targeted ViT-H samples. Importantly, this degradation in image quality was limited to targeted ViT-H attacks. This indicates that some adversarial targets are easier to achieve than others, which further motivates the use of similarity targeting as a method for identifying classifier "weak spots."

**Image quality.** We observe a clear disparity in the image quality of targeted and untargeted NatADiff variants (see Table 1). Targeted NatADiff samples exhibit lower FID-A but worse IS and FID-VAL, indicating that they are closer in distribution to known natural adversarial examples, albeit with lower image quality and less alignment to the ImageNet validation dataset. In contrast, untargeted NatADiff achieves IS and FID-VAL comparable to other generative methods, but with a higher FID-A, suggesting that overall image quality improves at the expense of alignment with natural adversarial samples. This follows from the known characteristics of natural adversarial samples, which often blend features from disparate classes (Hendrycks et al., 2021; Geirhos et al., 2020; Arjovsky et al., 2020). Replicating such blending places greater demands on the underlying diffusion model to locate plausible points on the image manifold, which can introduce artifacts and degrade image quality. In contrast, similarity targeting blends more related classes, yielding samples with higher visual fidelity but less alignment with natural adversarial distributions. Additionally, NCF (Yuan et al., 2022), DiffAttack (Chen et al., 2025), and ACA (Chen et al., 2023b) all achieve superior FID-A than AdvClass (Dai et al., 2024) and untargeted NatADiff. This can be attributed to the low FID-A of the clean baseline dataset that NCF, DiffAttack, and ACA use as their source, which causes them to inherit the same distributional properties. Conversely, AdvClass and NatADiff generate artificial samples and are thus constrained both by the distributional tendencies of the underlying diffusion model and the effect of similarity targeting.

## 6 CONCLUSION

We introduce NatADiff, an adversarial sampling scheme that leverages diffusion models to generate highly transferable adversarial samples. Our method is motivated by the observation that natural adversarial samples frequently contain features from the adversarial class, which deep learning models exploit to shortcut the classification processes. To leverage this behavior, we guide the diffusion trajectory towards the intersection of the true and adversarial classes. Our method achieves comparable white-box attack success rates to current state-of-the-art techniques, while exhibiting significantly higher transferability across models. Furthermore, samples generated using NatADiff align more closely with known natural adversarial samples than those generated via adversarial classifier guidance alone. These results demonstrate that NatADiff produces adversarial samples that transfer more effectively than existing attacks, and more faithfully resemble naturally occurring test-time errors than those generated from vanilla adversarial diffusion guidance.

### ACKNOWLEDGMENTS

Max Collins is the recipient of the Research Training Program scholarship funded by the Australian Government. Professor Ajmal Mian is the recipient of an Australian Research Council Future Fellowship Award (project number FT210100268) funded by the Australian Government. This research was partially supported by National Intelligence and Security Discovery Research Grants (project number NS220100007), funded by the Department of Defence, Australia.

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

# Appendix

## Table of Contents

## A    LIMITATIONS

While NatADiff is effective at generating highly transferable adversarial samples, it remains computationally expensive due to the iterative nature of diffusion and the overhead introduced by adversarial guidance. This is an inherent limitation of diffusion-based generative methods, and one unlikely to change without significant advances in generative modeling or architectural design. Additionally, the use of similarity targeting on datasets like ImageNet can lead to *subtle* misclassifications–e.g., between similar dog breeds–which may diminish the perceived severity of the attack. A potential refinement would be to surface a ranked list of similar classes, allowing users to select more divergent adversarial targets while retaining the semantic grounding of similarity-based selection. We also note that we used a conservative setting for the adversarial boundary guidance term, $\mu$, as larger values caused generated samples to occasionally include the adversarial class, as discussed in Appendix G.2. Finally, we restrict our evaluation to ImageNet classifiers, as ImageNet offers a diverse label space, which supports varied attack scenarios. Extending NatADiff to more specialized domains remains an avenue for future work.

## B    ETHICS AND BROADER IMPACTS

We adhere to the ICLR code of ethics. We acknowledge that this work explores the use of generative models as a means of creating highly transferable adversarial samples. While adversarial attacks raise legitimate concerns regarding misuse, our objective is to expose fundamental vulnerabilities of current classifiers and to better understand the structure of natural adversarial samples. By making our models and code publicly available, we aim to support transparency and reproducibility, and we believe that insight into generative adversarial mechanisms is a necessary step toward building more secure and interpretable classifiers. We do not use private or sensitive data, and all data and models used are publicly released and broadly studied. In future work, we plan to explore how NatADiff can be extended to detect or defend against naturally occurring adversarial samples.

## C    REPRODUCIBILITY

We ensure reproducibility by providing detailed descriptions of our algorithms (see Algorithms 1 and 2) and experiment parameter settings (see Tables 7 and 8). Our full codebase is included in the supplementary material, along with all configuration files required to replicate our experiments. Comparison methods are implemented using publicly available repositories, and we follow the authors' recommended hyperparameters unless otherwise stated.

## D    PERTURBATION-BASED ATTACKS AND DEFENCES

In this section, we provide a brief overview of existing constrained perturbation-based attack and defense strategies for image classification models. We focus on the optimization-based formulation of adversarial attacks and highlight the theoretical underpinnings of common training-time defenses.

### D.1    ADVERSARIAL ATTACKS

Szegedy et al. (2014) were the first to demonstrate that imperceptible perturbations to an image's pixel values could cause deep learning models to misclassify the image with a high probability (see Figure 1). Mathematically, these constrained adversarial attacks can be considered a solution to the following constrained optimization problem:

$$\min_{\boldsymbol{\delta} \in \mathcal{S}} \mathcal{L}_h(\boldsymbol{x} + \boldsymbol{\delta}; \boldsymbol{\theta}, \tilde{y}), \qquad (11)$$

where $\boldsymbol{\delta} \in \mathbb{R}^m$ is the computed perturbation, $\boldsymbol{x} \in \mathbb{R}^m$ is the vectorized "clean" image, $h : \mathbb{R}^m \to \mathcal{Y}$ is a "trained" classifier model with parameters $\boldsymbol{\theta}$, $\tilde{y} \in \mathcal{Y}$ is the class targeted by the adversarial attack, $\mathcal{L}_h(\cdot; \boldsymbol{\theta}, \tilde{y})$ is the loss of the classifier with respect to the target adversarial class, and $\mathcal{S} \triangleq \{\boldsymbol{\delta} \in \mathbb{R}^m : \|\boldsymbol{\delta}\|_p < L\}$ is a convex set of allowable perturbation sizes. Algorithms such as fast gradient sign method (FGSM) (Goodfellow et al., 2015), projected gradient descent (PGD) (Madry et al., 2018),

and AutoAttack (Croce & Hein, 2020), have been proposed to efficiently solve the optimization problem in (11).

Other attack methods have relaxed the constraint on the magnitude of the adversarial perturbation. These unconstrained adversarial attacks seek to alter the semantic information within an image, resulting in misclassification without visually altering the perceptible class. Additionally, techniques like selective cropping and rotation, texture remapping, color pallette transformations, and generative sampling have all been used to successfully "fool" modern deep learning models (Brown et al., 2018; Bhattad et al., 2020; Song et al., 2018; Wang et al., 2020; Dai et al., 2024; Xie et al., 2025).

### D.2 DEFENCES AGAINST ADVERSARIAL ATTACKS

Several defensive measures have been proposed that aim to purify adversarial inputs (Samangouei et al., 2018; Nie et al., 2022), harden model architectures against attacks (Gu & Rigazio, 2015; Xu et al., 2018), or improve training procedures (Szegedy et al., 2014; Madry et al., 2018). A key challenge in designing adversarial defences is preventing attackers from crafting new attacks that exploit the adapted model. For this reason *adversarial training* has become one of the most popular defences, as it both addresses the source of the adversarial attack, while providing theoretical guarantees of robustness against all possible perturbation-based adversaries.

Adversarial training can be formulated as the following saddle-point optimization problem:

$$\min_{\boldsymbol{\theta}} \ \mathbb{E}_{(\boldsymbol{x},y)\sim\mathcal{D}} \left[ \max_{\boldsymbol{\delta}\in\mathcal{S}} \ \mathcal{L}_h(\boldsymbol{x} + \boldsymbol{\delta}; \boldsymbol{\theta}, y) \right], \tag{12}$$

where $\mathcal{D}$ is the joint distribution of naturally occurring images and classes, and $y \in \mathcal{Y}$ is the true class label of $\boldsymbol{x}$ (Szegedy et al., 2014; Madry et al., 2018). The optimization problem in (12) can be thought of as minimizing the loss caused by the strongest possible adversarial attack. Thus, any model that minimizes (12) is theoretically guaranteed to be resistant to its strongest possible adversarial perturbations.

## E EXAMPLE NATURAL ADVERSARIAL SAMPLES

Figure 4 shows natural adversarial samples from the ImageNet-A dataset (Hendrycks et al., 2021), each paired with a heatmap of the classifier-guidance gradient with respect to the adversarial class, $\nabla_{\boldsymbol{x}} \log(p(\tilde{y} \mid \boldsymbol{x}))$. These gradients highlight the image features that contribute to misclassification and that would be emphasized during adversarial classifier-guided diffusion (Dai et al., 2024). In the first image, the classifier gradient is concentrated around the school bus and the snowbanks running alongside the road; in the second, it is concentrated on the snail and its shadow; and in the third, it is concentrated on the power switch. This suggests the classifier has "learned" to associate vehicles beside snowbanks with snowplows, dark elliptical objects with cockroaches, and vertical rectangular boxes with pay phones.

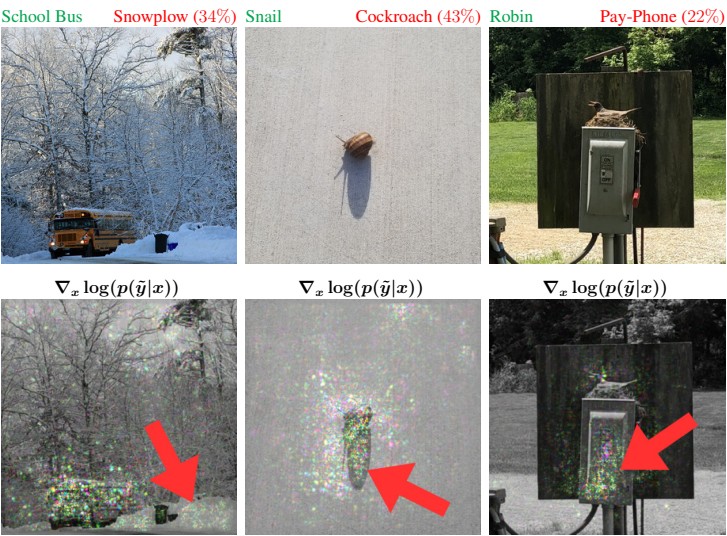

Figure 4: **Top:** Natural adversarial samples compiled by Hendrycks et al. (2021) for ImageNet (Deng et al., 2009) classifiers. The green labels denote the ground-truth classes; the red labels are the classes assigned by a ResNet-50 classifier (He et al., 2016). **Bottom:** Heatmap of the ResNet-50 adversarial classifier-guidance (Dai et al., 2024) gradient with respect to the adversarial classes. Arrows point to features from the adversarial class that affect the ResNet-50 classification.

## F    NatADiff similarity attack algorithm

Algorithm 2 provides the algorithm for the similarity targeted variant of NatADiff.

---

**Algorithm 2** NatADiff–Similarity

---

**Require:** adversarial guidance parameters: $\omega$, $\rho$, $\mu$, $s$; true class: $y$; candidate adversarial classes: $\mathcal{Y}_{\text{cand}} = \{\tilde{y}_1, \tilde{y}_2, \dots\}$; victim classifier: $h$; forward diffusion functions: $\alpha(t)$, $\beta(t)$; stable diffusion model: $\boldsymbol{\epsilon}_{\theta\star}$; VAE decoder: $V_{\text{dec}}$; CLIP text encoder: $C_{\text{enc}}$; collection of differentiable image transforms: $\{\mathcal{T}_1, \mathcal{T}_2, \dots\}$; sequence of sampling steps with $t_1 = 0$, $t_N = T$, and $t_{i+1} > t_i$: $\{t_i\}_{i=1}^N$; time-travel parameters: $R$, $r_l$, $r_u$; adversarial classifier bounds: $c_l$, $c_u$; number sampling attempts: $S$; guidance scalers: $\delta_\mu$, $\delta_s$

$\tilde{y} = \arg\min_{\gamma \in \mathcal{Y}_{\text{cand}}} \frac{C_{\text{enc}}(y) \cdot C_{\text{enc}}(\gamma)}{\|C_{\text{enc}}(y)\|_2 \|C_{\text{enc}}(\gamma)\|_2}$

$\boldsymbol{z}_T \sim \mathcal{N}(0, I)$

**for** $s = 1, \dots, S$ **do**

    **for** $i = N, \dots, 1$ **do**

        **if** $r_l \leq t_i \leq r_u$ **then**

            $\tilde{R} = R$

        **else**

            $\tilde{R} = 1$

        **end if**

        **for** $r = \tilde{R}, \dots, 1$ **do**        ▷ Time-travel loop

            $\boldsymbol{v}_y = \boldsymbol{\epsilon}_{\theta\star}(\boldsymbol{z}_{t_i}, t_i, y) - \boldsymbol{\epsilon}_{\theta\star}(\boldsymbol{z}_{t_i}, t_i)$

            $\boldsymbol{v}_{y \cap \tilde{y}} = \boldsymbol{\epsilon}_{\theta\star}(\boldsymbol{z}_{t_i}, t_i, y \cap \tilde{y}) - \boldsymbol{\epsilon}_{\theta\star}(\boldsymbol{z}_{t_i}, t_i)$

            $\hat{\boldsymbol{\epsilon}} = \boldsymbol{\epsilon}_{\theta\star}(\boldsymbol{x}_{t_i}, t_i) + (\omega - \mu\omega)\boldsymbol{v}_y + \mu\rho\boldsymbol{v}_{y \cap \tilde{y}}$

            **if** $c_l \leq t \leq c_u$ **then**

                $\hat{\boldsymbol{x}}_0 = V_{\text{dec}}\left(\frac{\boldsymbol{z}_{t_i} - \beta(t_i)\hat{\boldsymbol{\epsilon}}}{\alpha(t_i)}\right)$

                $\boldsymbol{g} = \nabla_{\boldsymbol{z}_{t_i}} \log\left(\sigma_{\tilde{y}}\left(\frac{1}{|\mathcal{T}|}\sum_{j=1}^{|\mathcal{T}|} h(\mathcal{T}_j(\hat{\boldsymbol{x}}_0))\right)\right)$

                $\boldsymbol{g} = \frac{\boldsymbol{g}}{\|\boldsymbol{g}\|_2}$

                $\hat{\boldsymbol{\epsilon}} = \hat{\boldsymbol{\epsilon}} - s\beta(t)\boldsymbol{g}$

            **end if**

            $\boldsymbol{z}_{t_{i-1}} \leftarrow$ reverse diffusion step using $\hat{\boldsymbol{\epsilon}}$

            **if** $r > 1$ **then**        ▷ Sampling $\boldsymbol{z}_{t_i} \sim p(\boldsymbol{z}_{t_i}|\boldsymbol{z}_{t_{i-1}})$

                $a = \frac{\alpha(t_i)}{\alpha(t_{i-1})}$

                $b^2 = \beta(t_i)^2 - (a\beta(t_{i-1}))^2$

                $\boldsymbol{z}_{t_i} \sim \mathcal{N}\left(a\boldsymbol{z}_{t_{i-1}}, b^2 \cdot I\right)$

            **end if**

        **end for**

    **end for**

    **if** $\arg\max(h(V_{\text{dec}}(\boldsymbol{z}_0))) \neq \tilde{y}$ **then**

        $\mu = \mu + \delta_\mu$

        $s = s + \delta_s$

    **else**

        **break**        ▷ End the search early if sample is found

    **end if**

**end for**

**return** $V_{\text{dec}}(\boldsymbol{z}_0)$

---

## G    NatADiff ablation studies

Here we provide ablation studies examining the effect of classifier augmentations, adversarial boundary guidance, and adversarial classifier guidance. We also include a visualisation of how these components influence the generated image (see Figure 5). Recall from the main paper that augmented adversarial classifier guidance introduces visual features from the adversarial class (see Appendix G.1), adversarial boundary guidance further increases the amount of adversarial structure added and improves image quality (see Appendix G.2), and time-travel sampling provides an additional improvement in image quality (Lugmayr et al., 2022) (see Figure 5).

### G.1    Effect of classifier augmentations

In the main paper, we argue that classifier augmentations reduce the local adversarial signal of the surrogate classifier gradient and thereby encourage the diffusion model to introduce semantically meaningful adversarial features into the generated image. To verify this claim, we compare NatADiff samples produced with and without classifier augmentations. Using a ResNet-50 (He et al., 2016) surrogate classifier, we report Attack Success Rate (ASR), Inception Score (IS) (Salimans et al., 2016), and Fréchet Inception Distance (FID) (Fréchet, 1957). For FID, we evaluate with respect to

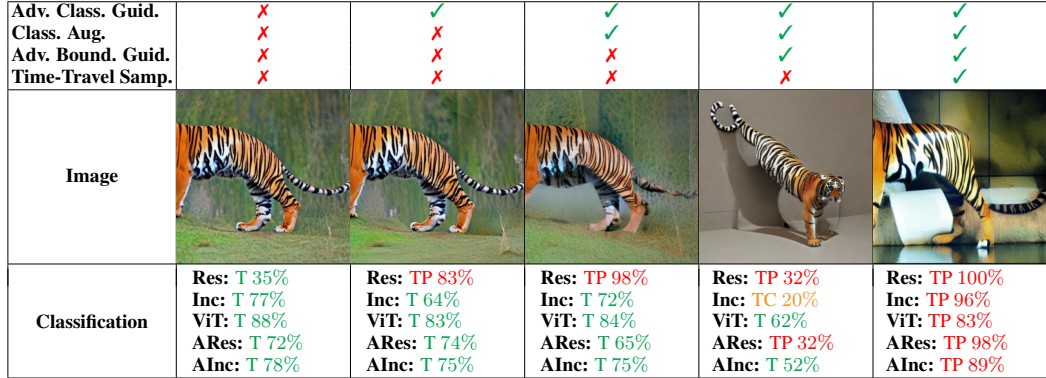

| Adv. Class. Guid. | ✗ | ✓ | ✓ | ✓ | ✓ |
| Class. Aug. | ✗ | ✗ | ✓ | ✓ | ✓ |
| Adv. Bound. Guid. | ✗ | ✗ | ✗ | ✓ | ✓ |
| Time-Travel Samp. | ✗ | ✗ | ✗ | ✗ | ✓ |

| Image | | | | | |

| Classification | **Res:** T 35%
**Inc:** T 77%
**ViT:** T 88%
**ARes:** T 72%
**AInc:** T 78% | **Res:** TP 83%
**Inc:** T 64%
**ViT:** T 83%
**ARes:** T 74%
**AInc:** T 75% | **Res:** TP 98%
**Inc:** T 72%
**ViT:** T 84%
**ARes:** T 65%
**AInc:** T 75% | **Res:** TP 32%
**Inc:** TC 20%
**ViT:** T 62%
**ARes:** TP 32%
**AInc:** T 52% | **Res:** TP 100%
**Inc:** TP 96%
**ViT:** TP 83%
**ARes:** TP 98%
**AInc:** TP 89% |

Figure 5: Effect of adversarial classifier guidance, classifier augmentations, adversarial boundary guidance, and time-travel sampling on samples generated by NatADiff. Prompt = "tiger", adversarial target = "toilet paper", surrogate model = ResNet-50 (He et al., 2016). Classification scores are given for ResNet-50, Inception-v3 (Szegedy et al., 2016), ViT-H (Dosovitskiy et al., 2021), and adversarially trained ResNet-50 and Inception victim models (Kurakin et al., 2018). Note: "T": "Tiger", "TP": "Toilet Paper", "TC": "Tiger Cat".

**NatADiff without Classifier Augmentations**

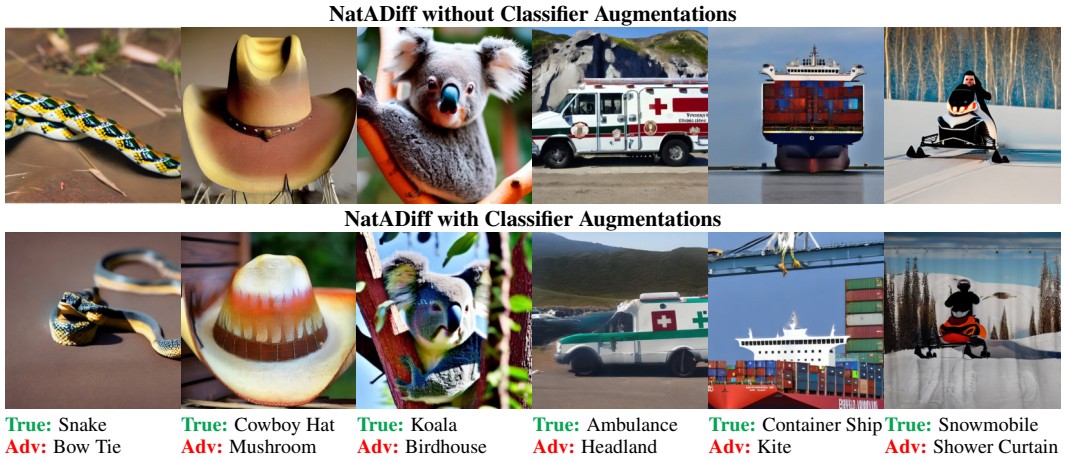

**NatADiff with Classifier Augmentations**

**True:** Snake **Adv:** Bow Tie    **True:** Cowboy Hat **Adv:** Mushroom    **True:** Koala **Adv:** Birdhouse    **True:** Ambulance **Adv:** Headland    **True:** Container Ship **Adv:** Kite    **True:** Snowmobile **Adv:** Shower Curtain

Figure 6: Comparison of samples generated by NatADiff under targeted attack settings with and without classifier augmentations. We use a ResNet-50 (He et al., 2016) surrogate model. We report the true class and adversarial target for each image.

both the ImageNet-Val (Deng et al., 2009) and ImageNet-A (Hendrycks et al., 2021) datasets to assess how closely NatADiff samples resemble natural images and known natural adversarial examples, respectively. The exact parameter settings for each experiment are provided in Table 7.

From Table 2 we observe that classifier augmentations significantly increase the transferability of adversarial samples, while retaining comparable white-box ASR (see Table 2). Images generated with

Table 2: **Attack success rate** (%) and **image quality** of adversarial samples generated by NatADiff under targeted attack settings with and without classifier augmentations. We use a ResNet-50 (He et al., 2016) surrogate model. **Bold** values highlight the best score. White-box ASR (same surrogate and victim model) is denoted with an *.

| Attack | Victim Model ASR (%) | | | | | | | | | Average ASR | IS (↑) | FID-Val (↓) | FID-A (↓) |
| | CNNs | | | | | Transformers | | | | | | | |
| | RN-50 | Inc-v3 | RN-152 | AdvRes | AdvInc | ViT-H | Max-ViT | Swin-B | DeIT | | | | |
| NatADiff | 96.9* | **60.1** | **56.5** | **55.3** | **58.9** | **36.8** | **45.3** | **49.0** | **52.3** | **56.8** | 26.0 | 66.5 | **77.3** |
| NatADiff (No-Aug) | **98.7*** | 48.5 | 45.6 | 44.1 | 46.8 | 31.7 | 38.6 | 40.6 | 43.3 | 48.7 | **30.5** | **56.7** | 81.7 |

classifier augmentations have slightly reduced overall image quality (IS and FID-VAL), but improved FID-A. This suggests that classifier augmentations introduce slightly more generative artifacts in an image, but also incorporate more meaningful adversarial features, which produces images that align more closely with known natural adversarial samples (see Figures 5 and 6)

### G.2 EFFECT OF BOUNDARY GUIDANCE STRENGTH

The adversarial boundary guidance term, $\mu$, governs how strongly features from the adversarial class are incorporated into the generated sample. To evaluate the effect of this parameter, we conduct an ablation study across $\mu \in \{0.0, 0.1, 0.2, 0.3, 0.4, 0.5\}$ using a ResNet-50 (He et al., 2016) surrogate model and applying NatADiff in targeted mode, i.e., with adversarial classes selected at random. We report attack success rate, Inception Score (IS) (Salimans et al., 2016) and Fréchet Inception Distance (FID) (Fréchet, 1957). For FID, we evaluate with respect to both the ImageNet-Val (Deng et al., 2009) and ImageNet-A (Hendrycks et al., 2021) datasets to assess how closely NatADiff samples resemble natural images and known natural adversarial examples, respectively. The exact parameter settings for each experiment are provided in Table 7.

Table 3: **Attack success rate** (ASR) and image quality of adversarial samples generated by NatADiff under targeted attack settings with varying adversarial boundary guidance strength, $\mu$. We use a ResNet-50 (He et al., 2016) surrogate model. **Bold** values highlight the best score. White-box ASR (same surrogate and victim model) is denoted with an $^*$.

| Attack | Victim Model ASR (%) | | | | | | | | | Average ASR | IS ($\uparrow$) | FID-Val ($\downarrow$) | FID-A ($\downarrow$) |
| | CNNs | | | | | Transformers | | | | | | | |
| | RN-50 | Inc-v3 | RN-152 | AdvRes | AdvInc | ViT-H | Max-ViT | Swin-B | DeIT | | | | |
|---|---|---|---|---|---|---|---|---|---|---|---|---|---|
| NatADiff ($\mu = 0.0$) | 95.2* | 54.2 | 49.6 | 48.7 | 53.7 | 32.6 | 42.4 | 43.7 | 48.4 | 52.1 | 26.1 | 67.7 | 78.9 |
| NatADiff ($\mu = 0.1$) | 95.4* | 55.2 | 52.5 | 51.5 | 53.9 | 33.8 | 44.2 | 45.0 | 49.3 | 53.4 | 26.6 | 63.6 | 78.1 |
| NatADiff ($\mu = 0.2$) | 96.9* | 60.1 | 56.5 | 55.3 | 58.9 | 36.8 | 45.3 | 49.0 | 52.3 | 56.8 | 26.0 | 66.5 | **77.3** |
| NatADiff ($\mu = 0.3$) | 97.4* | 62.4 | 60.0 | 57.8 | 61.2 | 42.6 | 50.7 | 53.4 | 55.0 | 60.1 | 27.6 | 63.8 | 77.8 |
| NatADiff ($\mu = 0.4$) | **98.5*** | 67.8 | 65.2 | 62.5 | 65.5 | 49.3 | 57.1 | 59.0 | 60.7 | 65.1 | 28.9 | 63.4 | 80.2 |
| NatADiff ($\mu = 0.5$) | **98.5*** | **71.6** | **70.1** | **68.0** | **70.6** | **53.7** | **62.7** | **63.8** | **66.4** | **69.5** | **32.0** | **61.7** | 80.1 |

We observe that attack success rate, IS, and FID-Val increase alongside $\mu$ (see Table 3). Interestingly, the lowest FID-A was observed at $\mu = 0.2$. These quantitative results suggest that larger values of $\mu$ tend to improve NatADiff performance; however, they do not capture the qualitative shift in sample structure. Large values of $\mu$ introduce two distinct phenomena: *dual class* samples, in which both the true and adversarial classes are present in the image (see Figure 7 (a) and (b)), and *flipped class* samples, in which the original class is entirely overwritten by the adversarial target (see Figure 7 (c) and (d)). Furthermore, as seen in Figure 7, the optimal value of $\mu$ appears to vary across true-adversarial class pairs. Thus, we select a conservative value of $\mu = 0.2$, as manual qualitative investigation found this did not lead to dual and flipped class samples, and experimental results indicate it best aligns with natural adversarial samples as measured by FID-A.

### G.3 EFFECT OF ADVERSARIAL BOUNDARY GUIDANCE PROMPT STRUCTURE

Adversarial boundary guidance requires that the true and adversarial classes be encoded into a textual prompt. To examine how this intersection prompt shapes attack performance, we compare NatADiff samples generated using different prompt templates. Concretely, we study four constructions–*"<class name> and <class name>"*, *"A photo of <class name> and <class name>"*, *"<class name> next to <class name>"*, and *"<class name> behind <class name>"*–and evaluate both class orderings, resulting in a total of 8 prompt formats. For each experiment we generate 150 targeted NatADiff samples using a ResNet-50 (He et al., 2016) surrogate model, and we report attack success rate, Inception Score (IS) (Salimans et al., 2016) and Fréchet Inception Distance (FID) (Fréchet, 1957). For FID, we evaluate with respect to both the ImageNet-Val (Deng et al., 2009) and ImageNet-A (Hendrycks et al., 2021) datasets to assess how closely NatADiff samples resemble natural images and known natural adversarial examples, respectively. The exact parameter settings for each experiment are provided in Table 7.

Overall, the prompt structure exerted relatively minimal influence on attack performance or image quality. Table 4 shows that average ASR varied by less than 7.7% across all prompt formats, indicating that adversarial boundary guidance is robust to the particular phrasing used to express the class intersection. The format *"A photo of <class name of $\tilde{y}$> and <class name of $y$>"* yielded the

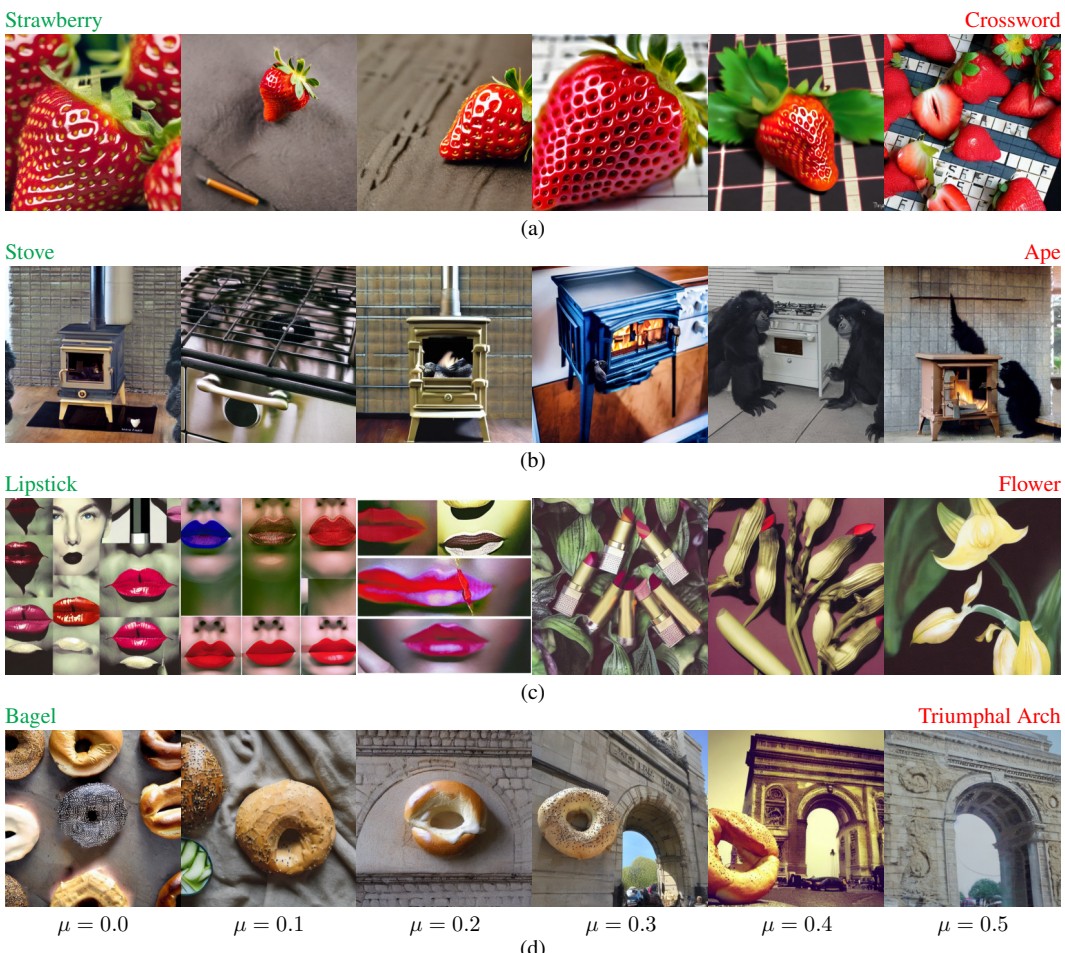

Figure 7: Samples generated by NatADiff using the same random seed while varying $\mu$ from $0.0$ to $0.5$, shown left to right. Green and red labels denote the true and adversarial classes, respectively. Images in (a) and (b) exhibit the *dual class* phenomenon, where large $\mu$ values cause objects from both the true and adversarial classes to appear. Images in (c) and (d) demonstrate the *flipped class* phenomenon, where large $\mu$ values causes the sample to fully adopt the adversarial class, suppressing the original class features.

highest average ASR, but this also coincided with the lowest IS, suggesting that the improved attack transferability may come at the cost of sample fidelity. For this reason, we adopt the prompt *"<class name of $\tilde{y}$> and <class name of $y$>"*, which attains the second-highest ASR while maintaining superior image fidelity, and therefore provides a more favourable balance between attack effectiveness and image quality.

### G.4 SELECTION OF ADVERSARIAL CLASSIFIER GUIDANCE STRENGTH

The adversarial classifier guidance term, $s$, governs the strength of the guidance provided by the surrogate classifier. This is analogous to the guidance term used in classifier-guided diffusion, where a trained classifier steers the sampling trajectory toward a desired class (Dhariwal & Nichol, 2021). Classifier-guided diffusion is a well-understood process that trades sample diversity for class adherence, and the optimal guidance strength varies with the underlying classifier model (Dhariwal & Nichol, 2021; Yu et al., 2023; Shen et al., 2024; Dai et al., 2024).

To investigate whether our understanding of classifier-guided diffusion extends to *adversarial* classifier guidance, we conduct an ablation study on the influence of the guidance strength term, $s$. For

Table 4: **Attack success rate** (ASR) and image quality of adversarial samples generated by NatADiff under targeted attack settings with varying adversarial boundary guidance prompt templates. We use a ResNet-50 (He et al., 2016) surrogate model. **Bold** values highlight the best score. White-box ASR (same surrogate and victim model) is denoted with an $^*$.

| Prompt Structure | Victim Model ASR (%) | | | | | | | | | Average ASR | IS ($\uparrow$) | FID-Val ($\downarrow$) | FID-A ($\downarrow$) |
| | CNNs | | | | | Transformers | | | | | | | |
| | RN-50 | Inc-v3 | RN-152 | AdvRes | AdvInc | ViT-H | Max-ViT | Swin-B | DeIT | | | | |
| $y$ and $\tilde{y}$ | 99.3* | 59.3 | 63.3 | 60.0 | 56.7 | 34.0 | 47.3 | 51.3 | 54.0 | 58.4 | 9.2 | 205.0 | 213.2 |
| $\tilde{y}$ and $y$ | 98.7* | 62.0 | 63.3 | 58.0 | 62.0 | 40.7 | 48.7 | 50.0 | 54.0 | 59.7 | 9.0 | 201.1 | 212.0 |
| A photo of $y$ and $\tilde{y}$ | 98.7* | 57.3 | 62.0 | 58.0 | 54.7 | 39.3 | 51.3 | 54.0 | 56.7 | 59.1 | 9.4 | 201.2 | 212.6 |
| A photo of $\tilde{y}$ and $y$ | 99.3* | 67.3 | 68.7 | 63.3 | 62.7 | 36.7 | 52.7 | 53.3 | 57.3 | 62.4 | 8.4 | 204.1 | 211.6 |
| $y$ next to $\tilde{y}$ | 97.3* | 54.7 | 52.7 | 52.0 | 56.7 | 33.3 | 46.0 | 47.3 | 52.0 | 54.7 | 8.4 | 205.6 | 214.6 |
| $\tilde{y}$ next to $y$ | 100.0* | 59.3 | 58.7 | 57.3 | 56.7 | 32.7 | 50.0 | 48.7 | 52.0 | 57.3 | 8.8 | 199.4 | 210.2 |
| $y$ behind $\tilde{y}$ | 98.0* | 59.3 | 60.0 | 60.0 | 55.3 | 41.3 | 48.7 | 52.0 | 55.3 | 58.9 | 8.9 | 205.4 | 216.3 |
| $\tilde{y}$ behind $y$ | 99.3* | 58.0 | 57.3 | 52.0 | 58.7 | 36.7 | 44.0 | 50.7 | 50.0 | 56.3 | 8.4 | 197.9 | 208.7 |

each ablation, we generate 50 targeted NatADiff samples using using classifier guidance strengths of $s = 20, 50, 100$, and 200. We report attack success rate (ASR), Inception Score (IS) (Salimans et al., 2016), and Fréchet Inception Distance (FID) (Fréchet, 1957) across ResNet-50 (He et al., 2016), Inception-v3 (Szegedy et al., 2016), and ViT-H (Dosovitskiy et al., 2021) classifiers using an adversarial boundary guidance strength of $\mu = 0.2$. We take FID with respect to both ImageNet-Val (Deng et al., 2009) and ImageNet-A (Hendrycks et al., 2021) to assess how closely samples resemble natural images and known natural adversarial examples, respectively. Additionally, we examine the average classifier gradient over the diffusion sampling path. At each sampling time $t$, we compute the average gradient as $\frac{1}{n}\sum_{j=1}^{n}\left(\nabla_{\boldsymbol{x}_t}\log(p(y|\boldsymbol{x}_t))\right)_j$. The exact parameter settings for each experiment are provided in Table 7.

From Table 5, we observe that ASR increases with adversarial classifier guidance strength. Gradient stability also depends on the guidance strength, as illustrated in Figures 8, 9, and 10. When the guidance strength is too low, NatADiff's ASR drops, and the gradient does not smoothly converge to 0 as $t \to 0$. This is likely because the guidance is insufficient to push the sample into regions of the adversarial class on the classifier manifold—evidenced by low ASR scores coinciding with weak guidance. Conversely, sufficiently large guidance strengths yield substantially higher ASR and gradients that smoothly converge to 0 as the sample enters the desired region of the manifold.

We find that FID scores degrade as guidance strength increases, while IS remains relatively unaffected in all cases except the ViT-H classifier. This suggests that increasing guidance strength reduces sample diversity, consistent with the conventional understanding of classifier-guided diffusion. However, excessive classifier guidance can also "push" samples off the image manifold, as reflected by the IS degradation observed for ViT-H. This aligns with findings from the main paper, where targeted attacks against ViT-H yielded reduced image quality (see Appendix K.5).

Overall, these results indicate that adversarial classifier guidance operates according to the same underlying mechanisms as standard classifier-guided diffusion. Following Dhariwal & Nichol (2021) and Dai et al. (2024), for each classifier we manually tuned $s$ by incrementally increasing it until NatADiff successfully generated adversarial samples. We additionally implemented an adaptive scaling scheme that increased the guidance strength if a sample failed to fool the surrogate classifier.

# H  MANIFOLD PLOTS

In the main paper we claim that adversarial boundary guidance directs the diffusion sampling trajectory towards class intersections. To further support the validity of this claim, we visualise the learned classifier manifolds using both UMAP (McInnes & Healy, 2018) and t-SNE (van der Maaten & Hinton, 2008), which are dimensionality-reduction techniques that allow the image manifolds to be viewed on a 2D plane. UMAP provides a topology-preserving embedding that tends to maintain global structure, while t-SNE preserves local neighbourhood relationships at the cost of deforming global structure. We use both methods to provide complementary views that illustrate how NatADiff samples relate to the true-class and adversarial-class distributions.

To construct the image manifolds, we use NatADiff to generate adversarial samples for 12 true–adversarial class pairs. All adversarial samples are generated using a ResNet-50 (He et al., 2016)

Table 5: **Attack success rate** (ASR) and image quality of adversarial samples generated by NatADiff under targeted attack settings with varying adversarial classifier guidance strength, $s$. We use a ResNet-50 (He et al., 2016) surrogate model. **Bold** values highlight the best score. White-box ASR (same surrogate and victim model) is denoted with an *.

| Surrogate Model | Attack | Victim Model ASR (%) | | | | | | | | | Average ASR | IS (↑) | FID-Val (↓) | FID-A (↓) |
|---|---|---|---|---|---|---|---|---|---|---|---|---|---|---|
| | | CNNs | | | | | Transformers | | | | | | | |
| | | RN-50 | Inc-v3 | RN-152 | AdvRes | AdvInc | ViT-H | Max-ViT | Swin-B | DeIT | | | | |
| RN-50 | NatADiff ($s = 20$) | 38.0* | 36.0 | 34.0 | 38.0 | 36.0 | 32.0 | 32.0 | 30.0 | 26.0 | 33.6 | 4.4 | **233.5** | **280.5** |
| | NatADiff ($s = 50$) | 62.0* | 42.0 | 46.0 | 38.0 | 46.0 | 36.0 | 38.0 | 36.0 | 36.0 | 42.2 | 4.3 | 242.4 | 282.7 |
| | NatADiff ($s = 100$) | 86.0* | 50.0 | 54.0 | 50.0 | 46.0 | 36.0 | 44.0 | 48.0 | 50.0 | 51.6 | 4.5 | 266.5 | 293.3 |
| | NatADiff ($s = 200$) | **96.0*** | 62.0 | 76.0 | **68.0** | 60.0 | **46.0** | 62.0 | 66.0 | 66.0 | 66.9 | 4.7 | 319.0 | 326.2 |
| Inc-v3 | NatADiff ($s = 20$) | 30.0* | 46.0 | 28.0 | 36.0 | 36.0 | 26.0 | 26.0 | 26.0 | 34.0 | 32.0 | 3.9 | **223.4** | **273.6** |
| | NatADiff ($s = 50$) | 44.0* | 70.0 | 44.0 | 40.0 | 48.0 | 34.0 | 34.0 | 42.0 | 42.0 | 44.2 | **4.2** | 244.5 | 284.4 |
| | NatADiff ($s = 100$) | 50.0* | 94.0 | 54.0 | 58.0 | 54.0 | 38.0 | 44.0 | 46.0 | 46.0 | 53.8 | **4.2** | 269.2 | 297.2 |
| | NatADiff ($s = 200$) | 72.0* | **96.0** | 66.0 | **68.0** | **80.0** | **46.0** | **54.0** | 60.0 | 56.0 | 66.4 | 4.1 | 319.4 | 320.8 |
| ViT-H | NatADiff ($s = 20$) | 28.0* | 34.0 | 26.0 | 30.0 | 28.0 | 30.0 | 24.0 | 30.0 | 28.0 | 28.7 | 3.9 | **224.7** | **279.0** |
| | NatADiff ($s = 50$) | 32.0* | 46.0 | 38.0 | 40.0 | 32.0 | 38.0 | 36.0 | 34.0 | 36.0 | 36.9 | **4.0** | 240.0 | 286.4 |
| | NatADiff ($s = 100$) | 46.0* | 46.0 | 46.0 | 40.0 | 44.0 | 52.0 | 42.0 | 48.0 | 48.0 | 45.8 | 3.9 | 242.2 | 279.5 |
| | NatADiff ($s = 200$) | 78.0* | 78.0 | **80.0** | **78.0** | 78.0 | **86.0** | **70.0** | **72.0** | **82.0** | **78.0** | 2.9 | 281.3 | 294.5 |

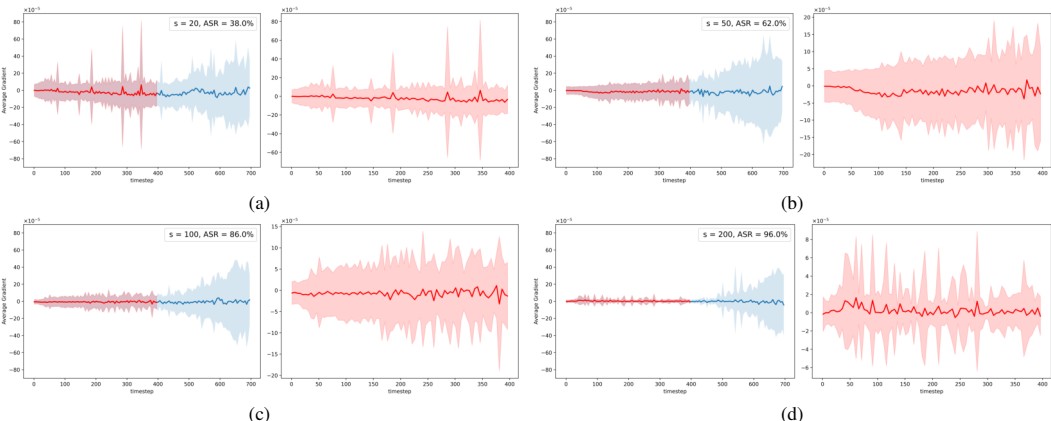

Figure 8: ResNet-50 (He et al., 2016) classifier gradient with respect to $x_t$ plotted against sampling time. The solid line denotes the average across the experimental run, while the shaded region is plus/minus 1 standard deviation. The final 400 sampling steps are enhanced and displayed in the second panel of each subfigure. For each subfigure, the classifier guidance strength ($s$) and white-box ASR are: (a) $s = 20$, ASR $= 38.0\%$; (b) $s = 50$, ASR $= 62.0\%$; (c) $s = 100$, ASR $= 86.0\%$; (d) $s = 200$, ASR $= 96.0\%$. Please zoom in for improved visibility.

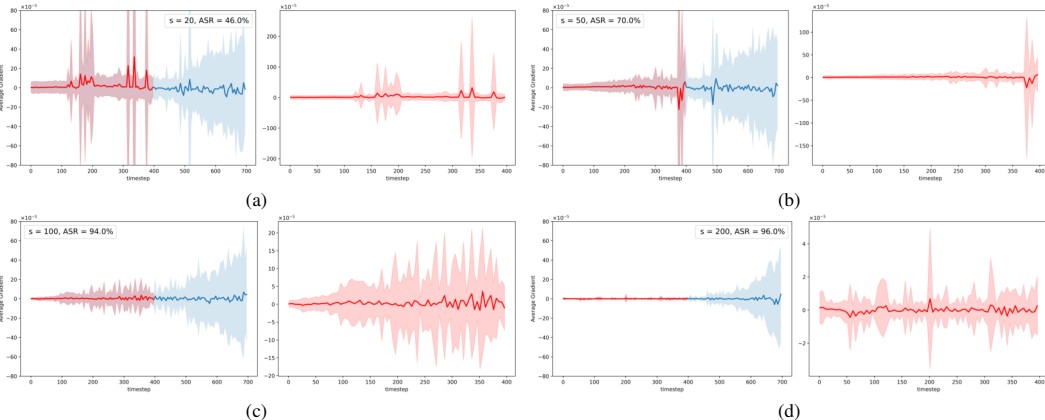

Figure 9: Inception-v3 (Szegedy et al., 2016) classifier gradient with respect to $x_t$ plotted against sampling time. The solid line denotes the average across all experiment runs, while the shaded region is plus/minus 1 standard deviation. The final 400 sampling steps are enhanced and displayed in the second panel of each subfigure. For each subfigure, the classifier guidance strength ($s$) and white-box ASR are: (a) $s = 20$, ASR $= 46.0\%$; (b) $s = 50$, ASR $= 70.0\%$; (c) $s = 100$, ASR $= 94.0\%$; (d) $s = 200$, ASR $= 96.0\%$. Please zoom in for improved visibility.

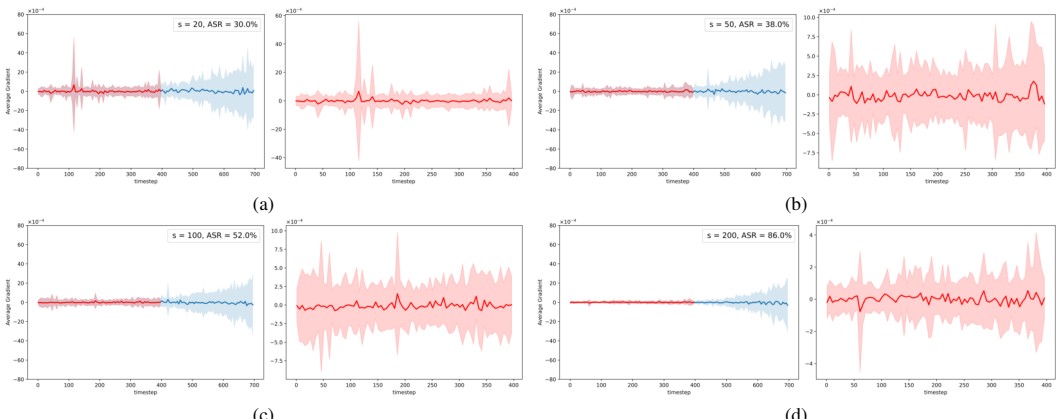

Figure 10: ViT-H (Dosovitskiy et al., 2021) classifier gradient with respect to $x_t$ plotted against sampling time. The solid line denotes the average across the experimental run, while the shaded region is plus/minus 1 standard deviation. The final 400 sampling steps are enhanced and displayed in the second panel of each subfigure. For each subfigure, the classifier guidance strength ($s$) and white-box ASR are: (a) $s = 20$, ASR = 30.0% (b) $s = 50$, ASR = 38.0%, (c) $s = 100$, ASR = 52.0%, (d) $s = 200$, ASR = 86.0%. Please zoom in for improved visibility.

surrogate model, and we pass UMAP and t-SNE the penultimate-layer feature embeddings from this classifier. The NatADiff samples are plotted alongside the feature embeddings of clean images from the true and adversarial classes taken from the ImageNet dataset (Deng et al., 2009). The resulting visualisations are provided in Figures 11 and 12.

Across both embeddings, NatADiff samples consistently form one or more distinct clusters that fall between the manifolds of the true and adversarial target classes. This positioning directly supports the main paper's claim: NatADiff guides samples toward class intersections, generating samples that naturally fall near class boundaries. Furthermore, this aligns with the established understanding of natural adversarial samples (test-time errors), which are believed to arise when classifiers rely on spurious contextual cues rather than truly discriminative features. As a sample moves closer to a classifier's decision boundary, the prevalence of contextual cues from the adversarial class increases, which provides a natural explanation for NatADiff's strong attack performance.

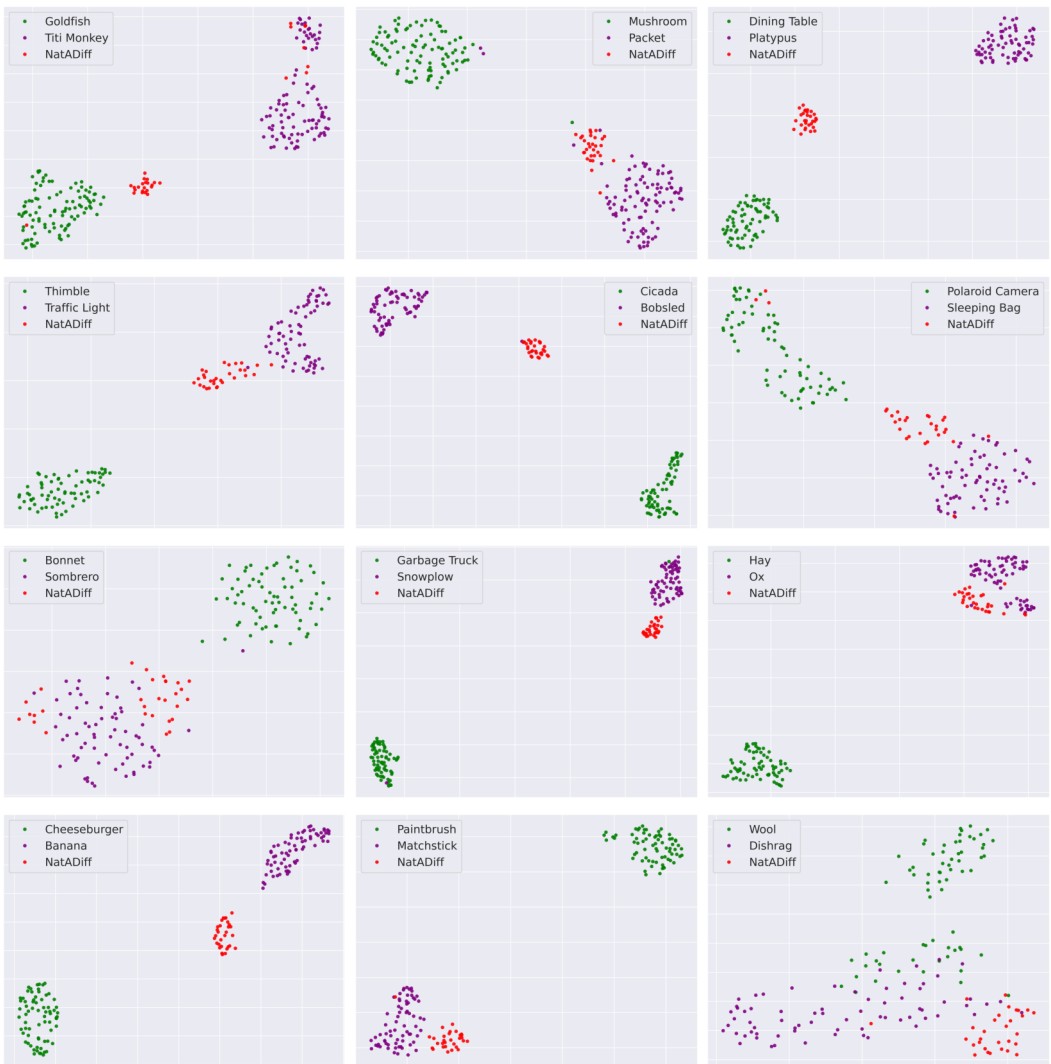

Figure 11: UMAP (McInnes & Healy, 2018) plots of NatADiff samples alongside the ground truth manifolds. **Green** and **purple** samples are real images of the true and adversarial classes taken from the ImageNet dataset (Deng et al., 2009). **Red** samples are generated by NatADiff using a ResNet-50 (He et al., 2016) surrogate model. UMAP is applied to embeddings taken from the penultimate layer of a ResNet-50 classifier. Please zoom in for a better visual.

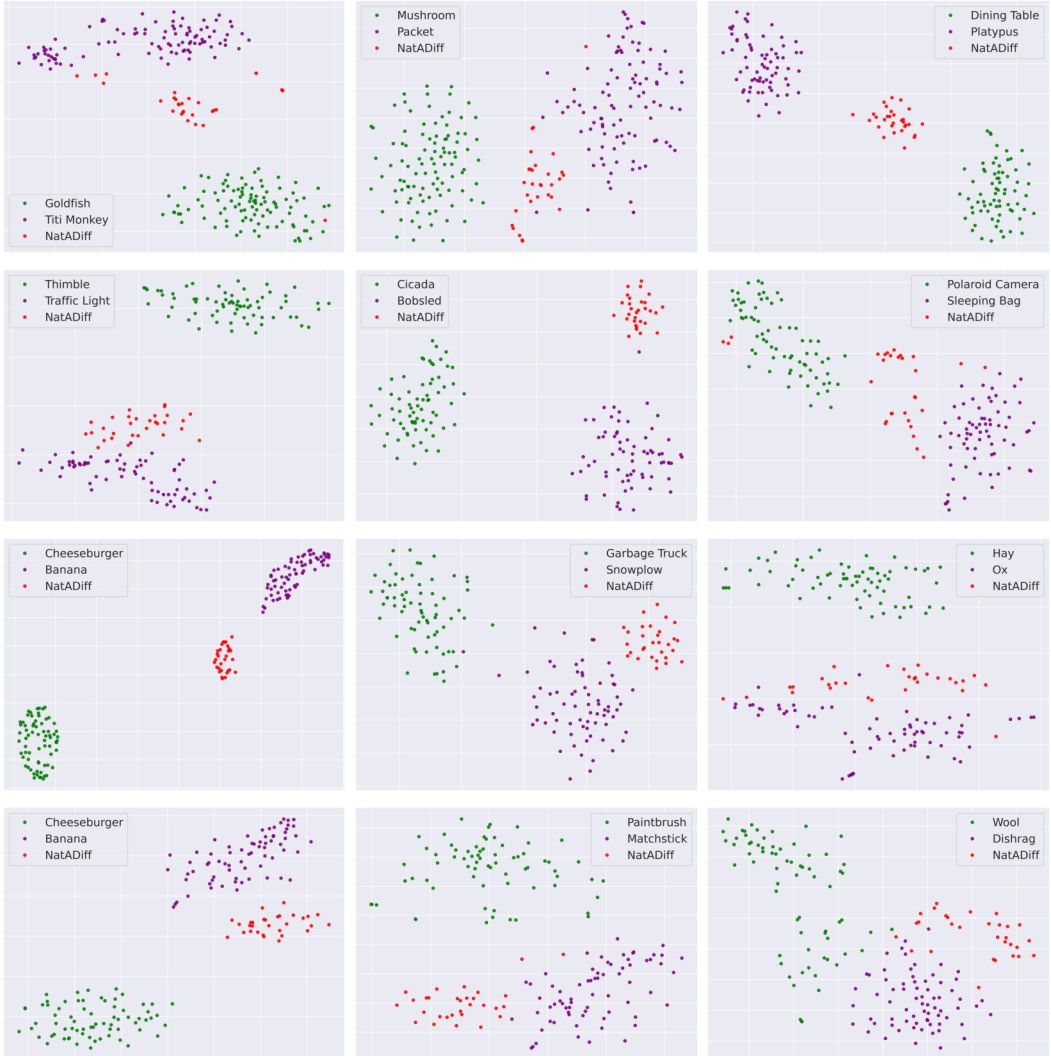

Figure 12: t-SNE (van der Maaten & Hinton, 2008) plots of NatADiff samples alongside the ground truth manifolds. **Green** and **purple** samples are real images of the true and adversarial classes taken from the ImageNet dataset (Deng et al., 2009). **Red** samples are generated by NatADiff using a ResNet-50 (He et al., 2016) surrogate model. t-SNE is applied to embeddings taken from the penultimate layer of a ResNet-50 classifier. Please zoom in for a better visual.

## I NatADiff on Oxford Pets

In the main paper, we present results for NatADiff when targeting ImageNet classifiers exclusively. To strengthen the claims made in the main paper, we extend our evaluation to the Oxford Pets dataset (Parkhi et al., 2012), which is more fine-grained and presents a unique challenge as opposed to ImageNet. In this setting, the adversarial structural elements introduced by NatADiff will need to be more subtle.

We use NatADiff to attack ResNet-50 (He et al., 2016), ResNet-152 (He et al., 2016), and ViT (Dosovitskiy et al., 2021) classifiers with weights sourced from HuggingFace (Wolf et al., 2020). For each class in Oxford Pets, we generate 5 adversarial samples using both targeted (randomly selected) and untargeted (similarity-based) variants of NatADiff. We compare NatADiff against PGD (Madry et al., 2018), AutoAttack (Croce & Hein, 2020), and adversarial classifier guidance (AdvClass) (Dai et al., 2024). We exclude comparisons with NCF (Yuan et al., 2022), DiffAttack (Chen et al., 2025), and ACA (Chen et al., 2023b), as these methods do not provide variants targeting Oxford Pets classifiers. We report attack success rate (ASR) and the image quality metrics BRISQUE (Mittal et al., 2012) and TReS (Golestaneh et al., 2022).

From Table 6, we observe that NatADiff behaves similarly to the ImageNet study, outperforming other methods in terms of attack transferability while maintaining competitive white-box ASR. NatADiff also typically achieves better BRISQUE and is close to AdvClass in terms of TReS, suggesting that NatADiff produces high-fidelity samples, as shown in Figure 13. Overall, NatADiff generalizes well to fine-grained datasets, with adversarial boundary guidance extending to minute structural changes. These results are consistent with the strong performance of similarity-targeted attacks in the ImageNet study, indicating that adversarial boundary guidance is particularly effective when classes share similar structural elements.

Table 6: **Attack success rate** (%) and **image quality** of adversarial samples generated for the Oxford Pets dataset (Parkhi et al., 2012). Superscripts T and U denote random and similarity targeted attacks, respectively. **Bold** and underlined values highlight the best and second best scores. White-box ASR (same surrogate and victim model) is denoted with an $*$. Note we do not report image quality for constrained perturbation-based attacks (these attacks make imperceptible image alterations).

| Surrogate Model | Attack | Victim Model ASR (%) | | | Average ASR | BRISQUE ($\downarrow$) | TReS ($\uparrow$) |
|---|---|---|---|---|---|---|---|
| | | ResNet-50 | ResNet-152 | ViT | | | |
| | Clean | 6.0 | 8.6 | 3.8 | 6.1 | - | - |
| ResNet-50 | PGD | **100**$^*$ | 10.3 | 3.8 | 38.0 | - | - |
| | AA | **100**$^*$ | 8.1 | 4.3 | 37.5 | - | - |
| | AdvClass$^T$ | 99.5$^*$ | 17.3 | 19.5 | 45.4 | 5.3 | 80.9 |
| | AdvClass$^U$ | 98.9$^*$ | 19.5 | 16.8 | 45.0 | 5.0 | **81.3** |
| | NatADiff$^T$ | 97.8$^*$ | 30.8 | 27.6 | 52.1 | **4.2** | 78.9 |
| | NatADiff$^U$ | 96.8$^*$ | **37.3** | **35.7** | **56.6** | 5.5 | 80.2 |
| ResNet-152 | PGD | 14.6 | 99.5$^*$ | 9.2 | 41.1 | - | - |
| | AA | 12.4 | **100**$^*$ | 6.5 | 39.6 | - | - |
| | AdvClass$^T$ | 34.6 | 98.4$^*$ | 22.2 | 51.7 | 7.2 | 80.3 |
| | AdvClass$^U$ | 37.3 | 98.9$^*$ | 26.5 | 54.2 | 6.9 | **80.9** |
| | NatADiff$^T$ | 51.9 | 95.7$^*$ | 37.3 | 61.6 | 5.7 | 79.4 |
| | NatADiff$^U$ | **57.8** | 98.9$^*$ | **48.6** | **68.5** | **5.2** | 79.6 |
| ViT | PGD | 13.0 | 10.8 | **100**$^*$ | | - | - |
| | AA | 7.6 | 8.6 | 97.3$^*$ | 37.8 | - | - |
| | AdvClass$^T$ | 28.1 | 22.2 | **100**$^*$ | 50.1 | 6.7 | 80.2 |
| | AdvClass$^U$ | 24.3 | 24.3 | **100**$^*$ | 49.5 | 6.7 | 80.1 |
| | NatADiff$^T$ | **56.2** | 29.2 | **100**$^*$ | 61.8 | **4.9** | **80.3** |
| | NatADiff$^U$ | 54.6 | **46.5** | **100**$^*$ | **67.0** | 5.3 | 80.1 |

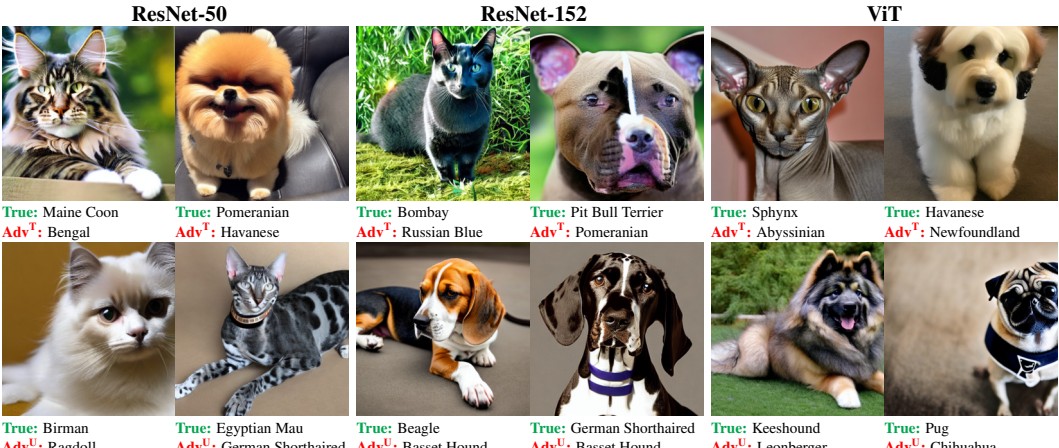

Figure 13: Adversarial samples generated by NatADiff for the Oxford Pets dataset (Parkhi et al., 2012) using ResNet-50 (He et al., 2016), ResNet-152 (He et al., 2016), and ViT (Dosovitskiy et al., 2021) surrogate models (see column labels). We report the true class and adversarial target for each image. Superscripts T and U denote random and similarity targeted attacks, respectively.

## J   NATADIFF EXPERIMENT PARAMETER SETTINGS

Here we provide the NatADiff parameter values used in our experiments (see Tables 7 and 8).

Table 7: **Experiment parameters** used in diffusion-based adversarial sampling experiments on the ImageNet dataset (Deng et al., 2009). Parameters refer to those defined in Algorithms 1 and 2. Experiments were conducted with ResNet-50 (He et al., 2016), Inception-v3 (Szegedy et al., 2016), and ViT-H (Dosovitskiy et al., 2021) surrogate models.

| Surrogate Model | Attack | NatADiff Parameters | | | | | | | | | | | |
| --- | --- | --- | --- | --- | --- | --- | --- | --- | --- | --- | --- | --- | --- |
| | | $\omega$ | $\rho$ | $\mu$ | $s$ | $R$ | $r_l$ | $r_u$ | $c_l$ | $c_u$ | $S$ | $\delta_\mu$ | $\delta_s$ |
| RN-50 | NatADiff[T] (Aug) | 7.5 | 7.5 | 0.2 | 50 | 5 | 500 | 800 | 0 | 700 | 5 | 0 | 15 |
| | NatADiff[T] ($\mu = 0.0$) | 7.5 | 7.5 | 0.0 | 50 | 5 | 500 | 800 | 0 | 700 | 5 | 0 | 15 |
| | NatADiff[T] ($\mu = 0.1$) | 7.5 | 7.5 | 0.1 | 50 | 5 | 500 | 800 | 0 | 700 | 5 | 0 | 15 |
| | NatADiff[T] ($\mu = 0.2$) | 7.5 | 7.5 | 0.2 | 50 | 5 | 500 | 800 | 0 | 700 | 5 | 0 | 15 |
| | NatADiff[T] ($\mu = 0.3$) | 7.5 | 7.5 | 0.3 | 50 | 5 | 500 | 800 | 0 | 700 | 5 | 0 | 15 |
| | NatADiff[T] ($\mu = 0.4$) | 7.5 | 7.5 | 0.4 | 50 | 5 | 500 | 800 | 0 | 700 | 5 | 0 | 15 |
| | NatADiff[T] ($\mu = 0.5$) | 7.5 | 7.5 | 0.5 | 50 | 5 | 500 | 800 | 0 | 700 | 5 | 0 | 15 |
| | NatADiff[T] (Prompt) | 7.5 | 7.5 | 0.2 | 50 | 5 | 500 | 800 | 0 | 700 | 5 | 0 | 15 |
| | NatADiff[T] ($s = 20$) | 7.5 | 7.5 | 0.2 | 20 | 5 | 500 | 800 | 0 | 700 | 5 | 0 | 15 |
| | NatADiff[T] ($s = 50$) | 7.5 | 7.5 | 0.2 | 50 | 5 | 500 | 800 | 0 | 700 | 5 | 0 | 15 |
| | NatADiff[T] ($s = 100$) | 7.5 | 7.5 | 0.2 | 100 | 5 | 500 | 800 | 0 | 700 | 5 | 0 | 15 |
| | NatADiff[T] ($s = 200$) | 7.5 | 7.5 | 0.2 | 200 | 5 | 500 | 800 | 0 | 700 | 5 | 0 | 15 |
| | AdvClass[T] | 7.5 | 0.0 | 0.0 | 500 | 0 | 0 | 0 | 0 | 200 | 5 | 0 | 250 |
| | AdvClass[U] | 7.5 | 0.0 | 0.0 | 500 | 0 | 0 | 0 | 0 | 200 | 5 | 0 | 250 |
| | NatADiff[T] | 7.5 | 7.5 | 0.2 | 50 | 5 | 500 | 800 | 0 | 700 | 5 | 0 | 15 |
| | NatADiff[U] | 7.5 | 7.5 | 0.2 | 50 | 5 | 500 | 800 | 0 | 700 | 5 | 0 | 25 |
| Inc-v3 | NatADiff[T] ($s = 20$) | 7.5 | 7.5 | 0.2 | 20 | 5 | 500 | 800 | 0 | 700 | 5 | 0 | 20 |
| | NatADiff[T] ($s = 50$) | 7.5 | 7.5 | 0.2 | 50 | 5 | 500 | 800 | 0 | 700 | 5 | 0 | 20 |
| | NatADiff[T] ($s = 100$) | 7.5 | 7.5 | 0.2 | 100 | 5 | 500 | 800 | 0 | 700 | 5 | 0 | 20 |
| | NatADiff[T] ($s = 200$) | 7.5 | 7.5 | 0.2 | 200 | 5 | 500 | 800 | 0 | 700 | 5 | 0 | 20 |
| | AdvClass[T] | 7.5 | 0.0 | 0.0 | 500 | 0 | 0 | 0 | 0 | 200 | 5 | 0 | 250 |
| | AdvClass[U] | 7.5 | 0.0 | 0.0 | 500 | 0 | 0 | 0 | 0 | 200 | 5 | 0 | 250 |
| | NatADiff[T] | 7.5 | 7.5 | 0.2 | 50 | 5 | 500 | 800 | 0 | 700 | 5 | 0 | 20 |
| | NatADiff[U] | 7.5 | 7.5 | 0.2 | 50 | 5 | 500 | 800 | 0 | 700 | 5 | 0 | 20 |
| ViT-H | NatADiff[T] ($s = 20$) | 7.5 | 7.5 | 0.2 | 20 | 5 | 500 | 800 | 0 | 700 | 5 | 0 | 50 |
| | NatADiff[T] ($s = 50$) | 7.5 | 7.5 | 0.2 | 50 | 5 | 500 | 800 | 0 | 700 | 5 | 0 | 50 |
| | NatADiff[T] ($s = 100$) | 7.5 | 7.5 | 0.2 | 100 | 5 | 500 | 800 | 0 | 700 | 5 | 0 | 50 |
| | NatADiff[T] ($s = 200$) | 7.5 | 7.5 | 0.2 | 200 | 5 | 500 | 800 | 0 | 700 | 5 | 0 | 50 |
| | AdvClass[T] | 7.5 | 0.0 | 0.0 | 500 | 0 | 0 | 0 | 0 | 200 | 5 | 0 | 250 |
| | AdvClass[U] | 7.5 | 0.0 | 0.0 | 500 | 0 | 0 | 0 | 0 | 200 | 5 | 0 | 250 |
| | NatADiff[T] | 7.5 | 7.5 | 0.2 | 100 | 5 | 500 | 800 | 0 | 700 | 5 | 0 | 50 |
| | NatADiff[U] | 7.5 | 7.5 | 0.2 | 100 | 5 | 500 | 800 | 0 | 700 | 5 | 0 | 50 |

Table 8: **Experiment parameters** used in diffusion-based adversarial sampling experiments on the Oxford Pets dataset (Parkhi et al., 2012). Parameters refer to those defined in Algorithms 1 and 2. Experiments were conducted with ResNet-50 (He et al., 2016), ResNet-152 (He et al., 2016), and ViT (Dosovitskiy et al., 2021) surrogate models.

| Surrogate Model | Attack | NatADiff Parameters | | | | | | | | | | | |
|---|---|---|---|---|---|---|---|---|---|---|---|---|---|
| | | $\omega$ | $\rho$ | $\mu$ | $s$ | $R$ | $r_l$ | $r_u$ | $c_l$ | $c_u$ | $S$ | $\delta_\mu$ | $\delta_s$ |
| RN-50 | AdvClass$^T$ | 7.5 | 0.0 | 0.0 | 500 | 0 | 0 | 0 | 0 | 200 | 5 | 0 | 250 |
| | AdvClass$^U$ | 7.5 | 0.0 | 0.0 | 500 | 0 | 0 | 0 | 0 | 200 | 5 | 0 | 250 |
| | NatADiff$^T$ | 7.5 | 7.5 | 0.2 | 50 | 5 | 500 | 800 | 0 | 600 | 5 | 0 | 25 |
| | NatADiff$^U$ | 7.5 | 7.5 | 0.2 | 50 | 5 | 500 | 800 | 0 | 600 | 5 | 0 | 25 |
| RN-152 | AdvClass$^T$ | 7.5 | 0.0 | 0.0 | 500 | 0 | 0 | 0 | 0 | 200 | 5 | 0 | 250 |
| | AdvClass$^U$ | 7.5 | 0.0 | 0.0 | 500 | 0 | 0 | 0 | 0 | 200 | 5 | 0 | 250 |
| | NatADiff$^T$ | 7.5 | 7.5 | 0.2 | 50 | 5 | 500 | 800 | 0 | 600 | 5 | 0 | 25 |
| | NatADiff$^U$ | 7.5 | 7.5 | 0.2 | 50 | 5 | 500 | 800 | 0 | 600 | 5 | 0 | 25 |
| ViT | AdvClass$^T$ | 7.5 | 0.0 | 0.0 | 500 | 0 | 0 | 0 | 0 | 200 | 5 | 0 | 250 |
| | AdvClass$^U$ | 7.5 | 0.0 | 0.0 | 500 | 0 | 0 | 0 | 0 | 200 | 5 | 0 | 250 |
| | NatADiff$^T$ | 7.5 | 7.5 | 0.2 | 50 | 5 | 500 | 800 | 0 | 600 | 5 | 0 | 25 |
| | NatADiff$^U$ | 7.5 | 7.5 | 0.2 | 50 | 5 | 500 | 800 | 0 | 600 | 5 | 0 | 25 |

# K  ADDITIONAL NATADIFF SAMPLES

We provide NatADiff samples alongside the classification scores of ResNet-50 (He et al., 2016), Inception-v3 (Szegedy et al., 2016), ViT-H (Dosovitskiy et al., 2021), and adversarially trained ResNet-50 and Inception victim models (Kurakin et al., 2018). Samples were generated using ResNet-50 (He et al., 2016), Inception-v3 (Szegedy et al., 2016), and ViT-H (Dosovitskiy et al., 2021) surrogate models.

## K.1  MIXED SAMPLES

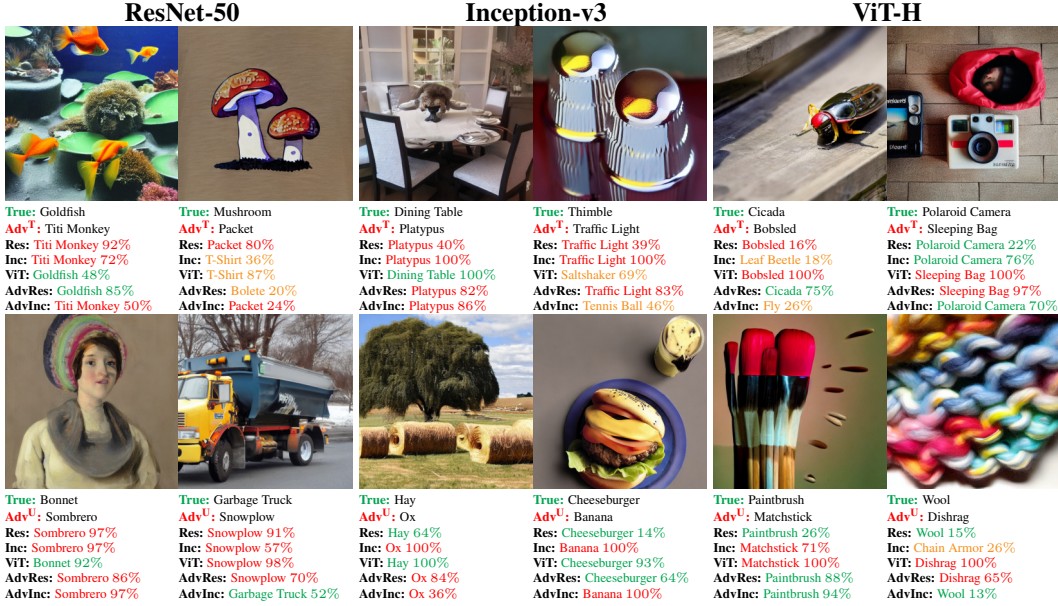

Figure 14: Adversarial samples generated using NatADiff with ResNet-50 (He et al., 2016), Inception-v3 (Szegedy et al., 2016), and ViT-H (Dosovitskiy et al., 2021) surrogate models (see column labels). We report the true class, adversarial target, and classification scores of the surrogate and adversarially trained ResNet-50 and Inception victim models (Kurakin et al., 2018). Superscripts T and U denote random and similarity targeted attacks, respectively.

## K.2 RESNET-50 SAMPLES

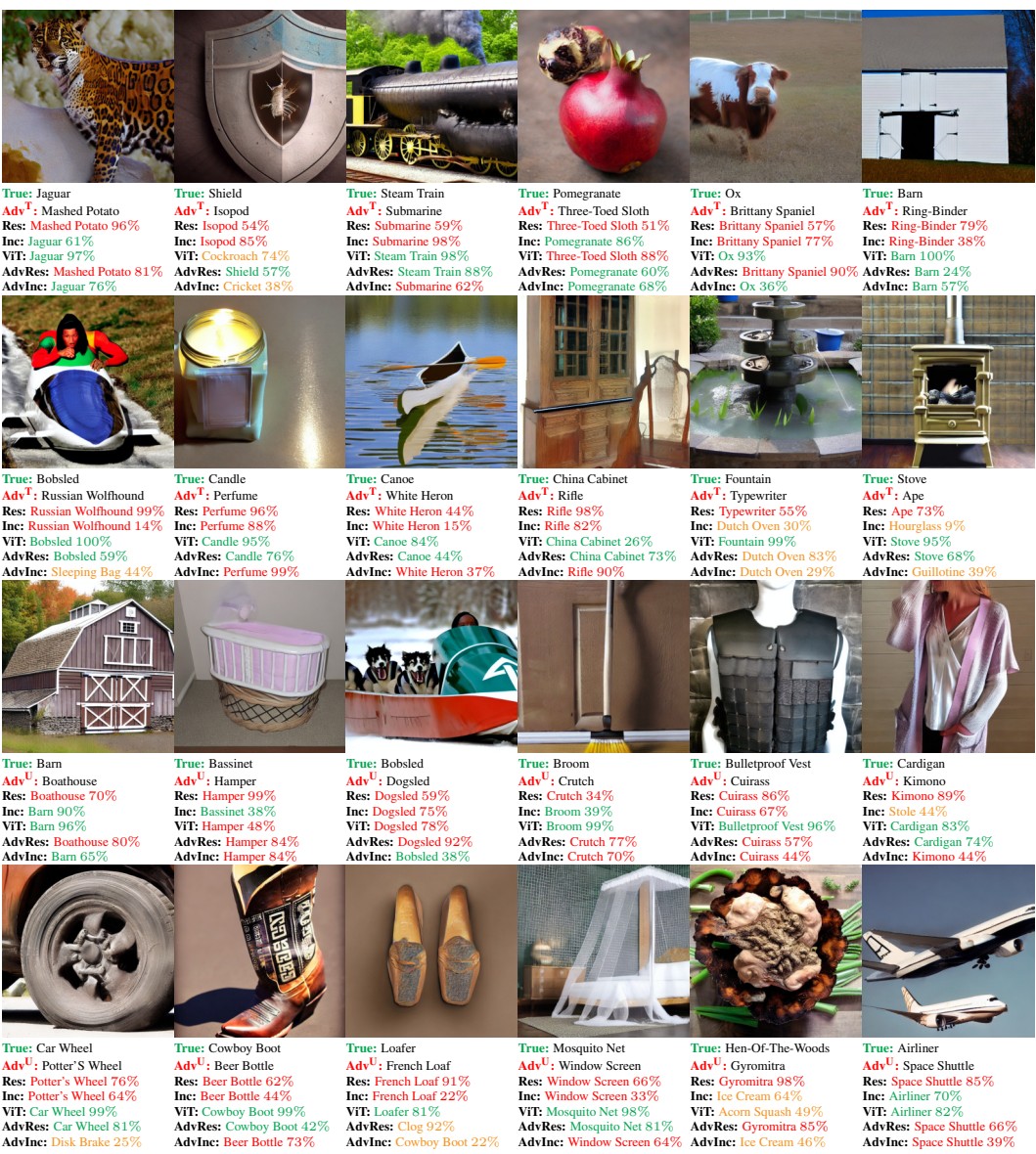

Figure 15: Adversarial samples generated by NatADiff with a ResNet-50 (He et al., 2016) surrogate model. We report the true class, adversarial target, and classification scores of ResNet-50 (He et al., 2016), Inception-v3 (Szegedy et al., 2016), ViT-H (Dosovitskiy et al., 2021), and adversarially trained ResNet-50 and Inception victim models (Kurakin et al., 2018). Superscripts T and U indicate targeted and untargeted (similarity-based) attacks, respectively.

### K.3 INCEPTION-V3 SAMPLES

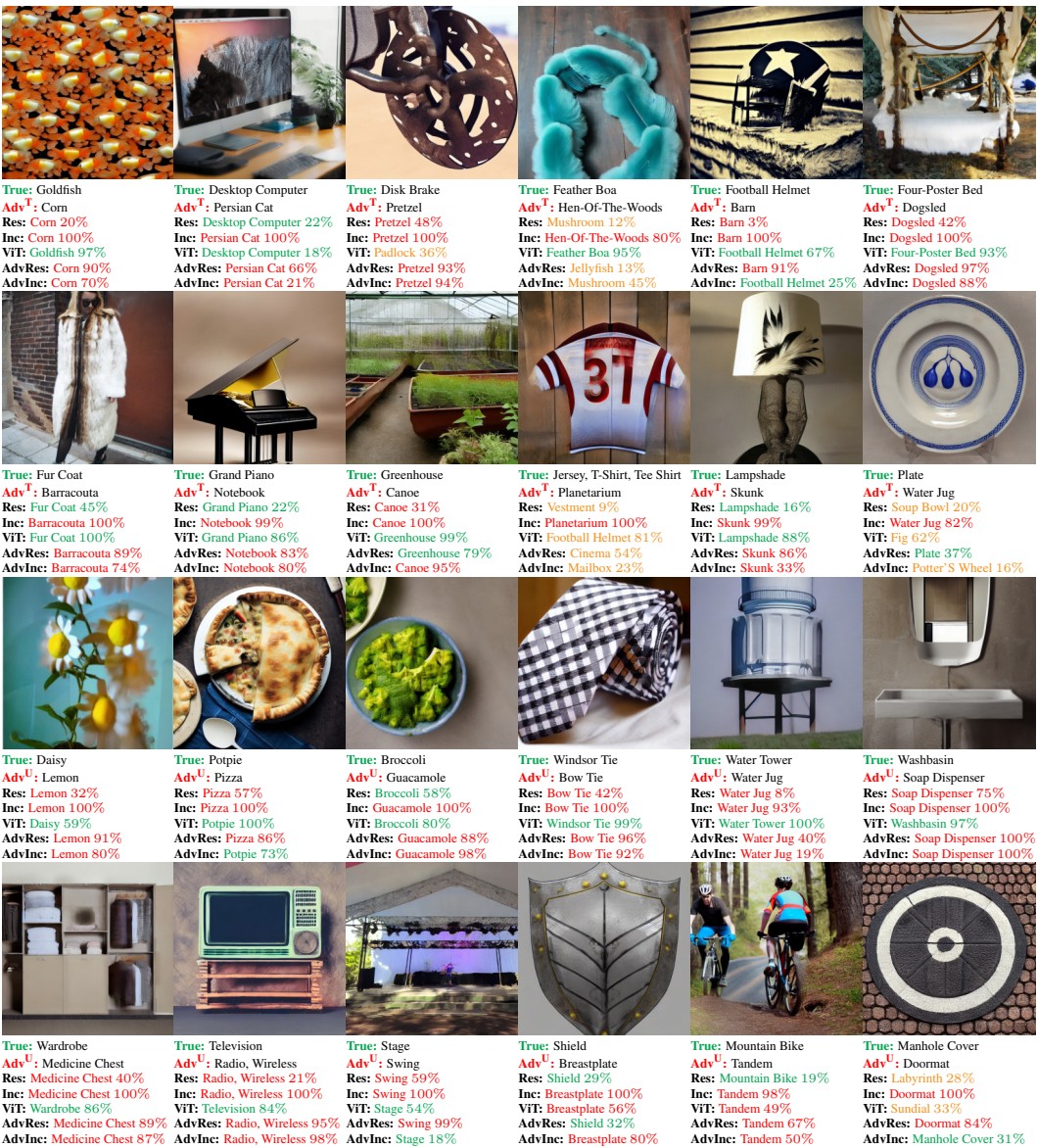

Figure 16: Adversarial samples generated by NatADiff with an Inception-v3 (Szegedy et al., 2016) surrogate model. We report the true class, adversarial target, and classification scores of ResNet-50 (He et al., 2016), Inception-v3 (Szegedy et al., 2016), ViT-H (Dosovitskiy et al., 2021), and adversarially trained ResNet-50 and Inception victim models (Kurakin et al., 2018). Superscripts T and U indicate targeted and untargeted (similarity-based) attacks, respectively.

## K.4 VIT SAMPLES

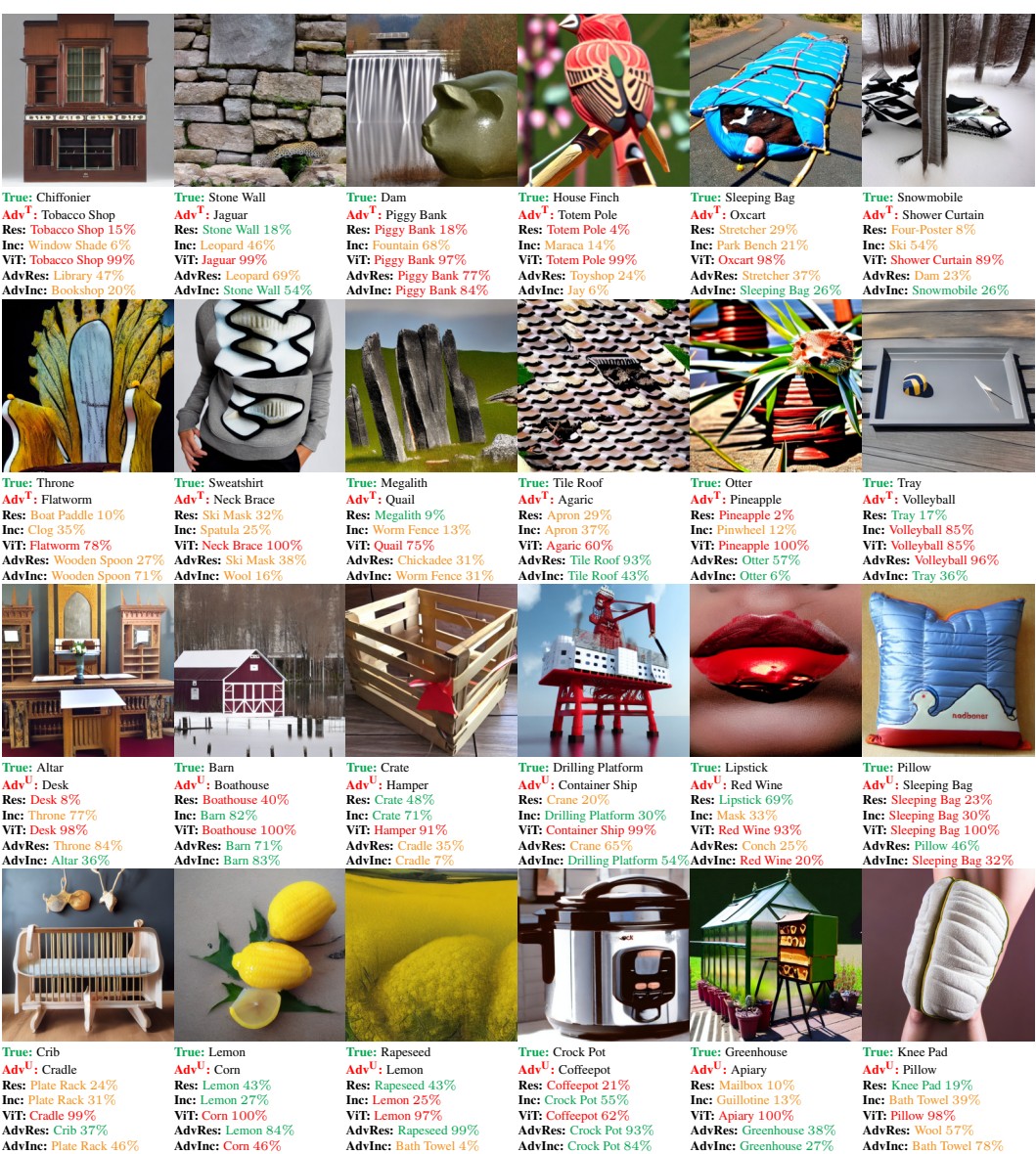

Figure 17: Adversarial samples generated by NatADiff with a ViT-H (Dosovitskiy et al., 2021) surrogate model. We report the true class, adversarial target, and classification scores of ResNet-50 (He et al., 2016), Inception-v3 (Szegedy et al., 2016), ViT-H (Dosovitskiy et al., 2021), and adversarially trained ResNet-50 and Inception victim models (Kurakin et al., 2018). Superscripts T and U indicate targeted and untargeted (similarity-based) attacks, respectively.

### K.5 LOW-QUALITY TARGETED ViT SAMPLES

Targeted ViT-H samples show degraded image quality, as seen in Figure 18 and supported by the IS and FID-Val scores in Table 1. Targeted attacks typically blend features from disparate classes, which places greater demands on the diffusion model to locate a feasible point on the image manifold. ViT-H is a strong classifier, and its decision boundaries between unrelated classes appear to be more accurate than those of the other classifiers we examined. As a result, the diffusion model struggles to generate feasible targeted adversarial samples for ViT-H, and artifacts are introduced. These artifacts substantially degrade image quality and artificially inflate the attack success rate of the targeted ViT-H samples (as seen in Table 1). Importantly, these artifacts occur only in the ViT-H targeted setting, which is directly observable from the image-quality metrics in Table 1.

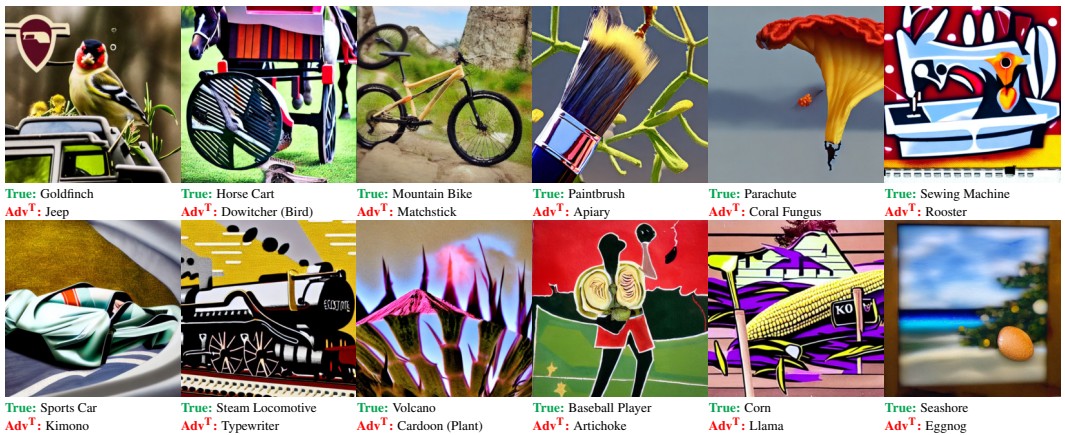

Figure 18: Low-quality adversarial samples generated by NatADiff with a ViT-H (Dosovitskiy et al., 2021) surrogate model under targeted attack settings. We report the true class and adversarial target for each image. Superscripts T and U indicate targeted and untargeted (similarity-based) attacks, respectively.

## L ADDITIONAL IMAGE QUALITY METRICS

We use NIQE (Mittal et al., 2013), BRISQUE (Mittal et al., 2012), and TReS (Golestaneh et al., 2022) to provide additional no-reference image quality evaluations of adversarial sampling methods. We assess the image quality of NCF (Yuan et al., 2022), DiffAttack (Chen et al., 2025), ACA (Chen et al., 2023b), adversarial classifier guidance (AdvClass) (Dai et al., 2024), and NatADiff across ResNet-50 (He et al., 2016), Inception-v3 (Szegedy et al., 2016), and ViT-H (Dosovitskiy et al., 2021) surrogate models. These metrics more closely align with human perception of image quality, but they do not address adherence to a target data distribution, i.e., how well samples "fit into" the ImageNet dataset.

Adversarial classifier guidance and DiffAttack frequently outperformed other methods on NIQE, BRISQUE, and TReS (see Table 9); however, as discussed in Section 5.2, this is likely because these methods apply constrained perturbations to source images and clean stable diffusion outputs, respectively. When considering only methods that make structural image alterations, we see that NatADiff outperforms NCF across all image metrics, and outperforms ACA on NIQE and BRISQUE, with similar TReS scores that slightly favour ACA. This supports the findings from the main paper that NatADiff is able to construct visually high-quality adversarial samples.

## M RESISTANCE TO ADVERSARIAL DEFENCES

It has previously been shown that perturbation-based adversarial attacks are sensitive to image transformations–such as rotations, crops, and translations–which can substantially reduce attack success rates (Guo et al., 2018). In addition to such transformation-based defences, purification approaches aim to remove adversarial noise prior to classification. One such method is *DiffPure* (Nie et al., 2022), which leverages a denoising diffusion model to project adversarial samples back

Table 9: **Image quality** of adversarial samples generated using ACA (Chen et al., 2023b), DiffAttack (Chen et al., 2025), adversarial classifier guidance (Dai et al., 2024), and NatADiff. **Bold** and underlined values highlight the best and second best scores for each surrogate model. Superscripts T and U denote targeted and untargeted attacks, respectively. Note that we report FID with respect to ImageNet-Val (FID-Val) and ImageNet-A (FID-A).

| Surrogate Model | Attack | Image Quality Metrics | | | | | |
|---|---|---|---|---|---|---|---|
| | | IS ($\uparrow$) | FID-Val ($\downarrow$) | FID-A ($\downarrow$) | NIQE ($\downarrow$) | BRISQUE ($\downarrow$) | TReS ($\uparrow$) |
| | Clean | 55.0 | 58.0 | 94.7 | 4.6 | 18.1 | 88.6 |
| RN-50 | NCF | 30.4 | 69.7 | 85.5 | 5.5 | 19.8 | 68.9 |
| | DiffAttack | 26.8 | 64.1 | **76.8** | 5.7 | 17.7 | **81.8** |
| | ACA | 23.9 | 65.0 | 77.9 | 6.8 | 24.4 | 80.8 |
| | AdvClass$^T$ | 38.3 | **48.9** | 92.4 | **4.3** | **11.9** | **81.8** |
| | AdvClass$^U$ | 38.5 | 50.2 | 92.7 | 4.6 | 12.2 | 81.3 |
| | NatADiff$^T$ | 26.0 | 66.5 | 77.3 | 4.8 | 12.3 | 76.0 |
| | NatADiff$^U$ | **43.2** | 51.4 | 95.9 | 4.8 | 12.0 | 77.9 |
| Inc-v3 | NCF | 31.7 | 69.1 | 83.0 | 4.7 | 19.0 | 76.3 |
| | DiffAttack | 33.2 | 63.7 | **78.2** | 5.8 | 18.0 | 81.2 |
| | ACA | 23.1 | 68.0 | 78.8 | 7.8 | 28.4 | 78.7 |
| | AdvClass$^T$ | 33.7 | 51.0 | 89.2 | **4.5** | 12.3 | **81.4** |
| | AdvClass$^U$ | 39.7 | **49.4** | 93.3 | **4.5** | 12.3 | 81.2 |
| | NatADiff$^T$ | 27.7 | 66.6 | **78.2** | 4.8 | 12.4 | 76.6 |
| | NatADiff$^U$ | **47.0** | 50.5 | 98.9 | 4.7 | **11.7** | 78.9 |
| ViT-H | NCF | **39.8** | 63.1 | 86.4 | 5.6 | 20.0 | 70.1 |
| | DiffAttack | 35.2 | 63.4 | **80.0** | 6.0 | 18.4 | **81.3** |
| | ACA | 25.5 | 64.2 | 80.9 | 7.4 | 25.5 | 79.2 |
| | AdvClass$^T$ | 38.9 | **48.5** | 95.2 | **4.2** | 14.8 | 77.7 |
| | AdvClass$^U$ | 39.2 | **48.5** | 98.8 | 4.3 | **13.5** | 79.8 |
| | NatADiff$^T$ | 15.3 | 88.0 | 93.5 | 4.9 | 20.9 | 74.7 |
| | NatADiff$^U$ | 31.9 | 53.9 | 96.2 | 4.6 | 14.8 | 78.8 |

onto the natural image manifold. Given an adversarial image, $\tilde{x}_0$, the forward diffusion process, $p(x_t|x_0 = \tilde{x}_0)$, is applied for $t \ll T$ (recall that $T$ is the termination time of the forward process), introducing Gaussian noise without fully destroying the original signal. The sample is then passed through the reverse-time diffusion process (Anderson, 1982) or flow ODE (Song et al., 2021b) to recover a purified image. Nie et al. (2022) empirically demonstrated and theoretically proved that this procedure effectively removes perturbation-based adversarial noise, enabling the reverse process to reconstruct a "clean" version of the image. Intuitively, the noise injected during forward diffusion overwhelms the adversarial signal, allowing the diffusion model to project the corrupted sample back onto the natural image manifold.

We evaluate the robustness of standard image transformations and DiffPure against adversarial samples generated by PGD (Madry et al., 2018), AutoAttack (Croce & Hein, 2020), NCF (Yuan et al., 2022), DiffAttack (Chen et al., 2025), ACA (Chen et al., 2023b), adversarial classifier guidance (AdvClass) (Dai et al., 2024), and NatADiff. We use a ResNet-50 (He et al., 2016) surrogate model and test NatADiff and adversarial classifier under both targeted and untargeted modes. Defences are applied as a pre-processing step to classification; for image transformations, we average classification probabilities over augmented views obtained via cropping, rotation, and grayscale conversion. To quantify defence effectiveness, we report attack success rate (ASR).

Our results show that transform-purification did not meaningfully reduce the efficacy of NatADiff, though it successfully defended against PGD attacks (see Table 10). In contrast, DiffPure provided a much stronger defence to most attacks, reducing NatADiff's average ASR by 7.9%. However, DiffPure occasionally degraded overall classifier accuracy, leading to increased ASR for non-surrogate classifiers. This likely occurred when the reverse diffusion process failed to recover the original image, rendering classification unreliable. Compared to other attacks, NatADiff still exhibited superior white-box performance and achieved either best or second-best transferability across all victim

classifiers. Interestingly, NCF showed a significant increase in transferability under the DiffPure defence. We hypothesize that this stems from NCF's color-based attacks pushing samples into low-probability regions of the manifold during the forward diffusion process. This increases the likelihood that DiffPure fails to recover the original image in the reverse process, thereby degrading classifier accuracy.

Consistent with findings in the main paper, NatADiff achieved best or near-best performance under both transformation and DiffPure defences. Given that natural adversarial samples are known to bypass perturbation-based defences and image transformations (Agarwal et al., 2022), these results further support our claim that NatADiff generates adversarial examples that are more semantically aligned with naturally occurring test-time errors.

Table 10: **Attack success rate** of **transform** and **DiffPure-purified** adversarial samples generated by PGD (Madry et al., 2018), AutoAttack (Croce & Hein, 2020), NCF (Yuan et al., 2022), DiffAttack (Chen et al., 2025), ACA (Chen et al., 2023b), adversarial classifier guidance (AdvClass) (Dai et al., 2024), and NatADiff. Samples are generated using a ResNet-50 (He et al., 2016) surrogate model. **Bold** and underlined values highlight the best and second best scores for each purification method. Superscripts T and U denote targeted and untargeted attacks, respectively. White-box ASR (same surrogate and victim model) is denoted with an $*$.

| Purification Method | Attack | Victim Model ASR (%) | | | | | | | | | Average ASR |
| | | CNNs | | | | | Transformers | | | | |
| | | RN-50 | Inc-v3 | RN-152 | AdvRes | AdvInc | ViT-H | Max-ViT | Swin-B | DeIT | |
| | Clean | 5.3 | 7.6 | 2.9 | 3.0 | 5.8 | 10.9 | 3.8 | 4.5 | 7.4 | 5.7 |
| None | PGD | 99.4* | 11.8 | 5.2 | 4.9 | 8.1 | 10.5 | 4.4 | 5.5 | 8.2 | 17.6 |
| | AA | **100***| 13.3 | 10.0 | 3.9 | 8.8 | 10.5 | 5.4 | 5.6 | 8.0 | 18.4 |
| | NCF | 74.8* | 33.4 | 37.3 | 28.2 | 31.2 | 17.2 | 24.0 | 31.7 | 37.2 | 35.0 |
| | DiffAttack | 92.5* | 47.1 | 52.5 | 35.3 | 43.3 | 28.4 | 44.6 | 42.4 | 38.9 | 47.2 |
| | ACA | 78.8* | 53.3 | 52.7 | 49.8 | 53.1 | 41.8 | 46.4 | 49.3 | 50.6 | 52.9 |
| | AdvClass$^T$ | 99.6* | 35.0 | 32.1 | 31.4 | 33.5 | 25.8 | 30.0 | 30.8 | 32.8 | 39.0 |
| | AdvClass$^U$ | 99.9* | 42.5 | 44.3 | 38.7 | 41.1 | 29.7 | 37.6 | 38.4 | 39.1 | 45.7 |
| | NatADiff$^T$ | 96.9* | 60.1 | 56.5 | 55.3 | 58.9 | 36.8 | 45.3 | 49.0 | 52.3 | 56.8 |
| | NatADiff$^U$ | 99.3* | **68.3** | **72.1** | **65.3** | **66.8** | **45.3** | **64.1** | **65.2** | **67.0** | **68.2** |
| Transform | PGD | 14.5* | 12.1 | 4.8 | 4.7 | 7.4 | 9.2 | 3.5 | 5.2 | 7.3 | 7.6 |
| | AA | 79.4* | 12.4 | 7.9 | 3.3 | 9.1 | 10.2 | 3.4 | 5.8 | 7.3 | 15.4 |
| | NCF | 60.2* | 35.1 | 39.6 | 28.4 | 31.2 | 16.7 | 27.1 | 33.7 | 37.7 | 34.4 |
| | DiffAttack | 73.9* | 48.6 | 50.1 | 39.9 | 45.8 | 28.6 | 43.4 | 46.7 | 39.0 | 46.2 |
| | ACA | 64.2* | 54.8 | 52.2 | 50.4 | 56.4 | 40.8 | 47.1 | 51.9 | 51.0 | 52.1 |
| | AdvClass$^T$ | 35.2* | 33.4 | 30.4 | 29.7 | 31.5 | 25.5 | 28.9 | 31.4 | 32.1 | 30.9 |
| | AdvClass$^U$ | 65.8* | 39.8 | 40.8 | 38.3 | 40.0 | 29.1 | 36.3 | 37.0 | 38.4 | 40.6 |
| | NatADiff$^T$ | 85.7* | 59.8 | 55.6 | 55.0 | 56.8 | 36.1 | 46.7 | 48.6 | 52.4 | 55.2 |
| | NatADiff$^U$ | **96.6*** | **68.7** | **73.3** | **67.3** | **68.3** | **45.0** | **65.4** | **66.8** | **70.3** | **69.1** |
| DiffPure | PGD | 21.9* | 30.8 | 20.3 | 23.3 | 26.5 | 21.3 | 16.7 | 19.4 | 20.6 | 22.3 |
| | AA | 23.5* | 32.5 | 19.8 | 22.7 | 28.4 | 21.7 | 18.9 | 21.2 | 23.6 | 23.6 |
| | NCF | 68.8* | 59.9 | 60.4 | 53.9 | 55.3 | 46.1 | 57.5 | **62.3** | 59.7 | 58.2 |
| | DiffAttack | 45.9* | 44.0 | 39.3 | 37.3 | 43.5 | 38.4 | 37.4 | 41.8 | 38.7 | 40.7 |
| | ACA | 60.8* | **63.1** | 55.3 | 57.3 | 60.1 | **50.4** | 55.1 | 56.7 | 56.5 | 57.3 |
| | AdvClass$^T$ | 35.0* | 37.8 | 34.2 | 35.2 | 37.2 | 30.3 | 34.3 | 34.9 | 36.7 | 35.1 |
| | AdvClass$^U$ | 42.4* | 42.6 | 39.8 | 39.8 | 41.9 | 34.3 | 38.8 | 40.3 | 41.3 | 40.1 |
| | NatADiff$^T$ | 56.3* | 56.6 | 52.2 | 53.1 | 54.8 | 42.3 | 49.1 | 51.0 | 53.4 | 52.1 |
| | NatADiff$^U$ | **71.3*** | 62.2 | **61.2** | **61.0** | **61.3** | 47.8 | **58.5** | 60.2 | **61.2** | **60.5** |

# N    RUNTIME COMPARISON

We provide a runtime comparison of PGD (Madry et al., 2018), AutoAttack (AA) (Croce & Hein, 2020), NCF (Yuan et al., 2022), DiffAttack (Chen et al., 2025), ACA (Chen et al., 2023b), adversarial classifier guidance (AdvClass) (Dai et al., 2024) and NatADiff. It is clear that the generative approach of NatADiff requires substantially greater runtime; however, this cost yields stronger adversarial samples that transfer more effectively across classifiers (see Table 1), and that are more resistant to adversarial purification (see Appendix M). We argue that the trade-off between runtime and state-of-the-art adversarial strength makes NatADiff a compelling attack strategy despite its slower speed.

Table 11: **Time comparison** of adversarial attack methods.

| Attack | Average Runtime per Sample (seconds) |
|---|---|
| PGD | 0.3 |
| AutoAttack | 0.7 |
| NCF | 6.9 |
| DiffAttack | 14.2 |
| ACA | 96.8 |
| AdvClass | 13.5 |
| NatADiff | 103.1 |

## O  USER STUDY

NatADiff is a generative attack method designed to produce adversarial samples that lie near the decision boundary between the true and adversarial classes. However, as discussed in Appendix G.2, the generated image may occasionally fully manifest the adversarial class, resulting in the loss of the intended "true" class. To mitigate this, we conservatively set the adversarial boundary guidance strength to $\mu = 0.2$. We conduct a human study to verify that this choice preserves the human-perceived class.

Accurate classification of ImageNet classes typically requires trained human annotators (Russakovsky et al., 2015; Shankar et al., 2020). Previous work has found that untrained annotators often exhibit significant class unawareness, particularly for fine-grained categories such as dog breeds (Russakovsky et al., 2015; Shankar et al., 2020). To address these limitations, we restrict our study to a curated subset of 148 easily recognisable ImageNet classes[1].

We present 22 human participants with 60 randomly selected NatADiff samples generated using a ResNet-50 surrogate classifier under both targeted and untargeted attack settings. Each participant is asked whether the image contains the true class, the adversarial class, or neither. Participants reported seeing the true class in $91\%$ of cases, the adversarial class in $7\%$, and neither class in $2\%$. NatADiff's label-flip rate of $9\%$ is comparable to the $10\%$ flip rate reported by Dai et al. (2024) for adversarial classifier guidance on the MNIST dataset (Deng, 2012). These results indicate that NatADiff reliably produces images that appear to belong to the true class for human observers, even while fooling classifiers with high probability. Moreover, in practical attack settings, an adversary could manually inspect generated images prior to deployment, further mitigating the impact of occasional class-flipped samples.

---

[1]Selected classes: 1, 9, 22, 47, 105, 130, 151, 152, 153, 154, 155, 156, 158, 159, 160, 161, 162, 163, 285, 286, 287, 288, 289, 290, 291, 292, 407, 425, 440, 448, 449, 462, 465, 468, 469, 470, 471, 472, 486, 493, 495, 496, 497, 502, 503, 505, 510, 514, 515, 516, 517, 520, 525, 526, 527, 532, 537, 555, 580, 607, 609, 663, 670, 671, 672, 701, 719, 734, 780, 787, 820, 909, 937, 963, 975, 999, 972, 973, 569, 933, 452, 958, 985, 900, 991, 400, 635, 920, 103, 482, 392, 843, 833, 259, 614, 757, 490, 715, 834, 222, 509, 374, 815, 927, 545, 531, 21, 303, 282, 132, 536, 913, 741, 624, 269, 547, 313, 640, 182, 849, 954, 524, 109, 657, 889, 884, 961, 94, 797, 817, 653, 134, 978, 803, 668, 436, 581, 519, 554, 942, 806, 410, 144, 693, 523, 26, 403, and 846.

# P  USEFUL DENOISING DIFFUSION RESULTS

This section outlines results necessary for working with denoising diffusion models. Citations of original authors are provided where applicable.

## P.1  CONDITIONAL FORWARD DISTRIBUTION

The following theorem describes the conditional forward distribution of the denoising diffusion model when conditioned on an arbitrary time $\tau$.

**Theorem P.1** (Conditional Forward Distribution for Denosing Diffusion). *Let $\boldsymbol{x}_t \in \mathbb{R}^m$, $f(t) : \mathbb{R} \to \mathbb{R}$ and $g(t) : \mathbb{R} \to \mathbb{R}$ be continuous functions of $t$, and $\cdot\, d\boldsymbol{B}_t$ denote an Itô integral with respect to the standard multi-dimensional Brownian motion process. Then the denoising diffusion model with forward process,*

$$d\boldsymbol{x}_t = f(t)\boldsymbol{x}_t dt + g(t) \cdot d\boldsymbol{B}_t, \tag{13}$$

*admits a conditional forward distribution of*

$$X_t | X_\tau \sim \mathcal{N}\left(\alpha(\tau,t)x_\tau,\ \beta(\tau,t)^2 I^{(m\times m)}\right)\ \forall\, t > \tau,$$

*where $\alpha(\tau,t) = \exp\left(\int_\tau^t f(u)du\right)$, $\beta(\tau,t)^2 = \alpha(\tau,t)^2 \int_\tau^t \frac{g(u)^2}{\alpha(\tau,u)^2}du$, and $I^{(m\times m)}$ is the m-dimensional identity matrix. Additionally, it is understood that with slight abuse of notation $\alpha(t) = \alpha(0,t)$ and $\beta(t) = \alpha(0,t)$*

*Proof.* The diffusion in (13) has Stratonovich representation (see (Pavliotis, 2014a) for a treatment of Itô and Stratonovich SDE formulations),

$$d\boldsymbol{x}_t = f(t)\boldsymbol{x}_t dt + g(t) \circ d\boldsymbol{B}_t,$$

where $\circ\, d\boldsymbol{B}_t$ denotes a Stratonovich integral with respect to the standard multi-dimensional Brownian motion process. Thus, the SDE can be solved in the usual manner:

$$d\boldsymbol{x}_t = f(t)\boldsymbol{x}_t dt + g(t) \circ d\boldsymbol{B}_t$$

$$\frac{d\boldsymbol{x}_t}{dt} = f(t)\boldsymbol{x}_t + g(t) \circ \frac{d\boldsymbol{B}_t}{dt}$$

$$\implies \left(e^{-\int_\tau^t f(u)du}\right)\frac{d\boldsymbol{x}_t}{dt} = \left(e^{-\int_\tau^t f(u)du}\right)f(t)\boldsymbol{x}_t$$

$$+ \left(e^{-\int_\tau^t f(u)du}\right)g(t) \circ \frac{d\boldsymbol{B}_t}{dt} \quad \forall\, t > \tau$$

$$\implies \left(e^{-\int_\tau^t f(u)du}\right)g(t) \circ \frac{d\boldsymbol{B}_t}{dt} = \left(e^{-\int_\tau^t f(u)du}\right)\frac{d\boldsymbol{x}_t}{dt} - \left(e^{-\int_\tau^t f(u)du}\right)f(t)\boldsymbol{x}_t$$

$$\implies \int_\tau^t \left(e^{-\int_\tau^v f(u)du}\right)g(v) \circ \frac{d\boldsymbol{B}_v}{dv}dv = \int_\tau^t \left(e^{-\int_\tau^v f(u)du}\right)\frac{dx_v}{dv} - \left(e^{-\int_\tau^v f(u)du}\right)f(v)x_v dv$$

$$\int_\tau^t \left(e^{-\int_\tau^v f(u)du}\right)g(v) \circ d\boldsymbol{B}_v = \left[x_v\left(e^{-\int_\tau^v f(u)du}\right)\right]\Big|_{v=\tau}^{v=t}$$

$$\int_\tau^t \left(e^{-\int_\tau^v f(u)du}\right)g(v) \circ d\boldsymbol{B}_v = \boldsymbol{x}_t\left(e^{-\int_\tau^t f(u)du}\right) - x_\tau$$

$$\int_\tau^t \frac{g(v)}{\alpha(\tau,v)} \circ d\boldsymbol{B}_v = \frac{\boldsymbol{x}_t}{\alpha(\tau,t)} - x_\tau$$

$$\implies \boldsymbol{x}_t = \alpha(\tau,t)x_\tau + \alpha(\tau,t)\int_\tau^t \frac{g(v)}{\alpha(\tau,v)} \circ d\boldsymbol{B}_v. \tag{14}$$

By rewriting (14) in its Itô representation we have

$$\boldsymbol{x}_t = \alpha(\tau,t)x_\tau + \alpha(\tau,t)\int_\tau^t \frac{g(v)}{\alpha(\tau,v)} \cdot d\boldsymbol{B}_v,$$

and as $\int_\tau^t \frac{g(u)}{\alpha(\tau,u)} \cdot d\boldsymbol{B}_u \sim \mathcal{N}\left(0, \int_\tau^t \frac{g(u)^2}{\alpha(\tau,u)^2} du \cdot I^{(m \times m)}\right)$, it follows that

$$\boldsymbol{x}_t | \boldsymbol{x}_\tau \sim \mathcal{N}\left(\alpha(\tau,t) x_\tau, \ \alpha(\tau,t)^2 \int_\tau^t \frac{g(u)^2}{\alpha(\tau,u)^2} du \cdot I^{(m \times m)}\right).$$

$\square$

## P.2 CONDITIONAL FORWARD ALTERNATE PARAMETERISATION

When implementing time-travel sampling (Lugmayr et al., 2022) we require access to the conditional forward distribution, $p(\boldsymbol{x}_t | \boldsymbol{x}_\tau)$. However, it is frequently the case that diffusion schemes are formulated with respect to the full forward distribution, $p(\boldsymbol{x}_t | \boldsymbol{x}_0)$, and some proposed method of sampling the reverse-time diffusion (Anderson, 1982), or solving the flow ODE (Song et al., 2021b). Thus, we provide a simple result to derive the conditional forward distribution, $p(\boldsymbol{x}_t | \boldsymbol{x}_\tau)$, from the parameterisation of the full forward, $p(\boldsymbol{x}_t | \boldsymbol{x}_0)$.

**Lemma P.2** (Conditional Forward Alternate Parameterisation). *Given the diffusion formulation in Theorem P.1, then the conditional forward distribution can alternately be expressed as*

$$X_t | X_\tau \sim \mathcal{N}\left(a x_\tau, \ b^2 I^{(m \times m)}\right) \ \forall \, t > \tau,$$

*where*

$$X_t | X_0 \sim \mathcal{N}\left(\alpha(0,t) x_0, \ \beta(0,t)^2 I^{(m \times m)}\right) \ \forall \, t > 0,$$

$a = \frac{\alpha(0,t)}{\alpha(0,\tau)}$, $b^2 = \beta(0,t)^2 - (a\beta(0,\tau))^2$, , *and* $I^{(m \times m)}$ *is the* $m$-*dimensional identity matrix. Additionally, it is understood that with slight abuse of notation* $\alpha(t) = \alpha(0,t)$ *and* $\beta(t) = \alpha(0,t)$

*Proof.* We need to show that $a = \alpha(\tau,t) = \exp\left(\int_\tau^t f(u)du\right)$ and $b^2 = \beta(\tau,t)^2 = \alpha(\tau,t)^2 \int_\tau^t \frac{g(u)^2}{\alpha(\tau,u)^2} du$ as per Theorem P.1. It follows that

$$
\begin{aligned}
a &= \frac{\alpha(0,t)}{\alpha(0,\tau)} \\
&= \frac{\exp\left(\int_0^t f(u)du\right)}{\exp\left(\int_0^\tau f(u)du\right)} \\
&= \exp\left(\int_0^t f(u)du - \int_0^\tau f(u)du\right) \\
&= \exp\left(\int_\tau^t f(u)du\right) \\
&= \alpha(\tau,t),
\end{aligned}
\tag{15}
$$

and

$$
\begin{aligned}
b^2 &= \beta(0,t)^2 - (a\beta(0,\tau))^2 \\
&= \beta(0,t)^2 - a^2\beta(0,\tau)^2 \\
&= \alpha(0,t)^2 \int_0^t \frac{g(u)^2}{\alpha(0,u)^2}du - \frac{\alpha(0,t)^2}{\alpha(0,\tau)^2}\alpha(0,\tau)^2 \int_0^\tau \frac{g(u)^2}{\alpha(0,u)^2}du \\
&= \alpha(0,t)^2 \left[ \int_0^t \frac{g(u)^2}{\alpha(0,u)^2}du - \int_0^\tau \frac{g(u)^2}{\alpha(0,u)^2}du \right] \\
&= \alpha(0,t)^2 \int_\tau^t \frac{g(u)^2}{\alpha(0,u)^2}du \\
&= \alpha(0,t)^2 \int_\tau^t \frac{g(u)^2}{\alpha(0,\tau)^2\alpha(\tau,u)^2}du \quad \text{as } \alpha(0,u) = \alpha(0,\tau)\alpha(\tau,u) \text{ by (15)} \\
&= \frac{\alpha(0,t)^2}{\alpha(0,\tau)^2} \int_\tau^t \frac{g(u)^2}{\alpha(\tau,u)^2}du \\
&= \alpha(\tau,t)^2 \int_\tau^t \frac{g(u)^2}{\alpha(\tau,u)^2}du \\
&= \beta(\tau,t)^2.
\end{aligned}
$$

$\square$

## P.3 Score-model link

The following Score-Model Link theorem is based on Karras et al.'s (Karras et al., 2022) argument. However, we provide a minor extension by conditioning on measurable sets taken from the sigma-algebra of an auxiliary random variable, $Y$. For additional treatments see (Karras et al., 2022; Hyvärinen, 2005; Vincent, 2011).

**Theorem P.3** (Score-Model Link). *Let $\boldsymbol{x}_t \in \mathbb{R}^m$, $f(t) : \mathbb{R} \to \mathbb{R}$ and $g(t) : \mathbb{R} \to \mathbb{R}$ be continuous functions of $t$, and $\cdot d\boldsymbol{B}_t$ denote an Itô integral with respect to the standard multi-dimensional Brownian motion process. Suppose that $\boldsymbol{x}_t$ evolves according to the diffusion,*

$$
d\boldsymbol{x}_t = f(t)\boldsymbol{x}_t dt + g(t) \cdot d\boldsymbol{B}_t, \tag{16}
$$

*with observed initial data distribution, $p_{data}(\boldsymbol{x}_0|y \in \xi)$, where $\xi$ is taken to be an arbitrary element of the sigma-algebra, $\mathcal{Y}$, associated with the random variable[2] $Y$, i.e., $\xi \in \mathcal{Y}$.*

*Define*

$$
\hat{\boldsymbol{x}}_0(\boldsymbol{x}_t, t, \xi) = \frac{\boldsymbol{x}_t - \beta(t)\boldsymbol{\epsilon}_\theta(\boldsymbol{x}_t, t, \xi)}{\alpha(t)}, \tag{17}
$$

*where $\alpha(t) = \exp\left(\int_0^t f(u)du\right)$, $\beta(t)^2 = \alpha(t)^2 \int_0^t \frac{g(u)^2}{\alpha(u)^2}du$, and $\boldsymbol{\epsilon}_\theta : \mathbb{R}^m \times \mathbb{R} \times \mathcal{Y} \to \mathbb{R}^m$ is a model parameterised by $\theta \in \Theta$ with sufficient capacity such that the Universal Approximation Theorem (Cybenko, 1989; Hornik, 1991) holds for all $\xi \in \mathcal{Y}$. Then if $\beta(t)^2 > 0 \ \forall \, t \in (0, T]$, the following statements are true:*

*1.*

$$
\nabla_{\boldsymbol{x}_t} \log(p(\boldsymbol{x}_t|y \in \xi)) = -\frac{1}{\beta(t)}\boldsymbol{\epsilon}_{\theta^\star}(\boldsymbol{x}_t, t, \xi) \quad \forall \, t \in (0, T], \ \boldsymbol{x}_t \in \mathbb{R}^m, \ \xi \in \mathcal{Y}; \tag{18}
$$

*2.*

$$
\boldsymbol{\epsilon}_{\theta^\star}(\boldsymbol{x}_t, t, \xi) = \frac{\mathbb{E}_{\boldsymbol{x}_0 \sim p_{data}(\boldsymbol{x}_0|y \in \xi)}\left[\frac{1}{\beta(t)}(\boldsymbol{x}_t - \alpha(t)\boldsymbol{x}_0)p(\boldsymbol{x}_t|\boldsymbol{x}_0)\right]}{p(\boldsymbol{x}_t|y \in \xi)} \quad \forall \, t \in (0, T], \tag{19}
$$

---

[2]Note that this is a slight abuse of notation. We are assuming that $(\Omega_Y, \mathcal{Y}, P)$ is a probability space and $Y : \Omega_Y \to \Omega_Y$ a random variable such that $Y(\omega) = \omega \ \forall \, \omega \in \Omega_Y$. That is to say, we do not need to take the pre-image when crafting probability statements.

$\boldsymbol{x}_t \in \mathbb{R}^m$, $\xi \in \mathcal{Y}$; where

$$\theta^\star \triangleq \arg\min_\theta \mathbb{E}_{\boldsymbol{x}_0, \boldsymbol{x}_t, t \sim p(\boldsymbol{x}_0, \boldsymbol{x}_t, t | y \in \xi)} \left[ \|\boldsymbol{x}_0 - \hat{\boldsymbol{x}}_0(\boldsymbol{x}_t, t, \xi)\|_2^2 \right], \tag{20}$$

$p(\boldsymbol{x}_0, \boldsymbol{x}_t, t | y \in \xi) = p(\boldsymbol{x}_t | \boldsymbol{x}_0) p_{data}(\boldsymbol{x}_0 | y \in \xi) p(t)$, and $p(t) \triangleq \frac{1}{T}$.

$$\tag{21}$$

*Proof.* Let $\{\boldsymbol{x}_0^{(1)}, \boldsymbol{x}_0^{(2)}, \dots, \boldsymbol{x}_0^{(N)}\}$ and $\{y^{(1)}, y^{(2)}, \dots, y^{(N)}\}$ denote observed values of $\boldsymbol{x}_0$ and $Y$. Then the data density function is given by:

$$p_{\text{data}}(\boldsymbol{x}_0 | y \in \xi) = \frac{\sum_{i=1}^N \delta(\boldsymbol{x}_0 - \boldsymbol{x}_0^{(i)}) \mathbb{1}_{\{y^{(i)} \in \xi\}}}{\sum_{i=1}^N \mathbb{1}_{\{y^{(i)} \in \xi\}}}, \tag{22}$$

where $\delta(\cdot)$ and $\mathbb{1}_{\{\cdot\}}$ denote the Dirac delta and indicator functions, respectively. The forward diffusion process in (16) is independent of $Y$, and thus,

$$\begin{aligned} p(\boldsymbol{x}_t | y \in \xi) &= \int_\Omega p(\boldsymbol{x}_t | \boldsymbol{x}_0) p_{\text{data}}(\boldsymbol{x}_0 | y \in \xi) d\boldsymbol{x}_0 \\ &= \frac{\sum_{i=1}^N p(\boldsymbol{x}_t | \boldsymbol{x}_0^{(i)}) \mathbb{1}_{\{y^{(i)} \in \xi\}}}{\sum_{i=1}^N \mathbb{1}_{\{y^{(i)} \in \xi\}}}. \end{aligned} \tag{23}$$

Substituting (23) into the LHS of (18),

$$\begin{aligned} \nabla_{\boldsymbol{x}_t} \log(p(\boldsymbol{x}_t | y \in \xi)) &= \nabla_{\boldsymbol{x}_t} \log \left( \frac{\sum_{i=1}^N p(\boldsymbol{x}_t | \boldsymbol{x}_0^{(i)}) \mathbb{1}_{\{y^{(i)} \in \xi\}}}{\sum_{i=1}^N \mathbb{1}_{\{y^{(i)} \in \xi\}}} \right) \\ &= \nabla_{\boldsymbol{x}_t} \log \left( \sum_{i=1}^N p(\boldsymbol{x}_t | \boldsymbol{x}_0^{(i)}) \mathbb{1}_{\{y^{(i)} \in \xi\}} \right) \\ &= \frac{\sum_{i=1}^N \nabla_{\boldsymbol{x}_t} p(\boldsymbol{x}_t | \boldsymbol{x}_0^{(i)}) \mathbb{1}_{\{y^{(i)} \in \xi\}}}{\sum_{i=1}^N p(\boldsymbol{x}_t | \boldsymbol{x}_0^{(i)}) \mathbb{1}_{\{y^{(i)} \in \xi\}}}. \end{aligned}$$

By Theorem P.1, $p(\boldsymbol{x}_t | \boldsymbol{x}_0) = \mathcal{N}\left(\alpha(t)\boldsymbol{x}_0, \beta(t)^2 I^{(m \times m)}\right) \implies \nabla_{\boldsymbol{x}_t} p(\boldsymbol{x}_t | \boldsymbol{x}_0) = -\frac{1}{\beta(t)^2}(\boldsymbol{x}_t - \alpha(t)\boldsymbol{x}_0) p(\boldsymbol{x}_t | \boldsymbol{x}_0)$, thus,

$$\nabla_{\boldsymbol{x}_t} \log(p(\boldsymbol{x}_t | y \in \xi)) = -\frac{1}{\beta(t)^2} \frac{\sum_{i=1}^N (\boldsymbol{x}_t - \alpha(t)\boldsymbol{x}_0^{(i)}) p(\boldsymbol{x}_t | \boldsymbol{x}_0^{(i)}) \mathbb{1}_{\{y^{(i)} \in \xi\}}}{\sum_{i=1}^N p(\boldsymbol{x}_t | \boldsymbol{x}_0^{(i)}) \mathbb{1}_{\{y^{(i)} \in \xi\}}}. \tag{24}$$

Now we consider the optimisation problem in (20).

$$\mathcal{L}(\xi;\theta) = \mathbb{E}_{\boldsymbol{x}_0,\boldsymbol{x}_t,t\sim p(\boldsymbol{x}_0,\boldsymbol{x}_t,t|y\in\xi)}\left[\|\boldsymbol{x}_0 - \hat{\boldsymbol{x}}_0(\boldsymbol{x}_t,t,\xi)\|_2^2\right]$$

$$= \mathbb{E}_{t\sim p(t)}\left[\mathbb{E}_{\boldsymbol{x}_0\sim p(\boldsymbol{x}_0|y\in\xi)}\left[\mathbb{E}_{\boldsymbol{x}_t\sim p(\boldsymbol{x}_t|\boldsymbol{x}_0)}\left[\|\boldsymbol{x}_0 - \hat{\boldsymbol{x}}_0(\boldsymbol{x}_t,t,\xi)\|_2^2\right]\right]\right]$$

$$= \mathbb{E}_{t\sim p(t)}\left[\mathbb{E}_{\boldsymbol{x}_0\sim p(\boldsymbol{x}_0|y\in\xi)}\left[\int_{\Omega_{\boldsymbol{x}_t}}\|\boldsymbol{x}_0 - \hat{\boldsymbol{x}}_0(\boldsymbol{x}_t,t,\xi)\|_2^2\ p(\boldsymbol{x}_t|\boldsymbol{x}_0)d\boldsymbol{x}_t\right]\right]$$

$$= \mathbb{E}_{t\sim p(t)}\left[\int_{\Omega_{\boldsymbol{x}_0}}\int_{\Omega_{\boldsymbol{x}_t}}\|\boldsymbol{x}_0 - \hat{\boldsymbol{x}}_0(\boldsymbol{x}_t,t,\xi)\|_2^2\ p(\boldsymbol{x}_t|\boldsymbol{x}_0)d\boldsymbol{x}_t\ p(\boldsymbol{x}_0|y\in\xi)d\boldsymbol{x}_0\right]$$

$$= \mathbb{E}_{t\sim p(t)}\left[\int_{\Omega_{\boldsymbol{x}_0}}\int_{\Omega_{\boldsymbol{x}_t}}\|\boldsymbol{x}_0 - \hat{\boldsymbol{x}}_0(\boldsymbol{x}_t,t,\xi)\|_2^2\ p(\boldsymbol{x}_t|\boldsymbol{x}_0)d\boldsymbol{x}_t\left(\frac{\sum_{i=1}^N \delta(\boldsymbol{x}_0 - \boldsymbol{x}_0^{(i)})\mathbb{1}_{\{y^{(i)}\in\xi\}}}{\sum_{i=1}^N \mathbb{1}_{\{y^{(i)}\in\xi\}}}\right)d\boldsymbol{x}_0\right]$$

$$= \mathbb{E}_{t\sim p(t)}\left[\frac{1}{\sum_{i=1}^N \mathbb{1}_{\{y^{(i)}\in\xi\}}}\sum_{i=1}^N\int_{\Omega_{\boldsymbol{x}_t}}\left\|\boldsymbol{x}_0^{(i)} - \hat{\boldsymbol{x}}_0(\boldsymbol{x}_t,t,\xi)\right\|_2^2\ p(\boldsymbol{x}_t|\boldsymbol{x}_0^{(i)})d\boldsymbol{x}_t\ \mathbb{1}_{\{y^{(i)}\in\xi\}}\right]$$

$$= \mathbb{E}_{t\sim p(t)}\left[\frac{1}{\sum_{i=1}^N \mathbb{1}_{\{y^{(i)}\in\xi\}}}\int_{\Omega_{\boldsymbol{x}_t}}\sum_{i=1}^N p(\boldsymbol{x}_t|\boldsymbol{x}_0^{(i)})\mathbb{1}_{\{y^{(i)}\in\xi\}}\left\|\boldsymbol{x}_0^{(i)} - \hat{\boldsymbol{x}}_0(\boldsymbol{x}_t,t,\xi)\right\|_2^2 d\boldsymbol{x}_t\right]$$

$$= \frac{1}{T}\frac{1}{\sum_{i=1}^N \mathbb{1}_{\{y^{(i)}\in\xi\}}}\int_0^T\int_{\Omega_{\boldsymbol{x}_t}}\sum_{i=1}^N p(\boldsymbol{x}_t|\boldsymbol{x}_0^{(i)})\mathbb{1}_{\{y^{(i)}\in\xi\}}\left\|\boldsymbol{x}_0^{(i)} - \hat{\boldsymbol{x}}_0(\boldsymbol{x}_t,t,\xi)\right\|_2^2 d\boldsymbol{x}_t dt$$

$$= \frac{1}{T}\frac{1}{\sum_{i=1}^N \mathbb{1}_{\{y^{(i)}\in\xi\}}}\int_0^T\int_{\Omega_{\boldsymbol{x}_t}}\underbrace{\sum_{i=1}^N p(\boldsymbol{x}_t|\boldsymbol{x}_0^{(i)})\mathbb{1}_{\{y^{(i)}\in\xi\}}\left\|\boldsymbol{x}_0^{(i)} - \frac{\boldsymbol{x}_t - \beta(t)\boldsymbol{\epsilon}_\theta(\boldsymbol{x}_t,t,\xi)}{\alpha(t)}\right\|_2^2}_{\ell(\boldsymbol{x}_t,t,\xi;\theta)} d\boldsymbol{x}_t dt$$

(25)

To minimise (25) with respect to $\theta$, it suffices to find $\theta$ such that $\ell(\boldsymbol{x}_t,t,\xi;\theta)$ is minimised for each combination of $\boldsymbol{x}_t$ and $t$. That is to say, we find the optimal value of $\boldsymbol{\epsilon}_\theta(\boldsymbol{x}_t,t,\xi)$ for each combination of $\boldsymbol{x}_t$ and $t$. Furthermore, $\ell(\boldsymbol{x}_t,t,\xi;\theta)$ constitutes a convex optimisation problem with respect to $\boldsymbol{\epsilon}_\theta(\boldsymbol{x}_t,t,\xi)$. Thus,

$$\frac{\partial\ell}{\partial\boldsymbol{\epsilon}_\theta} = 0 = \sum_{i=1}^N \frac{2\beta(t)}{\alpha(t)}p(\boldsymbol{x}_t|\boldsymbol{x}_0^{(i)})\mathbb{1}_{\{y^{(i)}\in\xi\}}\left(\boldsymbol{x}_0^{(i)} - \frac{\boldsymbol{x}_t - \beta(t)\boldsymbol{\epsilon}_\theta(\boldsymbol{x}_t,t,\xi)}{\alpha(t)}\right)$$

$$= \sum_{i=1}^N p(\boldsymbol{x}_t|\boldsymbol{x}_0^{(i)})\mathbb{1}_{\{y^{(i)}\in\xi\}}\left(\alpha(t)\boldsymbol{x}_0^{(i)} - \boldsymbol{x}_t + \beta(t)\boldsymbol{\epsilon}_\theta(\boldsymbol{x}_t,t,\xi)\right)$$

$$\implies \sum_{i=1}^N(\boldsymbol{x}_t - \alpha(t)\boldsymbol{x}_0^{(i)})p(\boldsymbol{x}_t|\boldsymbol{x}_0^{(i)})\mathbb{1}_{\{y^{(i)}\in\xi\}} = \sum_{i=1}^N \beta(t)\boldsymbol{\epsilon}_\theta(\boldsymbol{x}_t,t,\xi)p(\boldsymbol{x}_t|\boldsymbol{x}_0^{(i)})\mathbb{1}_{\{y^{(i)}\in\xi\}}$$

$$\implies \boldsymbol{\epsilon}_\theta^\star(\boldsymbol{x}_t,t,\xi) = \frac{1}{\beta(t)}\frac{\sum_{i=1}^N(\boldsymbol{x}_t - \alpha(t)\boldsymbol{x}_0^{(i)})p(\boldsymbol{x}_t|\boldsymbol{x}_0^{(i)})\mathbb{1}_{\{y^{(i)}\in\xi\}}}{\sum_{i=1}^N p(\boldsymbol{x}_t|\boldsymbol{x}_0^{(i)})\mathbb{1}_{\{y^{(i)}\in\xi\}}}$$

(26)

$$= \frac{1}{\beta(t)}\frac{\left(\frac{\sum_{i=1}^N(\boldsymbol{x}_t - \alpha(t)\boldsymbol{x}_0^{(i)})p(\boldsymbol{x}_t|\boldsymbol{x}_0^{(i)})\mathbb{1}_{\{y^{(i)}\in\xi\}}}{\sum_{i=1}^N \mathbb{1}_{\{y^{(i)}\in\xi\}}}\right)}{\left(\frac{\sum_{i=1}^N p(\boldsymbol{x}_t|\boldsymbol{x}_0^{(i)})\mathbb{1}_{\{y^{(i)}\in\xi\}}}{\sum_{i=1}^N \mathbb{1}_{\{y^{(i)}\in\xi\}}}\right)}$$

$$= \frac{\mathbb{E}_{\boldsymbol{x}_0\sim p_{\text{data}}(\boldsymbol{x}_0|y\in\xi)}\left[\frac{1}{\beta(t)}(\boldsymbol{x}_t - \alpha(t)\boldsymbol{x}_0)p(\boldsymbol{x}_t|\boldsymbol{x}_0)\right]}{p(\boldsymbol{x}_t|y\in\xi)},$$

(27)

$\forall\ \boldsymbol{x}_t\in\mathbb{R}^m,\ t\in(0,T],\ \xi\in\mathcal{Y}$. As $f(t)$ and $g(t)$ are both continuous functions then $\alpha(t)$ and $\beta(t)$ are also continuous. It follows that (26) is continuous with respect to $\boldsymbol{x}_t$ and $t$, as it is the sum of continuous functions and $\beta(t)^2 > 0\ \forall\ t\in(0,T]$. Thus, the Universal Approximation

Theorem (Cybenko, 1989; Hornik, 1991) holds and there exists a $\theta^\star \in \Theta$ such that $\epsilon_{\theta^\star}(\boldsymbol{x}_t, t, \xi) = \epsilon_\theta^\star(\boldsymbol{x}_t, t, \xi) \; \forall \; \boldsymbol{x}_t \in \mathbb{R}^m$, $t \in (0, T]$, $\xi \in \mathcal{Y}$. Finally, by comparing (24) and (26) we observe that,

$$\nabla_{\boldsymbol{x}_t} \log(p(\boldsymbol{x}_t | y \in \xi)) = -\frac{1}{\beta(t)} \epsilon_{\theta^\star}(\boldsymbol{x}_t, t, \xi),$$

which proves (18), and (19) follows from (27). $\qquad\qquad\square$

It is worth noting that Theorem P.3 unifies the score-model link between conditional and unconditional models. That is to say,

$$\begin{aligned}
\nabla_{\boldsymbol{x}_t} \log(p(\boldsymbol{x}_t)) &= \nabla_{\boldsymbol{x}_t} \log(p(\boldsymbol{x}_t | y \in \Omega_y)) \\
&= -\frac{1}{\beta(t)} \epsilon_{\theta^\star}(\boldsymbol{x}_t, t, \Omega_y).
\end{aligned}$$

