# OpenReview forum: "NatADiff: Adversarial Boundary Guidance for Natural Adversarial Diffusion"
_ICLR.cc/2026/Conference — ICLR 2026 Poster_

### Official Review · Reviewer_n2At · 2025-10-29

**Soundness:** 2
**Presentation:** 3
**Contribution:** 2
**Rating:** 4
**Confidence:** 4

**Summary:**

This paper presents a novel and well-motivated method for generating natural adversarial samples using diffusion models. The proposed NatADiff framework effectively combines adversarial boundary guidance, classifier augmentations, and time-travel sampling to produce highly transferable and semantically meaningful adversarial examples.

This paper introduces NatADiff, a diffusion-based method for generating natural adversarial samples. The key contributions are: 1）It proposes a novel technique that guides the diffusion sampling trajectory toward the intersection of the true and adversarial classes, encouraging the inclusion of adversarial features in a semantically plausible manner; 2) It uses differentiable image transformations to reduce the influence of constrained adversarial perturbations and promote more robust and transferable adversarial features; 3)
 It presents an untargeted attack strategy that selects adversarial classes based on semantic similarity using CLIP embeddings, improving attack success and visual quality.

**Strengths:**

1. The adversarial boundary guidance mechanism is a novel and intuitive way to incorporate adversarial features without deviating entirely from the true class, leading to more plausible and transferable samples.

2. NatADiff achieves significantly higher transferability than existing methods across a wide range of models, including adversarially trained classifiers.

3. The paper includes thorough ablation studies and visualizations that validate design choices and provide interpretability.

**Weaknesses:**

1. The methodological novelty is limited; the core components of NatADiff lack fundamental innovation. Classifier augmentations and time-travel sampling are directly borrowed from prior works to address known issues (e.g., constrained perturbations and sample quality degradation). The main proposed contribution, adversarial boundary guidance, is essentially a weighted interpolation between two established concepts: classifier-free guidance and the novel but straightforward idea of guiding towards a class intersection ($y ∩ ȳ$) using a composite text prompt. While the combination of these elements is new and effective, the paper does not introduce a groundbreaking new algorithm or theoretical insight. It is better characterized as a skillful and well-engineered integration of existing techniques towards a new objective.


2. NatADiff is computationally expensive (103 seconds per sample), primarily due to the iterative diffusion process and time-travel sampling.

3. Experiments are confined to ImageNet. It is unclear how well NatADiff generalizes to other domains. (eg. CUB-200, Stanford Cars in DiffAttack ).

4.  NatADiff often introduces significant semantic changes to the original content, leading to the generation of implausible or unnatural features (e.g., objects with distorted shapes or unrealistic textures). While this may effectively fool classifiers, it limits the applicability of the attack in real-world scenarios where the adversarial sample must remain a faithful representation of the source semantic content. For instance, in content authentication, medical imaging, or any context where the integrity of specific visual features is paramount, such drastic alterations are not permissible and would be easily flagged by a human observer.

5. The experimental comparison is skewed in favor of NatADiff. Methods like DiffAttack and ACA are reference-based attacks; they are constrained to be similar to a given clean source image. In contrast, NatADiff and AdvClass are generative-based attacks that start from random noise and have the freedom to generate any image on the manifold. This fundamental difference grants NatADiff a significantly larger attack surface, as it is not bound by the semantic content of a specific source image. Comparing these two distinct paradigms under the same "unconstrained attack" umbrella is misleading and overstates NatADiff's advantages.

**Questions:**

1. Why does adversarial boundary guidance improve transferability? Is there a theoretical link between class intersection and model shortcut learning?

2. How do humans perceive NatADiff-generated samples? Do they align with human judgment of class membership?

3. Can NatADiff be adapted to a "guided editing" setting, where it modifies a provided source image towards an adversarial target, rather than generating from scratch?

---

> ### Author Response · Authors · 2025-11-21
>
> We appreciate the reviewer's feedback and thank the reviewer for their comments and consideration. We are pleased that the reviewer found our proposed adversarial sampling scheme to be "novel" and "well-motivated". Additionally, we thank the reviewer for acknowledging our SOTA transferability and extensive ablation studies and visualisations.
>
> ## Weaknesses
> (W1): The methodological novelty is limited; the core components of NatADiff lack fundamental innovation...
>
> We thank the reviewer for their time and detailed feedback, and their recognition of adversarial boundary guidance as a novel methodological contribution. We emphasise that our work is the first to identify the core research problem in generating adversarial samples: steering the sampling process toward broad class boundaries, which we believe validates the importance of our proposed method. In contrast to existing approaches that rely naively on classifier gradients to sample from the image manifold within intra-class regions of low interest (e.g., PGD, AutoAttack, NCF, DiffAttack, ACA, AdvClass), we are, to the best of our knowledge, the first to deliberately introduce adversarial structure from an auxiliary source that investigates samples at inter-class boundaries. This design is motivated by current understanding of test-time errors [1,2,3].
>
> By identifying this problem and demonstrating that NatADiff achieves SOTA performance, we establish a new research direction for the field. This could potentially inspire more advanced future works in manipulating sampling trajectories towards class intersections. We also believe that boundary guidance may have broader applications beyond adversarial sampling.
>
> (W2): NatADiff is computationally expensive (103 seconds per sample), primarily due to the iterative diffusion process and time-travel sampling.
>
> We acknowledge that NatADiff is computationally expensive. However, we feel that the increased runtime is acceptable given its SOTA performance and the fact that adversarial sample generation is an offline process. Furthermore, NatADiff is only ~7% slower than the previous SOTA model, ACA [4].
>
> (W3): Experiments are confined to ImageNet. It is unclear how well NatADiff generalizes to other domains. (eg. CUB-200, Stanford Cars in DiffAttack ).
>
> We conducted our study on the ImageNet dataset as the relevant previous works were focused on ImageNet [4,5], with only [5] having a small ablation on MNIST. However, other reviewers have raised this concern, and we are actively working on providing ablations on an alternate dataset. We will hopefully provide the experimental results within the next week, and we will post an additional comment alongside them.
>
> (W4): NatADiff often introduces significant semantic changes to the original content, leading to the generation of implausible or unnatural features (e.g., objects with distorted shapes or unrealistic textures)...
>
> NatADiff is inherently limited by the strength of the underlying diffusion model. We use Stable Diffusion 1.5 (SD1.5), and the quality of our generated samples are consistent with what SD1.5 typically produces. In a realistic attack scenario, an adversary would generally be able to inspect NatADiff outputs before deploying them. Across the 70+ samples shown in the manuscript, we believe a sufficient proportion would pass human scrutiny, indicating that an attacker could fool ImageNet classifiers with high probability.
>
> Regarding alternate domains such as content authentication, medical imaging, or other feature-sensitive settings, we believe it is misleading to draw conclusions solely from the ImageNet results. These domains would require training a diffusion model with a narrower, task-specific generative scope. Such a model would be expected to preserve critical structural information far more reliably than SD1.5, whose broad generative range is not optimised for minute structural details. Thus, if a diffusion model for medical image generation was used, we believe NatADiff would produce sufficiently realistic samples.

---

> > ### Author Response · Authors · 2025-11-21
> >
> > (W5): The experimental comparison is skewed in favor of NatADiff...
> >
> > We believe that our experimental comparisons as fair as possible, and we provide comparisons across a range of adversarial sampling methods: PGD, AutoAttack, NCF, DiffAttack, ACA, and AdvClass. NCF, ACA, and AdvClass all claim to be "unrestricted/unconstrained" [4,5,6], and they can make significant changes to an image (see Figure 3). Furthermore, ACA outperforms both targeted and untargeted variants of AdvClass (with respect to transferability), and occasionally outperforms targeted versions of NatADiff (see Table 1). Thus, we are concerned that comparing NatADiff to AdvClass alone would actually overstate NatADiff's performance, as NatADiff would be both best and second-best across all victim models.
> >
> > That said, we acknowledge that only NatADiff and AdvClass have a truly unrestricted attack surface and we have updated the manuscript to make this distinction clearer (see line 356-358).
> >
> > ## Questions
> > (Q1): Why does adversarial boundary guidance improve transferability? Is there a theoretical link between class intersection and model shortcut learning?
> >
> > This is a very interesting question, and we thank the reviewer for raising it. Prior work has investigated the nature of natural adversarial samples (i.e., test-time errors) [1, 2, 3]. These works hypothesise that such errors arise from “shortcut learning,” where a model relies on spurious contextual cues rather than truly discriminative class features. These shortcuts are typically easy-to-learn features that frequently co-occur with the true class but are not themselves indicative of class identity. When these shortcut features appear without the class—or when the class appears without those features—the model misclassifies. Moreover, classifiers often converge to the same shortcut features, which contributes to the high transferability of natural adversarial samples [1].
> >
> > Our adversarial boundary guidance mechanism aims to target this phenomenon. By introducing features from the adversarial class that frequently co-occur with it--without introducing the adversarial class itself--we seek to intentionally inject the kinds of contextual signals that drive test-time errors. In this way, our method aims to explicitly attack the underlying mechanism thought to give rise to natural adversarial samples.
> >
> > (Q2): How do humans perceive NatADiff-generated samples? Do they align with human judgment of class membership?
> >
> > We provide over 70 NatADiff samples in the manuscript, each shown with its true and adversarial class labels. These examples demonstrate that the generated images generally appear as, or very close to, the intended “true” class. To further support this claim, we will conduct a small human study during the review period. We will report the results later in the week.
> >
> > (Q3): Can NatADiff be adapted to a "guided editing" setting, where it modifies a provided source image towards an adversarial target, rather than generating from scratch?
> >
> > This is an exciting research question, and we thank the reviewer for the suggestion. In principle, NatADiff could be adapted to modify a source image toward an adversarial target using mechanisms similar to SDEdit [7] or null-text inversion [8]. In such an approach, the source image would be partially noised and the reverse trajectory adjusted to incorporate adversarial features.
> >
> > Due to hardware and time constraints, we cannot guarantee that we will be able to provide a demonstration of this adaptation during the review period, as it would require substantial changes to our current codebase. However, we are genuinely interested in exploring this direction and will report the results of any experiments we are able to complete during the review process.

---

> > > ### Author Response · Authors · 2025-11-21
> > >
> > > ## References
> > > [1] Dan Hendrycks, Kevin Zhao, Steven Basart, Jacob Steinhardt, and Dawn Song. Natural adversarial examples, CVPR, 2021
> > >
> > > [2] Robert Geirhos, Jörn-Henrik Jacobsen, Claudio Michaelis, Richard Zemel, Wieland Brendel, Matthias Bethge, and Felix A. Wichmann. Shortcut learning in deep neural networks. Nature Machine Intelligence
> > >
> > > [3] Martin Arjovsky, Léon Bottou, Ishaan Gulrajani, and David Lopez-Paz. Invariant risk minimization. arXiv preprint arXiv:1907.02893, 2020
> > >
> > > [4] Zhaoyu Chen, Bo Li, Shuang Wu, Kaixun Jiang, Shouhong Ding, and Wenqiang Zhang. Content-based unrestricted adversarial attack. In Advances in Neural Information Processing Systems 36, NeurIPS, 2023
> > >
> > > [5] Xuelong Dai, Kaisheng Liang, and Bin Xiao. AdvDiff: Generating unrestricted adversarial examples using diffusion models. In European Conference on Computer Vision, ECCV, 2024.
> > >
> > > [6] Shengming Yuan, Qilong Zhang, Lianli Gao, Yaya Cheng, and Jingkuan Song. Natural Color Fool: Towards boosting black-box unrestricted attacks. In Sanmi Koyejo, S. Mohamed, A. Agarwal, Danielle Belgrave, K. Cho, and A. Oh (eds.), Advances in Neural Information Processing Systems 35, NeurIPS, 2022
> > >
> > > [7] SDEdit: Chenlin Meng, Yutong He, Yang Song, Jiaming Song, Jiajun Wu, Jun-Yan Zhu & Stefano Ermon. SDEdit: Guided Image Synthesis and Editing with Stochastic Differential Equations. ICLR 2022
> > >
> > > [8] Ron Mokady, Amir Hertz, Kfir Aberman, Yael Pritch & Daniel Cohen-Or. Null-text Inversion for Editing Real Images using Guided Diffusion Models. CVPR 2023

---

> ### Author Response · Authors · 2025-12-03
>
> We thank the reviewer for their patience. We have added the requested ablations to address concerns regarding NatADiff's generalisability to additional datasets and its perceived image class under human evaluation.
>
> ## Weaknesses / Questions
>
> (W3): In Appendix I we provide additional experiments on the Oxford Pets dataset which is a fine-grained dataset suggested by reviewer BqxE. We observe that NatADiff behaves similarly to the ImageNet study, outperforming other methods in terms of attack transferability while maintaining competitive white-box ASR and image quality. These results are consistent with the strong performance of similarity-targeted attacks in the ImageNet study, indicating that adversarial boundary guidance is particularly effective when classes share similar structural elements.
>
> (Q2): In Appendix O we conduct a user study to determine human agreement with NatADiff's target true class. We showed users 60 randomly selected NatADiff samples and found that participants reported seeing the true class in $91 \%$ of cases, the adversarial class in $7 \%$, and neither class in $2 \%$. NatADiff's label-flip rate of $9 \%$ is comparable to the $10 \%$ flip rate reported by [1] for adversarial classifier guidance on the MNIST dataset. These results indicate that NatADiff reliably produces images that appear to belong to the true class for human observers, even while fooling classifiers with high probability. Moreover, in practical attack settings, an adversary could manually inspect generated images prior to deployment, further mitigating the impact of occasional class-flipped samples.
>
> (Q3): Unfortunately, we did not have sufficient time to implement the substantial codebase changes required to support image editing. However, we again thank the reviewer for the suggestion, and we agree that this remains an exciting direction for future work.
>
> ## References
>
> [1] Xuelong Dai, Kaisheng Liang, and Bin Xiao. AdvDiff: Generating unrestricted adversarial examples using diffusion models. In European Conference on Computer Vision, ECCV, 2024

---

### Official Review · Reviewer_6oxz · 2025-10-30

**Soundness:** 3
**Presentation:** 3
**Contribution:** 2
**Rating:** 6
**Confidence:** 3

**Summary:**

This study introduces NatADiff, an adversarial sampling framework that leverages a denoising diffusion model to synthesize "natural adversarial examples" by guiding sampled images toward the intersection of the manifold between true and adversarial categories. The authors propose an adversarial boundary guidance method to generate adversarial examples that are more realistic and transferable than baseline attack strategies. Empirical results on ImageNet classifiers and a range of model architectures demonstrate that this approach improves attack transferability and approaches true test-time error.

**Strengths:**

1. This paper connects the concept of adversarial bound guidance with the generation of natural adversarial examples, formalizing the intuition that natural errors occur due to overreliance on contextual cues. This is a well-thought-out approach.
2. The method demonstrates excellent technical depth and implementation. It combines time-travel sampling, classifier-free guidance, gradient normalization, and boosting; ablation experiments are conducted for each component to evaluate its contribution.
3. Experimental results demonstrate that NatADiff achieves comparable or even higher attack success rates than state-of-the-art methods in both white-box and transfer settings. The high transferability of its attack is demonstrated across a variety of victim architectures, including convolutional and Transformer-based models.

**Weaknesses:**

1. The manuscript provides no ablation study or discussion on the robustness of the adversarial boundary guidance to variations in the textual implementation of the intersection prompt `y ∩ ỹ`. The stability of results across different prompt engineering strategies (e.g., varying templates or phrasing) remains entirely unexplored.
2. The appendix mentions that the adversarial guidance strength was "manually tuned" and notes that s behaves close to binarization (the attack succeeds only after reaching a threshold). The paper mentions "the optimal value of s varied across classifiers" and reports the specific values in Table 4 but does not verify how s affects gradient stability or vanishing/divergence.
3. All experiments were conducted only on ImageNet and its commonly used architectures. Whether this method can be extended to non-image data modalities, domain-transfer scenarios, or truly open-world environments has not been discussed or empirically verified.

**Questions:**

1. Could the authors provide more technical details or ablation results on the construction and effect of the intersection prompt `y ∩ ỹ`? Specifically, how stable are the results under different prompt generation strategies?
2. Could the authors provide rigorous analysis or experimental results on how the adversarial guidance strength s influences gradient vanishing or explosion phenomena, particularly given its threshold-like behavior and dependence on classifier architectures?
3. The paper has a great idea. What needs to be done to adapt NatADiff to domains beyond natural images (e.g., audio, tables, or multimodal data)? Are there any inherent limitations?

---

> ### Author Response · Authors · 2025-11-21
>
> We appreciate the reviewer's feedback and thank the reviewer for their comments and consideration. We are pleased that the reviewer found our proposed adversarial sampling scheme to be "well-thought-out" and that they felt it "demonstrates excellent technical depth and implementation". Additionally, we thank the reviewer for acknowledging our SOTA performance across a variety of classifier architectures.
>
> ## Weaknesses
> (W1): The manuscript provides no ablation study or discussion on the robustness of the adversarial boundary guidance to variations in the textual implementation...
>
> We thank the reviewer for suggesting this insightful ablation study. We are currently running expriments to address this during the review period. Due to hardware constraints these will be provided within the next week and we will post an additional comment alongside them.
>
> (W2): The appendix mentions that the adversarial guidance strength was "manually tuned" and notes that s behaves close to binarization...
>
> We thank the reviewer for requesting this clarification. We aim to address this with additional ablations during the review period. Due to hardware constraints these will be provided within the next week and we will post an additional comment alongside them.
>
> (W3): All experiments were conducted only on ImageNet and its commonly used architectures...
>
> We conducted our study on the ImageNet dataset as the relevant previous works were focused on ImageNet [1, 2], with only [2] having a small ablation on MNIST. Furthermore all previous work discussed in the manuscript was solely focused on static images. We feel that extending NatADiff to other domains and modalities is an exciting opportunity for future work, but is beyond the scope of this study.
>
> ## Questions
> (Q1): Could the authors provide more technical details or ablation results on the construction and effect of the intersection prompt...
>
> As discussed in (W1), we will be providing ablation studies to examine the robustness of adversarial boundary guidance to variations in the textual prompt. Due to hardware constraints these will be provided within the next week and we will post an additional comment alongside them.
>
> (Q2): Could the authors provide rigorous analysis or experimental results on how the adversarial guidance strength...
>
> As discussed in (W2), we will be providing ablation studies to examine the effect of the adversarial classifier guidance strength, $s$, on the classifier gradient. Due to hardware constraints these will be provided within the next week and we will post an additional comment alongside them.
>
> (Q3): The paper has a great idea. What needs to be done to adapt NatADiff to domains beyond natural images (e.g., audio, tables, or multimodal data)?
>
> We thank the reviewer for their appreciation and interest in our method. Fundamentally, NatADiff can be extended to any domain **provided** their exists a diffusion-based generative model with sufficient strength to construct in-domain samples for the $y$ and $y \cap \tilde{y}$ conditions. In our study we used a text-to-image model to facilitate this, but it would theoretically be possible to train a model specifically for the task. Furthermore, the method is not tied to images and could in theory be applied to other modalities.
>
> ## References
>
> [1] Zhaoyu Chen, Bo Li, Shuang Wu, Kaixun Jiang, Shouhong Ding, and Wenqiang Zhang. Content-based unrestricted adversarial attack. In Advances in Neural Information Processing Systems 36, NeurIPS, 2023
>
> [2] Xuelong Dai, Kaisheng Liang, and Bin Xiao. AdvDiff: Generating unrestricted adversarial examples using diffusion models. In European Conference on Computer Vision, ECCV, 2024

---

> > ### Comment · Reviewer_6oxz · 2025-11-26
> >
> > We look forward to seeing concrete experimental results within a week regarding the robustness of the text cue word variants and the guidance strength of the adversarial classifier, which will be crucial for evaluating the robustness and generalization of the method.

---

> ### Author Response · Authors · 2025-12-03
>
> We thank the reviewer for their patience. We have added the requested ablations to address concerns regarding NatADiff's performance across varying boundary-guidance prompt formats and the stability of the classifier gradient.
>
> ## Weaknesses / Questions
>
> (W1) / (Q1): In Appendix G.3 we provide ablations on the stability and effect of the intersection prompt, $\tilde{y} \cap y$. We find that the prompt structure exerted relatively minimal influence on attack performance or image quality. The average ASR varied by less than $7.7\%$ across all prompt formats, indicating that adversarial boundary guidance is robust to the particular phrasing used to express the class intersection. Furthermore, we found that the prompt construction used in the main paper afforded the best tradeoff between attack success rate and image quality.
>
> (W2) / (Q2): In Appendix G.4 we provide ablations on the classifier guidance gradient across various classifier architectures and guidance strengths. We observe that attack success rate increases with adversarial classifier guidance strength. When the guidance strength is too low, NatADiff's ASR drops, and the gradient does not smoothly converge to $0$ as $t \to 0$. This is likely because the guidance is insufficient to push the sample into regions of the adversarial class on the classifier manifold—evidenced by low ASR scores coinciding with weak guidance. Conversely, sufficiently large guidance strengths yield substantially higher ASR and gradients that smoothly converge to $0$ as the sample enters the desired region of the manifold. Furthermore, we did not encounter instances of exploding or vanishing gradients, which is in line with existing work on classifier guided diffusion [1, 2].
>
> Overall, these results indicate that adversarial classifier guidance operates according to the same underlying mechanisms as standard classifier-guided diffusion [1, 2].
>
> ## References
>
> [1] Prafulla Dhariwal and Alexander Quinn Nichol. Diffusion models beat gans on image synthesis. In Advances in Neural Information Processing Systems 34, NeurIPS, 2021.
>
> [2] Yifei Shen, Xinyang Jiang, Yifan Yang, Yezhen Wang, Dongqi Han, and Dongsheng Li. Understanding and improving training-free loss-based diffusion guidance. In Advances in Neural Information Processing Systems 38, NeurIPS, 2024.

---

### Official Review · Reviewer_rbeT · 2025-10-30

**Soundness:** 3
**Presentation:** 3
**Contribution:** 3
**Rating:** 8
**Confidence:** 3

**Summary:**

This paper proposes NatADiff, a diffusion-based method for generating highly transferable natural adversarial samples. It introduces adversarial boundary guidance and classifier augmentations to steer diffusion sampling toward class intersections, mimicking naturally occurring test-time errors. NatADiff achieves strong white-box attack success rates and significantly higher transferability than existing methods, with samples more closely resembling natural adversarial examples.

**Strengths:**

- Novel integration of adversarial boundary guidance and classifier augmentations in diffusion models.
- Strong empirical results: high transferability and competitive white-box performance.
- Well-motivated by the link between contextual cues and natural adversarial samples.
- Comprehensive evaluation across multiple architectures and adversarial defenses.

**Weaknesses:**

- Computationally expensive due to iterative diffusion sampling.
- Limited to ImageNet; evaluation on more specialized domains is future work.
- Similarity targeting may lead to subtle misclassifications (e.g., between similar classes).

**Questions:**

- Why do you only limit to ImageNet?
- Alg 1: What does the star after $\epsilon$ mean?
- Will you provide your code?
- You use the RTX 4090. You havn't specified the memory size. Maybe only 24 GB. Can you only attack one image after another or more images at once?

---

> ### Author Response · Authors · 2025-11-21
>
> We thank the reviewer for their feedback and consideration. We are pleased the reviewer found our proposed adversarial sampling scheme to be "novel" and "well-motivated by the link between contextual cues and natural adversarial samples" (test-time errors). We appreciate the reviewer acknowledging our comprehensive evaluation across model architectures and adversarial defences.
>
> ## Weaknesses
>
> (W1): The reviewer highlights that NatADiff is computationally expensive.
>
> We acknowledge that NatADiff is computationally expensive. However, we feel that the increased runtime is acceptable given its SOTA performance and the fact that adversarial sample generation is an offline process. Furthermore, NatADiff is only ~7% slower than the previous SOTA model, ACA [1].
>
> (W2): The reviewer highlights that we only apply NatADiff to ImageNet classifiers.
>
> We conducted our study on the ImageNet dataset as the relevant previous works were focused on ImageNet [1, 2], with only [2] having a small ablation on MNIST. However, other reviewers have raised this concern, and we are actively working on providing ablations on an alternate dataset. We will hopefully provide the experiments within the next week, and we will post an additional comment alongside them.
>
> (W3): The reviewer points out that similarity targeting can lead to subtle misclassifications.
>
> We acknowledge that similarity targeting can lead to subtle misclassifications, e.g., between breeds of dog. However, we note in Appendix A that a potential refinement would be to surface a ranked list of similar classes, allowing users to select more divergent adversarial targets while retaining the semantic grounding of similarity-based selection.
>
> ## Questions
>
> (Q1): Why do you only limit to ImageNet?
>
> We address this in our response to (W1).
>
> (Q2): Alg 1: What does the star after mean?
>
> The star after $\epsilon$ denotes an optimally trained diffusion model. This is a reference to the implicit assumption that all denoising diffusion models make when using $\epsilon$ to approximate the score function--the approximation is only valid if the diffusion model is optimal (see Theorem M.3 in the Appendix for additional details). We have added further clarification on line 151 of the updated manuscript.
>
> (Q3): Will you provide your code?
>
> The code is included as part of the supplementary materials. All configuration files and environment details necessary to reproduce our results are provided. Furthermore, the codebase will be made public on GitHub after paper acceptance.
>
> (Q4): You use the RTX 4090. You haven't specified the memory size. Maybe only 24 GB. Can you only attack one image after another or more images at once?
>
> Our 4090 has 24GB of VRAM. With this memory capacity we can generate 2 NatADiff samples at once.
>
> ## References
>
> [1] Zhaoyu Chen, Bo Li, Shuang Wu, Kaixun Jiang, Shouhong Ding, and Wenqiang Zhang. Content-based unrestricted adversarial attack. In Advances in Neural Information Processing Systems 36, NeurIPS, 2023
>
> [2] Xuelong Dai, Kaisheng Liang, and Bin Xiao. AdvDiff: Generating unrestricted adversarial examples using diffusion models. In European Conference on Computer Vision, ECCV, 2024

---

> ### Author Response · Authors · 2025-12-03
>
> We thank the reviewer for their patience. We have added an additional ablation to demonstrate NatADiff's generalisability to additional datasets.
>
> ## Weaknesses/Questions
>
> (W2) / (Q1): In Appendix I we provide additional experiments on the Oxford Pets dataset. We observe that NatADiff behaves similarly to the ImageNet study, outperforming other methods in terms of attack transferability while maintaining competitive white-box ASR and image quality. These results are consistent with the strong performance of similarity-targeted attacks in the ImageNet study, indicating that adversarial boundary guidance is particularly effective when classes share similar structural elements.

---

### Official Review · Reviewer_BqxE · 2025-10-31

**Soundness:** 3
**Presentation:** 3
**Contribution:** 3
**Rating:** 4
**Confidence:** 3

**Summary:**

This paper proposes NatADiff, a novel diffusion-based sampling scheme designed to generate natural adversarial samples. The method is motivated by the observation that deep learning models often rely on “contextual cues” to perform “shortcut learning”, which is hypothesized to be the cause of natural test-time errors. The core methodology is termed “Adversarial Boundary Guidance”, a technique that guides the diffusion trajectory toward the intersection of the true and adversarial class manifolds, notably implemented by manipulating text prompts (e.g., “True Class and Adversarial Class”). To enhance attack transferability, this method is combined with “augmented classifier guidance”, and “time-travel sampling” is used to preserve image quality. Experimental results demonstrate that while NatADiff achieves comparable white-box ASR to SOTA methods, it exhibits vastly superior transferability. Furthermore, the generated samples prove effective even against adversarially trained models, suggesting the attack targets a more fundamental vulnerability than traditional perturbation-based attacks.

**Strengths:**

(S1) Vastly Superior Transferability
The paper's primary strength lies in its demonstration of vastly superior transferability. As shown in Table 1, the proposed NatADiff significantly outperforms all competitors, including SOTA diffusion-based attacks like AdvClass and DiffAttack, in average transfer ASR. The ability to successfully attack a ViT-H model with samples generated from a CNN (RN-50) at such a high success rate strongly suggests the method identifies fundamental, architecture-agnostic vulnerabilities.
(S2) Novel and Well-Motivated Methodology
This strong empirical result is supported by a novel and well-motivated methodology. The core idea of “Adversarial Boundary Guidance”, which leverages the diffusion model’s text-conditioning to guide the sampling trajectory toward the intersection of the true and adversarial classes (e.g., using an “A and B” prompt), is a clever way to operationalize the “shortcut learning” hypothesis. This is logically combined with “augmented classifier guidance” to suppress non-transferable, pixel-level perturbations and force the model to manifest more robust structural features.
(S3) Effectiveness Against Adversarial Defenses
The effectiveness of NatADiff against adversarially trained models (AdvRes and AdvInc) is a significant finding. These models are specifically designed to resist traditional perturbation-based attacks, and NatADiff’s success against them underscores that it operates via a fundamentally different and more robust attack vector.
(S4) Insightful Analysis of Attack Variants
Finally, the paper provides an insightful analysis of the trade-off between the random-targeted (T) and similarity-untargeted (U) variants. The observation that the (T) variant achieves a better FID-A score (alignment with natural errors) at the cost of ASR, while the (U) variant does the opposite, lends support to the hypothesis that natural errors often arise from blending features of disparate classes.

**Weaknesses:**

(W1) Inconsistent motivation for Similarity Targeting: The authors motivate the use of similarity targeting (U) by stating that it “outperform[s] targeted attacks (T)”. While this holds true for CNN surrogates (RN-50, Inc-v3), the paper fails to acknowledge or analyze the contradictory result from the ViT-H surrogate, where the random targeted attack (T) significantly outperforms the similarity-based untargeted attack (U) in average ASR (73.2% vs 69.7%). This omission weakens the claim that similarity targeting is a universally superior strategy, especially for Transformer-based models.
(W2) Lack of Direct Evidence for the “Class Intersection” Claim: The paper’s core hypothesis is that NatADiff guides the trajectory towards the “intersection of the true and adversarial classes” 6. However, the evidence provided is indirect (FID-A scores) or qualitative (image samples). The paper would be significantly strengthened by providing direct, quantitative evidence. For example, a t-SNE/UMAP visualization plotting the embeddings of the generated samples against the manifolds of the true and adversarial classes would be necessary to truly validate that the generated samples lie in this hypothesized “intersection”.
(W3) Limited Scope of Evaluation: The evaluation is confined to the broad, 1000-class ImageNet dataset. It remains unclear how the “contextual cue” hypothesis holds on fine-grained datasets (e.g., Oxford-Pet) where inter-class similarity is high and shortcuts might be far more subtle. While the authors acknowledge this limitation in the appendix, an experiment on a fine-grained dataset would have substantially strengthened the paper's claims of generalizability.
(W4) In Equation 10, it seems you should use argmax instead of argmin. In this equation, you should specify the class with the highest similarity, not the lowest, as the adversarial target.

**Questions:**

Concerns are in the weakness section

**Details Of Ethics Concerns:**

Not ethical problems

---

> ### Author Response · Authors · 2025-11-21
>
> We appreciate the reviewer's feedback and thank the reviewer for their comments and consideration. We are pleased that the reviewer found our proposed adversarial sampling scheme to be "novel and well-motivated", and that they appreciated its superior transferability and resistance to adversarial defences.
>
> ## Weaknesses
>
> (W1): The reviewer notes an apparent inconsistency in the performance of similarity targeting with respect to ViT-H versus ResNet and Inception classifiers.
>
> In the main paper we state that untargeted attacks "frequently outperform targeted variants" (line 296). The apparent contradiction with respect to ViT-H arises because the targeted ViT-H samples show degraded image quality, as reflected by the IS and FID-Val scores in Table 1. Targeted attacks typically blend features from disparate classes, which places greater demands on the diffusion model to locate a feasible point on the image manifold. ViT-H is a strong classifier, and its decision boundaries between unrelated classes appear to be more accurate than those of the other classifiers we examined. As a result, the diffusion model struggles to generate feasible targeted adversarial samples for ViT-H, and artifacts are introduced. These artifacts substantially degrade image quality (see Appendix I.5 of the updated manuscript) and artificially inflate the attack success rate of the targeted ViT-H samples (as seen in Table 1). Importantly, these artifacts occur only in the ViT-H targeted setting, which is directly observable in the image-quality metrics in Table 1.
>
> We discuss this effect on lines 470–471, but did not explicitly connect the explanation to the ViT-H results. This has now been clarified in the updated manuscript (see lines 452-461 and Appendix I.5).
>
> (W2): The reviewer requests qualitative evidence that NatADiff samples lie in the hypothesized class “intersection”.
>
> We thank the reviewer for their suggestion and will be following through on t-SNE/UMAP visualisations during the review period. Due to hardware constraints these will be provided within the next week and we will post an additional comment alongside them.
>
> (W3): The reviewer highlights that we only apply NatADiff to ImageNet classifiers.
>
> We conducted our study on the ImageNet dataset as the relevant previous works were focused on ImageNet [1, 2], with only [2] having a small ablation on MNIST. However, we thank the reviewer for their feedback and agree that an ablation on a fine-grained dataset would strengthen the claims made in the main paper. We are currently working on this task and will provide the experimental results within the next week--we will post an additional comment alongside them.
>
> (W4) The reviewer notes a typo in the manuscript.
>
> We thank the reviewer for catching the typo in Equation 10. This has been corrected in the updated manuscript. Furthermore, we wish to reassure the reviewer that this was just a typo as can be seen from the implementation in the codebase provided in the supplementary materials.
>
> ## References
>
> [1] Zhaoyu Chen, Bo Li, Shuang Wu, Kaixun Jiang, Shouhong Ding, and Wenqiang Zhang. Content-based unrestricted adversarial attack. In Advances in Neural Information Processing Systems 36, NeurIPS, 2023
>
> [2] Xuelong Dai, Kaisheng Liang, and Bin Xiao. AdvDiff: Generating unrestricted adversarial examples using diffusion models. In European Conference on Computer Vision, ECCV, 2024

---

> > ### Comment · Reviewer_BqxE · 2025-11-27
> >
> > Thank you for reply
> > I will wait the additional results before Dec 03.

---

> ### Author Response · Authors · 2025-12-03
>
> We thank the reviewer for their patience. We have added the requested ablations to address concerns regarding the placement of NatADiff samples on the image manifold and the method's generalisability to additional datasets.
>
> ## Weaknesses
>
> (W2): In Appendix H we provide t-SNE and UMAP visualisations of NatADiff samples alongside the manifolds of the true and adversarial classes. We find that across both embeddings, NatADiff samples consistently form one or more distinct clusters that fall between the manifolds of the true and adversarial target classes. This positioning directly supports the main paper's claim: NatADiff guides samples toward class intersections, generating samples that fall near class boundaries.
>
> (W3): In Appendix I we provide additional experiments on the Oxford Pets dataset. We observe that NatADiff behaves similarly to the ImageNet study, outperforming other methods in terms of attack transferability while maintaining competitive white-box ASR and image quality. These results are consistent with the strong performance of similarity-targeted attacks in the ImageNet study, indicating that adversarial boundary guidance is particularly effective when classes share similar structural elements.

---

### Author Response · Authors · 2025-12-03

We present NatADiff, a generative adversarial sampling scheme that leverages mechanisms believed to underlie naturally occurring test-time errors. Reviewers BqxE and rbeT highlighted the novelty of our method, while reviewers 6oxz and n2At praised its technical depth and implementation quality. All reviewers acknowledged NatADiff's strong attack transferability.

A key contribution of our work is identifying what we argue is the central research problem in generating adversarial samples: steering the sampling process toward inter-class boundaries. To the best of our knowledge, NatADiff is the first method to deliberately introduce adversarial structure from an auxiliary source that probes sample behaviour specifically at inter-class boundaries—an approach motivated by current understanding of test-time errors [1,2,3].

During the rebuttal we conducted several additional experiments to address reviewer concerns:

- Appendix H (Rev. BqxE): t-SNE and UMAP visualisations show that NatADiff samples lie between the true- and adversarial-class manifolds, supporting our claim that adversarial boundary guidance drives diffusion trajectories toward classifier decision boundaries.
- Appendix I (Rev. BqxE, rbeT, n2At): Applying NatADiff to the Oxford Pets dataset demonstrates that our approach generalises to fine-grained classification tasks, achieving SOTA transferability on these more subtle misclassifications.
- Appendix G.3 (Rev. 6oxz): Ablations on adversarial guidance prompt structure show that the precise formulation of the intersection prompt has only a minimal effect on attack performance and image quality.
- Appendix G.4 (Rev. 6oxz): Ablations on adversarial classifier guidance strength demonstrate behaviour analogous to standard classifier-guided diffusion, with an optimal guidance value that is classifier-specific. We also observe no meaningful gradient-stability issues.
- Appendix O (Rev. n2At): A user study with 22 participants shows 91% agreement with the NatADiff “true class,” supporting the semantic validity of the generated samples.

In summary, we identify and address the core problem in adversarial sample generation: guiding the sampling process toward broad class boundaries. By explicitly probing inter-class regions, NatADiff achieves SOTA performance and establishes a promising direction for future research on manipulating adversarial sampling trajectories towards class intersections.

## References

[1] Dan Hendrycks, Kevin Zhao, Steven Basart, Jacob Steinhardt, and Dawn Song. Natural adversarial examples, CVPR, 2021

[2] Robert Geirhos, Jörn-Henrik Jacobsen, Claudio Michaelis, Richard Zemel, Wieland Brendel, Matthias Bethge, and Felix A. Wichmann. Shortcut learning in deep neural networks. Nature Machine Intelligence

[3] Martin Arjovsky, Léon Bottou, Ishaan Gulrajani, and David Lopez-Paz. Invariant risk minimization. arXiv preprint arXiv:1907.02893, 2020

---

### Meta-Review · Area_Chair_JYdP · 2026-01-05

**Summary:**

The paper proposes NatADiff, a method for generating natural adversarial examples using diffusion models by guiding the sampling trajectory toward the intersection of true and adversarial class manifolds. The authors introduce Adversarial Boundary Guidance to create transferable and semantically plausible adversarial samples.

**Reviewer Concerns:**

The reviewers initially appreciated the strong transferability results and the motivation linking adversarial samples to shortcut learning and class boundary intersections. However, the initial reception was mixed (Scores: 4, 8, 6, 4). The primary concerns centered on the limited scope of evaluation (ImageNet only), the lack of direct quantitative evidence for the class intersection hypothesis, questions regarding the robustness of the prompt engineering, and general concerns about computational cost and semantic validity of the generated images.

**Reviewer Scores:**

Given that the authors directly and successfully addressed the specific empirical deficits raised by the negative reviewers (BqxE and n2At), the decision leans strongly towards Acceptance.

---

### Decision · Program_Chairs · 2026-01-26

Accept (Poster)